# Transformers as Unsupervised Learning Algorithms:
# A study on Gaussian Mixtures

**Zhiheng Chen**[1*]**, Ruofan Wu**[2*]**, Guanhua Fang**[2†]
[1]Shanghai Center for Mathematical Sciences, Fudan University,
[2]Department of Statistics and Data Science, School of Management, Fudan University
`zhchen22@m.fudan.edu.cn, wuruofan1989@gmail.com, fanggh@fudan.edu.cn`

## Abstract

The transformer architecture has demonstrated remarkable capabilities in modern artificial intelligence, among which the capability of implicitly learning an internal model during inference time is widely believed to play a key role in the understanding of pre-trained large language models. However, most recent works have been focusing on studying supervised learning topics such as in-context learning, leaving the field of unsupervised learning largely unexplored. This paper investigates the capabilities of transformers in solving Gaussian Mixture Models (GMMs), a fundamental unsupervised learning problem through the lens of statistical estimation. We propose a transformer-based learning framework called Transformer for Gaussian Mixture Models (TGMM) that simultaneously learns to solve multiple GMM tasks using a shared transformer backbone. The learned models are empirically demonstrated to effectively mitigate the limitations of classical methods such as Expectation-Maximization (EM) or spectral algorithms, at the same time exhibit reasonable robustness to distribution shifts. Theoretically, we prove that transformers can efficiently approximate both the Expectation-Maximization (EM) algorithm and a core component of spectral methods—namely, cubic tensor power iterations. These results not only improve upon prior work on approximating the EM algorithm, but also provide, to our knowledge, the first theoretical guarantee that transformers can approximate high-order tensor operations. Our study bridges the gap between practical success and theoretical understanding, positioning transformers as versatile tools for unsupervised learning. [1]

## 1 Introduction

Large Language Models (LLMs) have achieved remarkable success across various tasks in recent years. Transformers(Vaswani et al., 2017), the dominant architecture in modern LLMs(Brown et al., 2020), outperform many other neural network models in efficiency and scalability. Beyond language tasks, transformers have also demonstrated strong performance in other domains, such as computer vision(Han et al., 2023; Khan et al., 2022) and reinforcement learning(Li et al., 2023a). Given their practical success, understanding the mechanisms behind transformers has attracted growing research interest. Existing studies often treat transformers as algorithmic toolboxes, investigating their ability to implement diverse algorithms(Von Oswald et al., 2023; Bai et al., 2023; Lin et al., 2024; Giannou et al., 2025; Teh et al., 2025)–a perspective linked to meta-learning(Hospedales et al., 2021).

However, most research has focused on supervised learning settings, such as regression(Bai et al., 2023) and classification(Giannou et al., 2025), leaving the unsupervised learning paradigm relatively unexplored. Since transformer models are typically trained in a supervised manner, unsupervised learning poses inherent challenges for transformers due to the absence of labeled data. Moreover, given the abundance of unlabeled data in real-world scenarios, investigating the mechanisms of

---

[*]Equal contribution.
[†]Corresponding to: `fanggh@fudan.edu.cn`
[1]Code available at `https://github.com/Rorschach1989/transformer-for-gmm`

transformers in unsupervised learning holds significant implications for practical applications. The Gaussian mixture model (GMM) represents one of the most fundamental unsupervised learning tasks in statistics, with a rich historical background(DAY, 1969; Aitkin & Wilson, 1980) and ongoing research interest(Zhang et al., 2021; Manduchi et al., 2021; Löffler et al., 2021; Ndaoud, 2022; Gribonval et al., 2021; Yu et al., 2021). Two primary algorithmic approaches are existing for solving GMM problems: (1) likelihood-based methods employing the Expectation-Maximization (EM) algorithm(Dempster et al., 1977; Balakrishnan et al., 2017), and (2) moment-based methods utilizing spectral algorithms(Hsu & Kakade, 2013; Anandkumar et al., 2014). However, both algorithms have inherent limitations. The EM algorithm is prone to convergence at local optima and is highly sensitive to initialization(Moitra, 2018; Jin et al., 2016). In contrast, while the spectral method is independent of initialization, it requires the number of components to be smaller than the data's dimensionality—an assumption that restricts its applicability to problems involving many components in low-dimensional GMMs(Hsu & Kakade, 2013).

In this work, we explore transformers for GMM parameter estimation to address two questions. (i) Can Transformers *provably* work for GMM in-context? (ii) Can Transformers *empirically* overcome the drawbacks of both EM algorithm and the spectral method? Our answers are affirmative. We find that meta-trained transformers exhibit strong performance on GMM tasks without the aforementioned limitations. Notably, we construct transformer-based solvers that efficiently solve GMMs with varying component counts simultaneously. The experimental phenomena are further backed up by novel theoretical establishments: We prove that transformers can effectively learn GMMs with different components by approximating both the EM algorithm and a key component of spectral methods on GMM tasks.

**Main Contributions**.
- We propose the TGMM framework that utilizes transformers to solve multiple GMM tasks with varying numbers of components simultaneously during inference time. Through extensive experimentation, the learned TGMM model is demonstrated to achieve competitive and robust performance over synthetic GMM tasks. Notably, TGMM outperforms the popular EM algorithm in terms of estimation quality, and approximately matches the strong performance of spectral methods while enjoying better flexibility.

- We establish theoretical foundations by proving that transformers can approximate both the EM algorithm and a key component of spectral methods. Our approximation of the EM algorithm fundamentally leverages the weighted averaging property inherent in softmax attention, enabling simultaneous approximation of both the E and M steps. Notably, our approximation results also hold across varying dimensions and mixture components in GMM.

- We proved that transformers (with RELU activation) can implement cubic tensor power iterations-a crucial component of spectral algorithms for GMM. The proof is highly dependent on the multi-head structure of transformers. To the best of our knowledge, this is the first theoretical demonstration of transformers' capacity for high-order tensor calculations.

**Related works.** Recent research has explored the mechanisms by which transformers can implement various supervised learning algorithms. For instance, Akyürek et al. (2023), Von Oswald et al. (2023), and Bai et al. (2023) demonstrate that transformers can perform gradient descent for linear regression problems in-context. Lin et al. (2024) shows that transformers are capable of implementing Upper Confidence Bound (UCB) algorithms, as well as other classical algorithms in reinforcement learning tasks. Giannou et al. (2025) reveals that transformers can execute in-context Newton's method for logistic regression problems. Teh et al. (2025) illustrates that transformers can approximate Robbins' estimator and solve Naive Bayes problems. Kim et al. (2024) studies the minimax optimality of transformers on nonparametric regression. Some literature on density estimation using LLMs is discussed in Section A.

**Comparison with prior theoretical works in unsupervised learning setting.** Several recent studies have investigated the mechanisms of transformer-based models in mixture model settings(He et al., 2025a; Jin et al., 2024; He et al., 2025b). Among these, He et al. (2025a) establishes that transformers can implement Principal Component Analysis (PCA) and leverages this to GMM clustering. However, their analysis is limited to the two-component case, restricting its broader applicability.

The paper Jin et al. (2024) investigates the in-context learning capabilities of transformers for mixture linear models, a setting that differs from ours. Furthermore, their approximation construction of the transformer is limited to two-component GMMs, leaving the general case unaddressed. While

they assume ReLU as the activation function–contrary to the conventional choice of softmax–their theoretical proofs rely on a key lemma from prior work Pathak et al. (2024) that assumes softmax activation, thereby introducing an inconsistency in their assumptions. The paper He et al. (2025b) studies the performance of transformers on multi-class GMM clustering, a setting closely related to ours. However, our work focuses on *parameter estimation* rather than *clustering*. We give a discussion of our theoretical improvements over their work in detail in the following paragraph. From an empirical perspective, their experiments are conducted on a small-scale transformer, which fails to validate their theoretical claims.

**Sharpness of our results**. Our theoretical analysis fully leverages key architectural components of Transformers: the query-key-value mechanism, multi-head attention, and the properties of the activation function. It is worth pointing out that our result improves the prior work for EM approximation in several points: First, Our analysis shows that Transformers can approximate L-step EM algorithms with just O(L) layers, a significant improvement over prior work (He et al., 2025b) , which requires O(KL) layers (dependent on the number of components K). Second, unlike He et al. (2025b), which needs number of attention heads $M \to +\infty$ to get valid bounds, our results hold with $M = O(1)$, aligning better with real-world designs. Third, our approximation bounds scale polynomially in dimension $d$, unlike He et al. (2025b)'s exponential dependence–a crucial improvement for high-dimensional settings. We believe our results and proofs can offer profound insights for subsequent theoretical research on transformers.

**Organization**. The rest of paper is organized as follows. In Section 2, some background knowledge is introduced. In Section 3, we present the experimental details and findings. The theoretical results are proposed in Section 4, and some discussions are given in Section 5. The proofs and additional experimental results are given in the appendix.

**Notations.** We introduce the following notations. Let $[n] := \{1, 2, \cdots, n\}$. All vectors are represented as column vectors unless otherwise specified. For a vector $v \in \mathbb{R}^d$, we denote $\|v\|$ as its Euclidean norm. For two sequences $a_n$ and $b_n$ indexed by $n$, we denote $a_n = O(b_n)$ if there exists a universal constant $C$ such that $a_n \leq Cb_n$ for sufficiently large $n$.

## 2 METHODOLOGY

### 2.1 PRELIMINARIES

The Gaussian mixture model (GMM) is a cornerstone of unsupervised learning in statistics, with deep historical roots and enduring relevance in modern research. Since its early formalizations(DAY, 1969; Aitkin & Wilson, 1980), GMM has remained a fundamental tool for clustering and density estimation, widely applied across diverse domains. Recent advances have further explored the theoretical foundations of Gaussian Mixture Models (GMMs)(Löffler et al., 2021; Ndaoud, 2022; Gribonval et al., 2021), extended their applications in incomplete data settings(Zhang et al., 2021), and integrated them with deep learning frameworks(Manduchi et al., 2021; Yu et al., 2021). Due to their versatility and interpretability, GMMs remain indispensable in unsupervised learning, effectively bridging classical statistical principles with modern machine learning paradigms. We consider the (unit-variance) isotropic Gaussian Mixture Model with $K$ components, with its probability density function as

$$p(x|\boldsymbol{\theta}) = \sum_{k=1}^{K} \pi_k \phi(x; \mu_k) \,, \tag{1}$$

where $\phi(x; \mu)$ is the standard Gaussian kernel, i.e. $\phi(x; \mu) = \frac{1}{(2\pi)^{d/2}} \exp\left(-\frac{1}{2}(x - \mu)^\top (x - \mu)\right)$. The parameter $\boldsymbol{\theta}$ is defined as $\boldsymbol{\theta} = \boldsymbol{\pi} \cup \boldsymbol{\mu}$, where $\boldsymbol{\pi} := \{\pi_1, \pi_2, \cdots, \pi_K\}$, $\pi_k \in \mathbb{R}$ and $\boldsymbol{\mu} = \{\mu_1, \mu_2, \cdots, \mu_K\}, \mu_k \in \mathbb{R}^d$, $k \in [K]$. We take $N$ samples $\mathbf{X} = \{X_i\}_{i \in [N]}$ from model (1). $\{X_i\}_{i \in [N]}$ can be also rewritten as

$$X_i = \mu_{y_i} + Z_i,$$

where $\{y_i\}_{i \in [N]}$ are i.i.d. discrete random variables with $\mathbb{P}(y = k) = \pi_k$ for $k \in [K]$ and $\{Z_i\}_{i \in [N]}$ are i.i.d. standard Gaussian random vector in $\mathbb{R}^d$.

The EM algorithm(Dempster et al., 1977) remains the most widely used approach for GMM parameter estimation. Due to space constraints, we propose the algorithm in Section B. Alternatively, the

spectral algorithm(Hsu & Kakade, 2013) offers an efficient moment-based approach that estimates parameters through low-order observable moments. A key component of this method is cubic tensor decomposition(Anandkumar et al., 2014). For brevity, we defer the algorithmic details to Section B.

Next, we give a rigorous definition of the transformer model. To maintain consistency with existing literature, we adopt the notational conventions presented in Bai et al. (2023), with modifications tailored to our specific context. We consider a sequence of $N$ input vectors $\{h_i\}_{i=1}^{N} \subset \mathbb{R}^D$, which can be compactly represented as an input matrix $\mathbf{H} = [h_1, \ldots, h_N] \in \mathbb{R}^{D \times N}$, where each $h_i$ corresponds to a column of $\mathbf{H}$ (also referred to as a token).

Here we introduce several useful definitions and their full notations are given in Appendix C.

**Definition 1** (Attention layer). *A (self-)attention layer with $M$ heads is denoted as $\mathrm{Attn}_{\boldsymbol{\Theta}_{\mathtt{attn}}}(\cdot)$ with parameters $\boldsymbol{\Theta}_{\mathtt{attn}} = \{(\mathbf{V}_m, \mathbf{Q}_m, \mathbf{K}_m)\}_{m \in [M]} \subset \mathbb{R}^{D \times D}$.*

**Definition 2** (MLP layer). *A (token-wise) MLP layer with hidden dimension $D'$ is denoted as $\mathrm{MLP}_{\boldsymbol{\Theta}_{\mathtt{mlp}}}(\cdot)$ with parameters $\boldsymbol{\Theta}_{\mathtt{mlp}} = (\mathbf{W}_1, \mathbf{W}_2) \in \mathbb{R}^{D' \times D} \times \mathbb{R}^{D \times D'}$.*

**Definition 3** (Transformer). *An $L$-layer transformer, denoted as $\mathrm{TF}_{\boldsymbol{\Theta}_{\mathtt{TF}}}(\cdot)$, is a composition of $L$ self-attention layers each followed by an MLP layer:*

$$\mathrm{TF}_{\boldsymbol{\Theta}_{\mathtt{TF}}}(\mathbf{H}) = \mathrm{MLP}_{\boldsymbol{\Theta}_{\mathtt{mlp}}^{(L)}}\left(\mathrm{Attn}_{\boldsymbol{\Theta}_{\mathtt{attn}}^{(L)}}\left(\cdots \mathrm{MLP}_{\boldsymbol{\Theta}_{\mathtt{mlp}}^{(1)}}\left(\mathrm{Attn}_{\boldsymbol{\Theta}_{\mathtt{attn}}^{(1)}}(\mathbf{H})\right)\right)\right).$$

## 2.2 THE TGMM ARCHITECTURE

A recent line of work(Xie et al., 2021; Garg et al., 2022; Bai et al., 2023; Akyürek et al., 2023; Li et al., 2023b) has been studying the capability of transformer that functions as a data-driven algorithm under the context of in-context learning (ICL). However, in contrast to the setups therein where inputs consist of both features and labels, under the unsupervised GMM setup, there is no explicitly provided label information. Therefore, we formulate the learning problem as learning an *estimation* algorithm instead of learning a *prediction* algorithm as in the case of ICL. A notable property of GMM is that the structure of the estimand depends on an unknown parameter $K$, which is often treated as a hyper-parameter in GMM estimation(Titterington et al., 1985; McLachlan & Peel, 2000). For clarity of representation, we define an isotropic Gaussian mixture task as $\mathcal{T} = (\boldsymbol{\theta}, \mathbf{X}, K)$, where $\mathbf{X}$ is a i.i.d. sample generated according to ground truth $\boldsymbol{\theta}$ according to the isotropic GMM law and $K$ is the configuration used during estimation which we assume to be the same as the number of components of the ground truth $\boldsymbol{\theta}$. The GMM task is solved via applying some algorithm $\mathcal{A}$ that takes $\mathbf{X}$ and $K$ as inputs and outputs an estimate of the ground truth $\widehat{\boldsymbol{\theta}} = \mathcal{A}(\mathbf{X}; K)$.

In this paper, we propose a transformer-based architecture, transformers-for-Gaussian-mixtures (TGMM), as a GMM task solver that allows flexibility in its outputs, while at the same time being parameter-efficient, as illustrated in Figure 1: A TGMM model supports solving $s$ different GMM tasks with $K \in \mathcal{K} := \{K_1, \ldots, K_s\}$. Given inputs $N$ data points $\mathbf{X} \in \mathbb{R}^{d \times N}$ and a structure configuration of the estimand $K$. TGMM first augments the inputs with auxiliary configurations about $K$ via concatenating it with a task embedding $\mathbf{P} = \mathrm{embed}(K)$, i.e., $\mathbf{H} = [\mathbf{X} \| \mathbf{P}]$, and use a linear $\mathrm{Readin}$ layer to project the augmented inputs onto a shared hidden representation space for several estimand structures $\{K_1, \ldots, K_s\}$, which is then manipulated by a shared transformer backbone that produces task-aware hidden representations. The TGMM estimates are then decoded by task-specific $\mathrm{Readout}$ modules. More precisely, with target decoding parameters of $K$ components, the $\mathrm{Readout}$ module first performs an attentive-pooling operation(Lee et al., 2019):

$$\mathbf{O} = (\mathbf{V}_o \mathbf{H})\mathrm{SoftMax}\left((\mathbf{K}_o \mathbf{H})^\top \mathbf{Q}_o\right) \in \mathbb{R}^{(d+K) \times K},$$

where $\mathbf{V}_o, \mathbf{K}_o \in \mathbb{R}^{(d+K) \times D}$, $\mathbf{Q}_o \in \mathbb{R}^{(d+K) \times K}$. The estimates for mixture probability are then extracted by a row-wise mean-pooling of the first $K$ rows of $\mathbf{O}$, and the estimates for mean vectors are the last $d$ rows of $\mathbf{O}$. We wrap the above procedure as $\{\widehat{\pi}_k, \widehat{\mu}_k\}_{i \in [K]} = \mathrm{Readout}_{\boldsymbol{\Theta}_{\mathtt{out}}}(\mathbf{H})$. TGMM is parameter-efficient in the sense that it only introduces extra parameter complexities of the order $O(sdD)$ in addition to the backbone. We give a more detailed explanation of the parameter efficiency of TGMM in appendix Section D. We wrap the TGMM model into the following form:

$$\mathrm{TGMM}_{\boldsymbol{\Theta}}(\mathbf{X}; K) = \mathrm{Readout}_{\boldsymbol{\Theta}_{\mathtt{out}}}(\mathrm{TF}_{\boldsymbol{\Theta}_{\mathtt{TF}}}(\mathrm{Readin}_{\boldsymbol{\Theta}_{\mathtt{in}}}([\mathbf{X} \| \mathrm{embed}(K)]))).$$

Above, the parameter $\boldsymbol{\Theta} = (\boldsymbol{\Theta}_{\mathtt{TF}}, \boldsymbol{\Theta}_{\mathtt{in}}, \boldsymbol{\Theta}_{\mathtt{out}})$ consists of the parameters in the transformer $\boldsymbol{\Theta}_{\mathtt{TF}}$ and the parameters in the $\mathrm{Readin}$ and the $\mathrm{Readout}$ functions $\boldsymbol{\Theta}_{\mathtt{in}}, \boldsymbol{\Theta}_{\mathtt{out}}$.

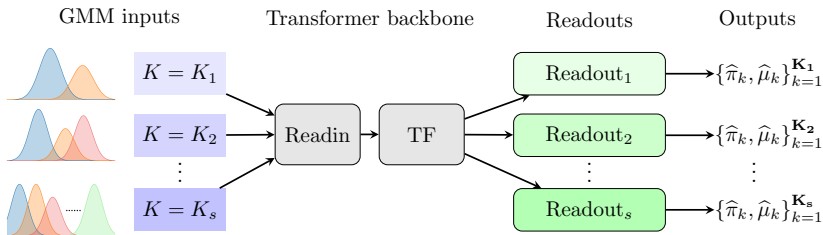

Figure 1: Illustration of the proposed TGMM architecture: TGMM utilizes a shared transformer backbone that supports solving $s$ different kind of GMM tasks via a task-specific Readout strategies.

---

**Algorithm 1** TaskSampler

**Require:** sampling distributions $p_\mu, p_\pi, p_N, p_K$.
1: Sample the type of task (i.e., number of mixture components) $K \sim p_K$.
2: Sample a GMM task according to the type of task
$$\boldsymbol{\theta} = (\boldsymbol{\mu}, \boldsymbol{\pi}),$$
$$\boldsymbol{\mu} \sim p_\mu, \boldsymbol{\pi} \sim p_\pi,$$
where $\boldsymbol{\mu} = \{\mu_1, \cdots, \mu_K\}$, $\boldsymbol{\pi} = \{\pi_1, \cdots, \pi_K\}$.
3: Sample the size of inputs $N \sim p_N$.
4: Sample the data points $\mathbf{X} = (X_1, \ldots, X_N) \overset{\text{i.i.d.}}{\sim} p(\cdot|\boldsymbol{\theta})$.
5: **return** An (isotropic) GMM task $\mathcal{T} = (\mathbf{X}, \boldsymbol{\theta}, K)$.

---

**Algorithm 2** (Meta) Training procedure for TGMM

**Require:** task dimension $d$, task types $\mathcal{K} = \{K_1, \ldots, K_s\}$, number of tasks $n$ per step, number of steps $T$.
1: Initialize a TGMM model $\text{TGMM}_{\boldsymbol{\Theta}^{(0)}}$.
2: **for** $t = 1 : T$ **do**
3:    Sample $n$ tasks $\{\mathcal{T}_i\}_{i \in [n]}$ independently using the TaskSampler from Algorithm 1.
4:    Compute the training objective $\widehat{L}_n\left(\boldsymbol{\Theta}^{(t-1)}\right)$ as in (2).
5:    Update $\boldsymbol{\Theta}^{(t-1)}$ into $\boldsymbol{\Theta}^{(t)}$ using any gradient based training algorithm like AdamW.
6: **end for**
7: **return** Trained model $\text{TGMM}_{\boldsymbol{\Theta}^{(T)}}$.

---

### 2.3 META TRAINING PROCEDURE

We adopt the meta-training framework as in Garg et al. (2022); Bai et al. (2023) and utilize diverse synthetic tasks to learn the TGMM model. In particular, during each step of the learning process, we first use a TaskSampler routine (described in Algorithm 1) to generate a batch of $n$ tasks, with each task having a probably distinct sample size. The TGMM model outputs estimates for each task, i.e., $\{\widehat{\mu}_k, \widehat{\pi}_k\}_{k \in [K]} = \text{TGMM}_{\boldsymbol{\Theta}}(\mathbf{X}; K)$. Define $\widehat{\boldsymbol{\pi}} := \{\widehat{\pi}_k\}_{k \in [K]}$ and $\widehat{\boldsymbol{\mu}} := \{\widehat{\mu}_k\}_{k \in [K]}$. For a batch of tasks $\{\mathcal{T}_i\}_{i \in [n]} = \{\mathbf{X}_i, \boldsymbol{\theta}_i, K_i\}_{i \in [n]}$, denote by $\boldsymbol{\theta}_i = \boldsymbol{\mu}_i \cup \boldsymbol{\pi}_i$ and $\widehat{\boldsymbol{\theta}}_i = \widehat{\boldsymbol{\mu}}_i \cup \widehat{\boldsymbol{\pi}}_i = \text{TGMM}_{\boldsymbol{\Theta}}(\mathbf{X}_i; K_i)$, $i \in [n]$. Then the learning objective is thus:

$$\widehat{L}_n(\boldsymbol{\Theta}) = \frac{1}{n} \sum_{i=1}^{n} \ell_\mu(\widehat{\boldsymbol{\mu}}_i, \boldsymbol{\mu}_i) + \ell_\pi(\widehat{\boldsymbol{\pi}}_i, \boldsymbol{\pi}_i). \tag{2}$$

where $\ell_\mu$ and $\ell_\pi$ are loss functions for estimation of $\mu$ and $\pi$, respectively. We will by default use square loss for $\ell_\mu$ and cross entropy loss for $\ell_\pi$. Note that the task sampling procedure relies on several sampling distributions $p_\mu, p_\pi, p_N, p_K$, which are themselves dependent upon some global configurations such as the dimension $d$ as well as the task types $\mathcal{K}$. We will omit those dependencies on global configurations when they are clear from context. The (meta) training procedure is detailed in Algorithm 2.

## 3 EXPERIMENTS

In this section, we empirically investigate TGMM's capability of learning to solve GMMs. We focus on the following research questions (RQ):

**RQ1 Effectiveness**: How well do TGMM solve GMM problems, compared to classical algorithms?
**RQ2 Robustness**: How well does TGMM perform over test tasks unseen during training?
**RQ3 Flexibility**: Can we extend the current formulation by adopting alternative backbone architectures or relaxing the isotropic setting to more sophisticated models like anisotropic GMM?

## 3.1 EXPERIMENTAL SETUP

We pick the default backbone of TGMM similar to that in Garg et al. (2022); Bai et al. (2023), with a GPT-2 type transformer encoder(Radford et al., 2019) of 12 layers, 4 heads, and 128-dimensional hidden state size. The task embedding dimension is fixed at 128. Across all the experiments, we use AdamW(Loshchilov et al., 2017) as the optimizer and use both learning rate and weight decay coefficient set to $10^{-4}$ without further tuning. During each meta-training step, we fix the batch size to be 64 and train $10^6$ steps. For the construction of TaskSampler, the sampling distributions are defined as follows: For $p_K$, We sample $K$ uniformly from $\{2, 3, 4, 5\}$; For $p_\mu$, given dimension $d$ and number of components $K$, we sample each component uniformly from $[-5, 5]^d$. Additionally, to prevent collapsed component means(Ndaoud, 2022), we filter the generated mean vectors with a maximum pairwise cosine similarity threshold of $0.8$. For $p_\pi$, given $K$, we sample each $\pi_k$ uniformly from $[0.2, 0.8]$ and normalize them to be a probability vector; For $p_N$, Given a maximum sample size $N_0$, we sample $N$ uniformly from $[N_0/2, N_0]$. The default choice of $N_0$ is 128. During evaluation, we separately evaluate 4 tasks with $2, 3, 4, 5$ components, respectively. With a sample size of 128 and averaging over 1280 randomly sampled tasks.

**Metrics.** We use $\ell_2\text{-error}$ as evaluation metrics in the experiments. We denote the output of the TGMM as $\widehat{\boldsymbol{\theta}} := \{\widehat{\pi}_1, \widehat{\mu}_1, \widehat{\pi}_2, \widehat{\mu}_2, \cdots \widehat{\pi}_K, \widehat{\mu}_K\}$. The rigorous definition is

$$\frac{1}{K} \sum_{k \in [K]} \left( \frac{1}{d} \left\| \widehat{\mu}_{\tilde{\sigma}(i)} - \mu_i \right\|^2 + \left( \widehat{\pi}_{\tilde{\sigma}(i)} - \pi_i \right)^2 \right),$$

where $\tilde{\sigma}$ is the permutation such that $\tilde{\sigma} = \arg\min_\sigma \sum_{k \in [K]} \left\| \widehat{\mu}_{\sigma(i)} - \mu_i \right\|^2$. We obtain the permutation via solving a linear assignment program using the Jonker-Volgenant algorithm(Crouse, 2016). We also report all the experimental results under two alternative metrics: cluster-classification accuracy and log-likelihood in Section H.2.

## 3.2 RESULTS AND FINDINGS

**RQ1: Effectiveness**
We compare the performance of a learned TGMM with the classical EM algorithm and spectral algorithm under 4 scenarios where the problem dimension ranges over $\{2, 8, 32, 128\}$. The results are reported in Figure 2. We observe that all three algorithms perform competitively (reaching almost zero estimation error) when $K = 2$. However, as the estimation problem gets more challenging as $K$ increases, the EM algorithm gets trapped in local minima and underperforms both spectral and TGMM. Moreover, while the spectral algorithm performs comparably with TGMM, it cannot handle cases when $K > d$, which is effectively mitigated by TGMM, with corresponding performances surpassing those of the EM algorithm. This demonstrates the effectiveness of TGMM for learning an estimation algorithm that efficiently solves GMM problems.

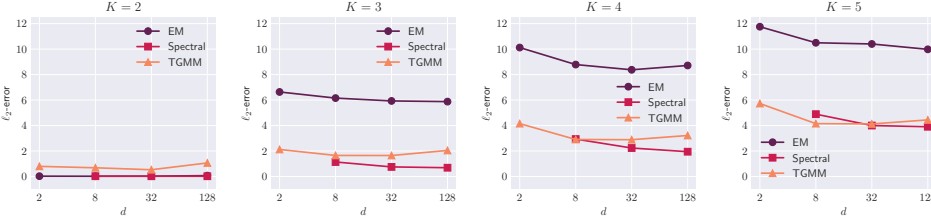

Figure 2: Performance comparison between TGMM and two classical algorithms, reported in $\ell_2$-error.

**RQ2: Robustness** To assess the robustness of the learned TGMM, we consider two types of test-time distribution shifts:
**1. Shifts in sample size $N$** Under this scenario, we evaluate the learned TGMM model on tasks with sample size $N^{\text{test}}$ that are unseen during training.
**2. Shifts in sampling distributions** Under this scenario, we test the learned TGMM model on tasks that are sampled from different sampling distributions that are used during training. Specifically, we use the same training sampling configuration as stated in Section 3.1 and test on the following

perturbed sampling scheme, with $\tilde{\mu}_k = \mu_k + \sigma_p \varepsilon_k$, where $\mu_k \overset{i.i.d.}{\sim} \text{Unif}\left([-5,5]^d\right)$, $\varepsilon_k \overset{i.i.d.}{\sim} \mathcal{N}(0, I_d)$, $k \in [K]$ and $\{\varepsilon_k\}_{k \in [K]}$ is independent with $\{\mu_k\}_{k \in [K]}$.

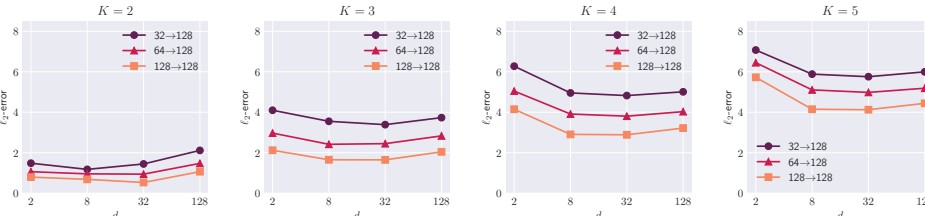

Figure 3: Assessments of TGMM under test-time task distribution shifts I: A line with $N_0^{\text{train}} \to N^{\text{test}}$ draws the performance of a TGMM model trained over tasks with sample size randomly sampled in $[N_0^{\text{train}}/2, N_0^{\text{train}}]$ and evaluated over tasks with sample size $N^{\text{test}}$. We can view the configuration $128 \to 128$ as an in-distribution test and the rest as out-of-distribution tests.

In Figure 3, we report the assessments regarding shifts in sample size, where we set $N_{\text{test}}$ to be 128 and vary the training configuration $N_0$ to range over $\{32, 64, 128\}$, respectively. The results demonstrate graceful performance degradation of out-of-domain testing performance in comparison to the in-domain performance. To measure performance over shifted test-time sampling distributions, we vary the perturbation scale $\sigma_p \in \{0, 1, \ldots, 10\}$ with problem dimension fixed at $d = 8$. The results are illustrated in Figure 4 along with comparisons to EM and spectral baselines. As shown in the results, with the increase of the perturbation scale, the estimation problem gets much harder. Nevertheless, the learned TGMM can still outperform the EM algorithm when $K > 2$. Both pieces of evidence suggest that our meta-training procedure indeed learns an algorithm instead of overfitting to some training distribution.

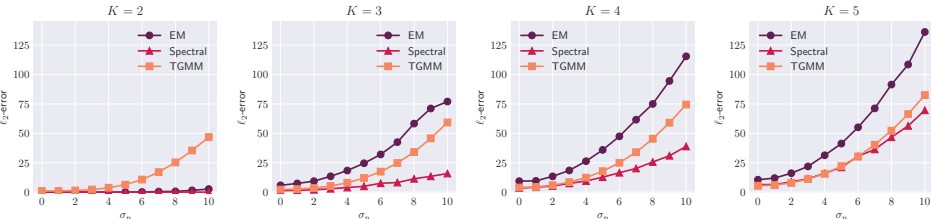

Figure 4: Assessments of TGMM under test-time task distribution shifts II: $\ell_2$-error of estimation when the test-time tasks $\mathcal{T}^{\text{test}}$ are sampled using a mean vector sampling distribution $p_\mu^{\text{test}}$ different from the one used during training.

**RQ3: Flexibility** Finally, we initiate two studies that extend both the TGMM framework and the (meta) learning problem of solving isotropic GMMs. In our first study, we investigated alternative architectures for the TGMM backbone. Motivated by previous studies(Park et al., 2024) that demonstrate the in-context learning capability of linear attention models such as Mamba series(Gu & Dao, 2023; Dao & Gu, 2024). We test replacing the backbone of TGMM with a Mamba2(Dao & Gu, 2024) model with its detailed specifications and experimental setups listed in Section H.1. The results are reported in Figure 5, suggesting that while utilizing Mamba2 as the TGMM backbone still yields non-trivial estimation efficacy, it is in general inferior to transformer backbone under comparable model complexity.

In our second study, we adapted TGMM to be compatible with more sophisticated GMM tasks via relaxing the isotropic assumption. Specifically, we construct anisotropic GMM tasks via equipping it with another scale sampling mechanism $p_\sigma$, where for each task we sample $\sigma \sim \text{softplus}(\tilde{\sigma})$ with $\tilde{\sigma}$ being sampled uniformly from $[-1, 1]^d$. We adjust the output structure of TGMM accordingly so that its outputs can be decoded into both estimates of both mean vectors, mixture probabilities, and scales, which are detailed in Section H.1. Note that the spectral algorithm does not directly apply to anisotropic setups, limiting its flexibility. Consequently, we compare TGMM with the EM approach and plot results in Figure 6 with the $\ell_2$-error metric accommodating errors from scale estimation. The

results demonstrate a similar trend as in evaluations in the isotropic case, showcasing TGMM as a versatile tool in GMM learning problems.

**Additional experiments** We postpone some further evaluations to Section H, where we present a complete report consisting of more metrics and conduct several ablations on the effects of backbone scales and sample sizes.

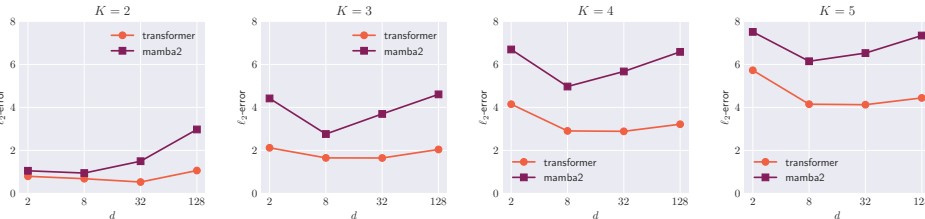

Figure 5: Performance comparisons between TGMM using transformer and Mamba2 as backbone, reported in $\ell_2$-error.

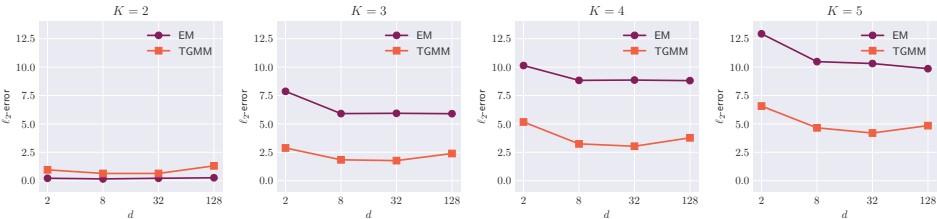

Figure 6: Performance comparison between TGMM and the EM algorithm on anisotropic GMM tasks, reported in $\ell_2$-error.

**Remark 1.** *One might be concerned with the fairness of comparisons between TGMM pre-training and EM/spectral method. We would like to point out that the only additional information that TGMM receives during meta-training is the (implicitly provided) distributional information. The empirical results show that TGMM can generalize beyond the meta-training distribution.*

## 4 THEORETICAL UNDERSTANDINGS

In this section, we provide some theoretical understandings for the experiments.

### 4.1 UNDERSTANDING TGMM

We investigate the expressive power of transformers-for-Gaussian-mixtures(TGMM) as demonstrated in Section 3. Our analysis presents two key findings that elucidate the transformer's effectiveness for GMM estimation: 1. Transformer can approximate the EM algorithm; 2. Transformer can approximate the power iteration of cubic tensor.

**Transformer can approximate the EM algorithm.** We show that transformer can efficiently approximate the EM algorithm (Algorithm B.1; see Section B) and estimate the parameters of GMM. Moreover, we show that transformer with one backbone can handle tasks with different dimensions and components simultaneously. The formal statement appears in Section F due to space limitations.

**Theorem 1** (Informal)**.** *There exists a $2L$-layer transformer $\mathrm{TF}_{\boldsymbol{\Theta}}$ such that for any $d \leq d_0$, $K \leq K_0$ and task $\mathcal{T} = (\mathbf{X}, \boldsymbol{\theta}, K)$ satisfying some regular conditions, given suitable embeddings, $\mathrm{TF}_{\boldsymbol{\Theta}}$ approximates EM algorithm $L$ steps and estimates $\boldsymbol{\theta}$ efficiently.*

**Transformer can approximate power iteration of cubic tensor.** Since directly implementing the spectral algorithm with transformers proves prohibitively complex, we instead demonstrate that transformers can effectively approximate its core computational step–the power iteration for cubic tensors (Algorithm 1 in Anandkumar et al. (2014); see Section B). Specifically, we prove that a single-layer transformer can approximate the iteration step:

$$v^{(j+1)} = T\Big(I, v^{(j)}, v^{(j)}\Big), \ j \in \mathbb{N}, \tag{3}$$

where $I$ denotes the identity matrix and $T$ represents the given cubic tensor. For technical tractability, we assume the attention layer employs a *ReLU* activation function. The formal statement appears in Section G due to space limitations.

**Theorem 2** (Informal). *There exists a $2L$-layer transformer $\text{TF}_\Theta$ with ReLU activation such that for any $d \le d_0$, $T \in \mathbb{R}^{d \times d \times d}$ and $v^{(0)} \in \mathbb{R}^d$, given suitable embeddings, $\text{TF}_\Theta$ implements $L$ steps of (3) exactly.*

We give some discussion of the theorems in the following remarks.

**Remark 2.** *(1) Theorem 1 demonstrates that a transformer architecture can approximate the EM algorithm for GMM tasks with varying numbers of components using a single shared set of parameters (i.e., one backbone $\Theta$). This finding supports the empirical effectiveness of TGMM (**RQ1** in Section 3.2). Additionally, Theorem 2 establishes that transformers can approximate power iterations for third-order tensors across different dimensions, further corroborating the model's ability to generalize across GMMs with varying component counts.*

*(2) Theorem 1 holds uniformly over sample sizes $N$ and sampling distributions under mild regularity conditions, aligning with the observed robustness of TGMM (**RQ2** in Section 3.2).*

**Remark 3.** *Different "readout" functions are also required to extract task-specific parameters in our theoretical analysis, aligning with the architectural design described in Section 2.2. For further discussion, refer to Remark F.3 in Section F.2.*

### 4.2 PROOF IDEAS

**Proof Idea of Theorem 1.** We present a brief overview of the proof strategy for Theorem 1. Our approach combines three key components: (1) the convergence properties of the population-EM algorithm(Kwon & Caramanis, 2020), (2) concentration bounds between population and sample quantities (established via classical empirical process theory), and (3) a novel transformer architecture construction. The transformer design is specifically motivated by the weighting properties of the *softmax* activation function, which naturally aligns with the EM algorithm's update structure. For intuitive understanding, Figure 7 provides a graphical illustration of this construction. The full proof is in Section F.

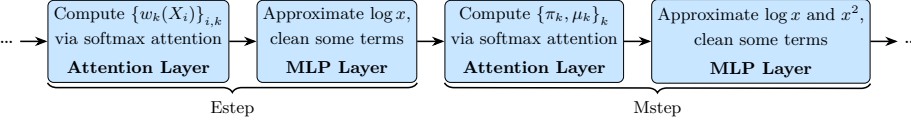

Figure 7: (Informal version)Transformer Construction for Approximating EM Algorithm Iterations. The word "clean" means setting all positions of the corresponding vector to zero.

**Proof Idea of Theorem 2.** To approximate (3), we perform a two-dimensional computation within a single-layer transformer. The key idea is to leverage the number of attention heads $M$ to handle one dimension while utilizing the $Q, K, V$ structure in the attention layer. Specifically, let $T = (T_{i,j,m})_{i,j,m \in [d]}$ and $v^{(j)} = (v_i^{(j)})_{i \in [d]}$. Then, (3) can be rewritten as $v^{(j+1)} = \sum_{j,m \in [d]} v_j v_l T_{:,j,m}$, where $T_{:,j,m} = (T_{i,j,m})_{i \in [d]} \in \mathbb{R}^d$. This operation can be implemented using $d$ attention heads, where each head processes a dimension of size $d$ (Figure 8). The complete construction and proof are provided in Section G.

$$\tilde{\mathbf{h}}_i = \mathbf{h}_i + \frac{1}{d} \sum_{m=1}^{d} \sum_{j=1}^{d} \sigma\Big(\big\langle \; \mathbf{Q}_m \mathbf{h}_i \;,\; \mathbf{K}_m \mathbf{h}_j \; \big\rangle\Big) \; \mathbf{V}_m \mathbf{h}_j \longrightarrow v^{j+1} = \sum_{m=1}^{d} \sum_{j=1}^{d} \Big( \; v_m \cdot v_j \; \Big) \; T_{:,j,m}$$

Figure 8: Illustration of implementing (3) via a multi-head attention structure, where colored boxes denote corresponding implementation components. Here $\sigma$ denotes the ReLU function.

### 5 CONCLUSION AND DISCUSSIONS

In this paper, we investigate the capabilities of transformers in GMM tasks from both theoretical and empirical perspectives. Our work is among the earliest studies to investigate the mechanism of transformers in unsupervised learning settings. Our results establish fundamental theoretical guarantees that Transformers can efficiently implement classical algorithms—such as the EM algorithm and spectral methods. This is consistent with our empirical finding that the performance of our meta-training algorithm can interpolate between EM and the spectral method. It also opens a room for future improvement of attention-based meta-training algorithms in a broader class of unsupervised learning problems. We discuss the limitations and potential future research directions in Section E.

## ACKNOWLEDGMENTS

We appreciate the constructive comments from the anonymous reviewers. Guanhua Fang is partly supported by Shanghai Educational Development Foundation (23CGA02) and National Natural Science Foundation of China (nos. 12301376).

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

# Appendix

## Table of Contents

**Disclosure of LLM usage.** We used LLMs solely for grammatical correction, language polishing, and drafting preliminary code snippets.

**Organization of the Appendix.** The appendix is organized as follows. Section A provides additional literature review. Section B formally presents the GMM algorithms referenced in Section 2, and Section C details the complete notation for the network architecture. We analyze the parameter efficiency of TGMM in Section D. Section E discusses limitations and outlines directions for future work. Rigorous statements and proofs of Theorem 1 and Theorem 2 are provided in Section F and Section G, respectively. Additional experimental details are included in Section H.

**Additional notations in the Appendix.** The maximum between two scalars $a, b$ is denoted as $a \vee b$. For a vector $v \in \mathbb{R}^d$, let $\|v\|_\infty := \max_{i \in [d]} |v_i|$ be its infinity norm. We use $\mathbf{0}_d$ to denote the zero vector and $\mathbf{e}_i \in \mathbb{R}^d$ to denote the $i$-th standard unit vector in $\mathbb{R}^d$. For a matrix $\mathbf{A} \in \mathbb{R}^{d_1 \times d_2}$, we denote $\|\mathbf{A}\|_2 := \sup_{\|x\|_2 = 1} \|\mathbf{A}x\|$ as its operator norm. We use $\widetilde{O}(\cdot)$ to denote $O(\cdot)$ with hidden log factors. For clarify, we denote the ground-truth parameters of GMM with a superscript $*$, i.e. $\{\pi_k^*, \mu_k^*\}_{k \in [K]}$, throughout this appendix.

## A    LITERATURE ON DENSITY ESTIMATION USING LLMS

Recent studies have explored the capabilities of large language models (LLMs) for in-context probability density estimation. For instance, Liu et al. (2025) interprets LLM learning as an adaptive form of Kernel Density Estimation, revealing divergent learning trajectories compared to traditional methods. Schaeffer et al. (2024) introduces a more general framework for in-context learning by modeling unconstrained energy functions, enabling effective learning even when input and output

spaces are mismatched. Meanwhile, Fakoor et al. (2020) leverages self-attention mechanisms to perform empirical density estimation across heterogeneous data types. Whereas these efforts prioritize empirical performance in distribution estimation, our paper focuses on the theoretical expressive power of transformers, specifically in the context of GMM estimation.

## B  ALGORITHM DETAILS

We state the classical algorithms of GMM mention in Section 2 in this section.

---

**Algorithm B.1** EM algorithm for GMM

---

**Require:** $\{X_i, i \in [N]\}, \boldsymbol{\theta}^{(0)} = \{\pi_1^{(0)}, \mu_1^{(0)}, \cdots \pi_K^{(0)}, \mu_K^{(0)}\}$
1: $j \leftarrow 0$
2: **while** not converge **do**
3:    **E-step:** $w_k^{(j+1)}(X_i) = \frac{\pi_k^{(j)}\phi(X_i;\mu_k^{(j)})}{\sum_{k\in[K]}\pi_k^{(j)}\phi(X_i;\mu_k^{(j)})}, i \in [N], k \in [K]$
4:    **M-step:** $\pi_k^{(j+1)} = \frac{\sum_{i\in[N]} w_k^{(j+1)}(X_i)}{N}, \mu_k^{(j+1)} = \frac{\sum_{i\in[N]} w_k^{(j+1)}(X_i)X_i}{\sum_{i\in[N]} w_k^{(j+1)}(X_i)}, k \in [K]$
5:    $j \leftarrow j + 1$
6: **end while**

---

**Algorithm B.2** Spectral Algorithm for GMM

---

**Require:** $\{X_i, i \in [N]\}$
1: Compute the empirical moments $\hat{M}_2$ and $\hat{M}_3$ by

$$\hat{M}_2 = \frac{1}{N}\sum_{i\in[N]} X_i \otimes X_i - I_d,$$

$$\hat{M}_3 = \frac{1}{N}\sum_{i\in[N]} X_i \otimes X_i \otimes X_i - \frac{1}{N}\sum_{i\in[N],j\in[d]}(X_i\otimes\mathbf{e}_j\otimes\mathbf{e}_j + \mathbf{e}_j\otimes X_i\otimes\mathbf{e}_j + \mathbf{e}_j\otimes\mathbf{e}_j\otimes X_i)$$

2: Do first $K$-th singular value decomposition(SVD) for $\hat{M}_2$: $\hat{M}_2 \approx UDU^\top$ and let $W = UD^{-1/2}$, $B = UD^{1/2}$
3: Do first $K$-th robust tensor decomposition (Algorithm 1 in Anandkumar et al. (2014), see Algorithm B.3) for $\tilde{M}_3 = \hat{M}_3(W, W, W)$:

$$\tilde{M}_3 \approx \sum_{k\in[K]} \lambda_k v_k^{\otimes 3}$$

   **return** $\hat{\pi}_k = \lambda_k^{-2}, \hat{\mu}_k = \lambda_k B v_k, k \in [K]$.

---

## C  FULL NOTATION OF NETWORK ARCHITECTURE

**Definition 4** (Attention layer). *A (self-)attention layer with $M$ heads is denoted as* $\mathrm{Attn}_{\boldsymbol{\Theta}_{\mathtt{attn}}}(\cdot)$ *with parameters* $\boldsymbol{\Theta}_{\mathtt{attn}} = \{(\mathbf{V}_m, \mathbf{Q}_m, \mathbf{K}_m)\}_{m\in[M]} \subset \mathbb{R}^{D\times D}$. *On any input sequence* $\mathbf{H} \in \mathbb{R}^{D\times N}$,

$$\widetilde{\mathbf{H}} = \mathrm{Attn}_{\boldsymbol{\Theta}_{\mathtt{attn}}}(\mathbf{H}) := \mathbf{H} + \sum_{m=1}^M(\mathbf{V}_m\mathbf{H})\,\mathbf{softmax}\left((\mathbf{K}_m\mathbf{H})^\top(\mathbf{Q}_m\mathbf{H})\right) \in \mathbb{R}^{D\times N},$$

*In vector form,*

$$\widetilde{\mathbf{h}}_i = [\mathrm{Attn}_{\boldsymbol{\Theta}_{\mathtt{attn}}}(\mathbf{H})]_i = \mathbf{h}_i + \sum_{m=1}^M\sum_{j=1}^N\left[\mathbf{softmax}\left(\left((\mathbf{Q}_m\mathbf{h}_i)^\top(\mathbf{K}_m\mathbf{h}_j)\right)_{j=1}^N\right)\right]_j \mathbf{V}_m\mathbf{h}_j.$$

*Here* $\mathbf{softmax}$ *is the activation function defined by* $\mathbf{softmax}(v) = \left(\frac{\exp(v_1)}{\sum_{i=1}^d\exp(v_i)}, \cdots, \frac{\exp(v_d)}{\sum_{i=1}^d\exp(v_i)}\right)$ *for* $v \in \mathbb{R}^d$.

---

**Algorithm B.3** Robust Tensor Power Method

---

**Require:** symmetric tensor $T \in \mathbb{R}^{d \times d \times d}$, number of iterations $L$, $N$.
**Ensure:** the estimated eigenvector/eigenvalue pair; the deflated tensor.
 1: **for** $\tau = 1$ to $L$ **do**
 2:     Draw $v_0^{(\tau)}$ uniformly at random from the unit sphere in $\mathbb{R}^d$.
 3:     **for** $t = 1$ to $N$ **do**
 4:         Compute power iteration update:
 5:         $v_t^{(\tau)} := \frac{T(I, v_{t-1}^{(\tau)}, v_{t-1}^{(\tau)})}{\|T(I, v_{t-1}^{(\tau)}, v_{t-1}^{(\tau)})\|}$
 6:     **end for**
 7: **end for**
 8: Let $\tau^* := \arg\max_{\tau \in [L]}\{T(v_N^{(\tau)}, v_N^{(\tau)}, v_N^{(\tau)})\}$.
 9: Do $N$ power iteration updates (line 5) starting from $v_N^{(\tau^*)}$ to obtain $\hat{v}$.
10: Set $\hat{\lambda} := \tilde{T}(\hat{v}, \hat{v}, \hat{v})$.
11: **return** the estimated eigenvector/eigenvalue pair $(\hat{v}, \hat{\lambda})$; the deflated tensor $\tilde{T} - \hat{\lambda}\hat{v}^{\otimes 3}$.

---

The Multilayer Perceptron(MLP) layer is defined as follows.

**Definition 5** (MLP layer). *A (token-wise) MLP layer with hidden dimension $D'$ is denoted as* $\mathrm{MLP}_{\Theta_{\mathtt{mlp}}}(\cdot)$ *with parameters* $\Theta_{\mathtt{mlp}} = (\mathbf{W}_1, \mathbf{W}_2) \in \mathbb{R}^{D' \times D} \times \mathbb{R}^{D \times D'}$. *On any input sequence* $\mathbf{H} \in \mathbb{R}^{D \times N}$,

$$\widetilde{\mathbf{H}} = \mathrm{MLP}_{\Theta_{\mathtt{mlp}}}(\mathbf{H}) := \mathbf{H} + \mathbf{W}_2\sigma(\mathbf{W}_1\mathbf{H}),$$

*where* $\sigma : \mathbb{R} \to \mathbb{R}$ *is the ReLU function. In vector form, we have* $\widetilde{\mathbf{h}}_i = \mathbf{h}_i + \mathbf{W}_2\sigma(\mathbf{W}_1\mathbf{h}_i)$.

Then we can use the above definitions to define the transformer model.

**Definition 6** (Transformer). *An $L$-layer transformer, denoted as* $\mathrm{TF}_{\Theta_{\mathrm{TF}}}(\cdot)$, *is a composition of $L$ self-attention layers each followed by an MLP layer:*

$$\mathrm{TF}_{\Theta_{\mathrm{TF}}}(\mathbf{H}) = \mathrm{MLP}_{\Theta_{\mathtt{mlp}}^{(L)}}\left(\mathrm{Attn}_{\Theta_{\mathtt{attn}}^{(L)}}\left(\cdots \mathrm{MLP}_{\Theta_{\mathtt{mlp}}^{(1)}}\left(\mathrm{Attn}_{\Theta_{\mathtt{attn}}^{(1)}}(\mathbf{H})\right)\right)\right).$$

*Here the parameter* $\Theta_{\mathrm{TF}} = (\Theta_{\mathtt{attn}}^{(1:L)}, \Theta_{\mathtt{mlp}}^{(1:L)})$ *consists of the attention layers* $\Theta_{\mathtt{attn}}^{(\ell)} = \{(\mathbf{V}_m^{(\ell)}, \mathbf{Q}_m^{(\ell)}, \mathbf{K}_m^{(\ell)})\}_{m \in [M^{(\ell)}]} \subset \mathbb{R}^{D \times D}$, *the MLP layers* $\Theta_{\mathtt{mlp}}^{(\ell)} = (\mathbf{W}_1^{(\ell)}, \mathbf{W}_2^{(\ell)}) \in \mathbb{R}^{D^{(\ell)} \times D} \times \mathbb{R}^{D \times D^{(\ell)}}$.

## D  ON THE PARAMETER EFFICIENCY OF TGMM

Aside from its backbone, the extra parameters in a TGMM comprises the following:

**Parameters in the task embedding module**  This part has a parameter count of $s \times d_{\text{task}}$.

**Parameters in the** Readin **layer**  This part has a parameter count of $O((d_{\text{task}} + d) \times D)$.

**Parameters in the** Readout **layer**  This part has a parameter count of $O(sdD)$, which comprises of parameters from $s$ distinct attention mechanisms.

As $d_{\text{task}}$ is typically of the order $O(D)$, we conclude that the total extra parameter complexity is of the order $O(sdD)$, which in practice is often way smaller than the parameter complexity of the backbone, i.e., of the order $O(LD^2)$ Meanwhile, a naive implementation of adapting transformer architecture to solve $s$ distinct GMM tasks require a different transformer backbone. As the complexity of backbone often dominate those of extra components, the TGMM implementation can reduce the parameter complexity by an (approximate) factor of $1/s$ in practice.

## E  LIMITATIONS AND FUTURE WORK DIRECTIONS

First, while our theoretical analysis focuses on the approximation ability of transformers, the optimization dynamics remain unexplored. This is a common theoretical challenge in ICL literature;

see Bai et al. (2023); Lin et al. (2024); Giannou et al. (2025). Second, approximating the full spectral algorithm (Algorithm B.2; see Section B) presents a significant challenge, which we leave for future work. Third, our study is limited to the expressivity of transformers on classical GMM tasks; exploring their performance on other unsupervised learning tasks is an interesting direction that warrants further investigation.

## F   FORMAL STATEMENT OF THEOREM 1 AND PROOFS

For analytical tractability, we implement $\mathrm{Readin}$ as an identity transformation and define $\mathrm{Readout}$ to extract targeted matrix elements hence they are both fixed functions. Actually, we also need "Readout" functions to get the estimated parameters for different tasks, see Remark F.3. To theoretical convenience, we use the following norm of transformers, which differs slightly from the definition in Bai et al. (2023).

$$\|\mathbf{\Theta}\| := \max_{\ell\in[L]}\left\{\max_{m\in[M]}\left\{\left\|\mathbf{Q}_m^{(\ell)}\right\|_2,\left\|\mathbf{K}_m^{(\ell)}\right\|_2,\left\|\mathbf{V}_m^{(\ell)}\right\|_2\right\}+\left\|\mathbf{W}_1^{(\ell)}\right\|_2+\left\|\mathbf{W}_2^{(\ell)}\right\|_2\right\}.$$

Then the transformer class can be defined as

$$\mathcal{F} := \mathcal{F}(L,D,D',M,B_{\mathbf{\Theta}}) = \left\{\mathrm{TF}_{\mathbf{\Theta}}, \|\mathbf{\Theta}\| \le B_{\mathbf{\Theta}}, D^{(\ell)} \le D', M^{(\ell)} \le M, \ell\in[L]\right\}.$$

### F.1   FORMAL STATEMENT OF THEOREM 1

First, we introduce some notations. We define $\pi_{\min} = \min_i \pi_i^*$, $\rho_\pi = \max_i \pi_i^* / \min_i \pi_i^*$. We use $R_{ij} = \|\mu_i^* - \mu_j^*\|$ to denote the pairwise distance between components and $R_{\min} = \min_{i\neq j} R_{ij}$, $R_{max} = (\max_{i\neq j} R_{ij}) \vee (\max_{i\in[K]} \|\mu_i^*\|)$. Without the loss of generality, we assume that $R_{\max} \ge 1$. For dimension and components adaptation, we assume $d \le d_0$ and $K \le K_0$. Since in practice the sample size $N$ is much larger than the number of components $K$, we assume that $N$ is divisible by $K$, i.e. $N/K \in \mathbb{N}$. Otherwise, we only consider the first $K\lfloor N/K \rfloor$ samples and drop the others. We encode $\mathbf{X} = \{X_i\}_{i=1}^N$ into an input sequence $\mathbf{H}$ as the following:

$$\mathbf{H} = \begin{bmatrix} \overline{X}_1 & \overline{X}_2 & \dots & \overline{X}_N \\ \mathbf{p}_1 & \mathbf{p}_2 & \dots & \mathbf{p}_N \end{bmatrix} \in \mathbb{R}^{D\times N}, \quad \mathbf{p}_i = \begin{bmatrix} \overline{\boldsymbol{\theta}}_i \\ \mathbf{r}_i \end{bmatrix}, \tag{4}$$

$$\overline{\boldsymbol{\theta}}_i = \begin{bmatrix} \overline{\boldsymbol{\pi}}_{\log} \\ \overline{\mu}_{i\%K} \\ c_{i\%K} \\ \mathbf{0}_{3K_0} \end{bmatrix} \in \mathbb{R}^{d_0+4K_0+1}, \quad \mathbf{r}_i = \begin{bmatrix} \mathbf{0}_{\tilde{D}} \\ 1 \\ \mathbf{e}_{i\%K} \end{bmatrix} \in \mathbb{R}^{D-(2d_0+3K_0+1)}, \tag{5}$$

where $\overline{X}_i = [X_i^\top, \mathbf{0}_{d_0-d}^\top]^\top$, $\overline{\boldsymbol{\pi}}_{\log} = [\boldsymbol{\pi}_{\log}^\top, \mathbf{0}_{K_0-K}^\top]^\top$, $\overline{\mu}_{i\%K} = [\mu_{i\%K}^\top, \mathbf{0}_{d_0-d}^\top]^\top$, $c_{i\%K} \in \mathbb{R}$ and $\mathbf{e}_{i\%K} \in \mathbb{R}^{K_0}$ denotes the $i\%K$-th standard unit vector. To match the dimension, $\tilde{D} = D - (2d_0 + 5K_0 + 2)$. We choose $D = O(d_0 + K_0)$ to get the encoding above. For the initialization, we choose $\boldsymbol{\pi}_{\log} = \log \boldsymbol{\pi}^{(0)}$, $\mu_i = \mu_i^{(0)}$, $c_i = \|\mu_i^{(0)}\|_2^2$.

To guarantee convergence of the EM algorithm, we adopt the following assumption for the initialization parameters, consistent with the approach in Kwon & Caramanis (2020).

(A1) Suppose the GMM has parameters $\{(\pi_j^*, \mu_j^*) : j \in [K]\}$ such that

$$R_{\min} \ge C \cdot \sqrt{\log(\rho_\pi K)},$$

and suppose the mean initialization $\mu_1^{(0)}, ..., \mu_K^{(0)}$ satisfies

$$\forall i \in [K], \left\|\mu_i^{(0)} - \mu_i^*\right\| \le \frac{R_{\min}}{16}.$$

Also, suppose the mixing weights are initialized such that

$$\forall i \in [K], \left|\pi_i^{(0)} - \pi_i^*\right| \le \pi_i/2.$$

We denote the output of the transformer $\mathrm{TF}_{\Theta}$ as $\boldsymbol{\theta}^{\mathrm{TF}} := \{\pi_1^{\mathrm{TF}}, \mu_1^{\mathrm{TF}}, \pi_2^{\mathrm{TF}}, \mu_2^{\mathrm{TF}}, \cdots \pi_K^{\mathrm{TF}}, \mu_K^{\mathrm{TF}}\}$ and assume matched indices. Define

$$D_{\Theta}^{\mathrm{TF}} := \max_{i \in [K]} \left\{ \left\| \mu_i^{\mathrm{TF}} - \mu_i^* \right\| \vee \left( \left| \pi_i^{\mathrm{TF}} - \pi_i^* \right| / \pi_i \right) \right\}.$$

Now we propose the theorem that transformer can efficient approximate the EM Algorithm (Algorithm B.1), which is the formal version of Theorem 1.

**Theorem F.1.** *Fix $0 < \delta, \beta < 1$ and $1/2 < a < 1$. Suppose there exists a sufficiently large universal constant $C \geq 128$ for which assumption (A1) holds. If $N$ is suffcient large and $\varepsilon \leq 1/(100K_0)$ sufficient small such that*

$$\frac{\tilde{c}_1}{(1-a)\pi_{\min}} \sqrt{\frac{R_{\max}(R_{\max} \vee d) \log\left(\frac{24K}{\delta}\right)}{N}}$$

$$+ \tilde{c}_2 \left( R_{\max} + d \left( 1 + \sqrt{\frac{2\log(\frac{4N}{\delta})}{d}} \right) \right) N\varepsilon < \frac{1}{2} \left( a - \frac{1}{2} \right),$$

*and*

$$\epsilon(N, \varepsilon, \delta, a) := \frac{\tilde{c}_3}{(1-a)\,\pi_{\min}} \sqrt{\frac{Kd\log(\frac{\tilde{C}N}{\delta})}{N}}$$

$$+ \tilde{c}_4 \left( \frac{1}{\pi_{\min}} + N \left( R_{\max} + d + \sqrt{2d\log\left(\frac{4N}{\delta}\right)} \right) \right) \varepsilon$$

$$< a(1-\beta),$$

*hold, where $\tilde{c}_1$-$\tilde{c}_4$ are universal constants, $\tilde{C} = 288K^2(\sqrt{d} + 2R_{\max} + \frac{1}{1-a})^2$. Then there exists a $2(L+1)$-layer transformer $\mathrm{TF}_{\Theta}$ such that*

$$D_{\Theta}^{\mathrm{TF}} \leq a\beta^L + \frac{1}{1-\beta}\epsilon(N, \varepsilon, \delta, a) \tag{6}$$

*holds with probability at least $1 - \delta$. Moreover, $\mathrm{TF}_{\Theta}$ falls within the class $\mathcal{F}$ with parameters satisfying:*

$$D = O(d_0 + K_0), D' \leq \tilde{O}\big(K_0 R_{\max}(R_{\max} + d_0)\varepsilon^{-1}\big),$$
$$M = O(1),$$
$$\log B_{\Theta} \leq \tilde{O}(K_0 R_{\max}(R_{\max} + d_0)).$$

*Notably, (6) holds for all tasks satisfying $d \leq d_0$ and $K \leq K_0$, where the parameters of transformer $\Theta$ remains fixed across different tasks $\mathcal{T}$.*

**Remark F.1.** *From Theorem F.1, if we take $\varepsilon = \tilde{O}\big(N^{-3/2}d^{-1/2}\big)$ and $L = O(\log N)$, then we have*

$$D_{\Theta}^{\mathrm{TF}} \leq \tilde{O}\left(\sqrt{\frac{d}{N}}\right),$$

*which matches the canonical parametric error rate.*

**Remark F.2.** *We give some explanations for the notations in Theorem F.1. Define*

$$D_j^{\mathrm{pEM}} := \max_{i \in [K]} \left\{ \left\| \tilde{\mu}_i^{(j)} - \mu_i \right\| \vee \left( \left| \tilde{\pi}_i^{(j)} - \pi_i \right| / \pi_i \right) \right\},$$

*where $\{\tilde{\mu}_i^{(j)}, \tilde{\pi}_i^{(j)}\}_{i \in [K]}$ are the parameters obtained at the $j$-th iteration of the population-EM algorithm (see Section F.3 for details). In the convergence analysis of the population-EM algorithm (Kwon & Caramanis, 2020), it is shown that after the first iteration, the parameters lie in a small neighborhood of the true parameters with high probability (i.e., $D_1^{\mathrm{pEM}} \leq a$ for some $1/2 \leq a < 1$). Furthermore, the authors prove that the algorithm achieves linear convergence (i.e., $D_{j+1}^{\mathrm{pEM}} \leq \beta D_j^{\mathrm{pEM}}$ for $j \in \mathbb{N}_+$ and some $0 < \beta < 1$) with high probability if $D_1^{\mathrm{pEM}} \leq a$ holds. Following their notations, here $a$ represents the radius of the neighborhood after the first iteration, while $\beta$ is the linear convergence rate parameter. Finally, $\varepsilon$ controls the approximation error of the transformer.*

## F.2 CONSTRUCTION OF TRANSFORMER ARCHITECTURE AND FORMAL VERSION OF FIGURE 7

In this section, we give the transformer architecture construction in Theorem F.1. We denote $w_{ij} = w_j(X_i), i \in [N], k \in [K]$ in this subsection for simplicity. Recall that we have assumed that $d \le d_0, K \le K_0$ and $N$ is divisible by $K(N/K \in \mathbb{N})$. We first restate the encoding formulas in (4):

$$\mathbf{H} = \begin{bmatrix} \overline{X}_1 & \overline{X}_2 & \dots & \overline{X}_N \\ \overline{\boldsymbol{\theta}}_1 & \overline{\boldsymbol{\theta}}_2 & \dots & \overline{\boldsymbol{\theta}}_N \\ \mathbf{p}_1 & \mathbf{p}_2 & \dots & \mathbf{p}_N \end{bmatrix} \in \mathbb{R}^{D \times N},$$

$$\overline{\boldsymbol{\theta}}_i = \begin{bmatrix} \overline{\boldsymbol{\pi}}_{\log} \\ \overline{\mu}_{i\%K} \\ c_{i\%K} \\ \overline{\mathbf{w}}_i \\ \overline{\mathbf{w}}_{i\log} \\ \overline{\boldsymbol{\pi}} \end{bmatrix} \in \mathbb{R}^{d_0+4K_0+1}, \quad \mathbf{p}_i := \begin{bmatrix} \mathbf{0}_{D-(2d_0+5K_0+2)} \\ 1 \\ \mathbf{e}_{i\%K} \end{bmatrix} \in \mathbb{R}^{D-(2d_0+3K_0+1)},$$

where $\overline{X}_i = [X_i^\top, \mathbf{0}_{d_0-d}^\top]^\top$, $\overline{\boldsymbol{\pi}}_{\log} = [\boldsymbol{\pi}_{\log}^\top, \mathbf{0}_{K_0-K}^\top]^\top$, $\overline{\mu}_{i\%K} = [\mu_{i\%K}^\top, \mathbf{0}_{d_0-d}^\top]^\top$, $\overline{\mathbf{w}}_i = [\mathbf{w}_i^\top, \mathbf{0}_{K_0-K}^\top]^\top$, $\overline{\mathbf{w}}_{i\log} = [\mathbf{w}_{i\log}^\top, \mathbf{0}_{K_0-K}^\top]^\top$, $\overline{\boldsymbol{\pi}} = [\boldsymbol{\pi}^\top, \mathbf{0}_{K_0-K}^\top]^\top$, $c_{i\%K} \in \mathbb{R}$ and $\mathbf{e}_{i\%K} \in \mathbb{R}^{K_0}$ denotes the $i\%K$-th standard unit vector. For the initialization, we choose $\boldsymbol{\pi}_{\log} = \log \boldsymbol{\pi}^{(0)}$, $\mu_i = \mu_i^{(0)}$, $c_i = \|\mu_i^{(0)}\|_2^2$. and $\boldsymbol{\pi} = \mathbf{w}_i = \mathbf{w}_{i\log} = \mathbf{0}_K, i \in [K]$. Finally, take $\mathbf{H}^{(0)} = \mathbf{H}$ which is defined in **??**.

Then in E-step, we consider the following attention structures: we define matrices $\mathbf{Q}^{(1)}, \mathbf{K}^{(1)}, \mathbf{V}^{(1)}$, such that

$$\mathbf{Q}^{(1)} \mathbf{h}_i^{(0)} = \begin{bmatrix} \overline{X}_i \\ \overline{\boldsymbol{\pi}}_{\log} \\ 1 \\ \mathbf{0} \end{bmatrix}, \quad \mathbf{K}^{(1)} \mathbf{h}_j^{(0)} = \begin{bmatrix} -\overline{\mu}_{j\%K} \\ \mathbf{e}_{j\%K} \\ \frac{1}{2} c_{j\%K} \\ \mathbf{0} \end{bmatrix}, \quad \mathbf{V}^{(1)} \mathbf{h}_j^{(0)} = \begin{bmatrix} \mathbf{0}_{d_0} \\ \mathbf{0}_{K_0} \\ \mathbf{0}_{d_0+1} \\ \mathbf{e}_{j\%K} \\ \mathbf{0}_{D-(2d_0+2K_0+1)} \end{bmatrix},$$

and use the standard softmax attention, thus

$$\begin{aligned} \widetilde{\mathbf{h}}_i^{(1)} &= \left[ \text{Attn}_{\boldsymbol{\Theta}_{\text{attn}}^{(1)}} (\mathbf{H}^{(0)}) \right]_{:,i} \\ &= \mathbf{h}_i^{(0)} + \sum_{j=1}^N \left[ \text{softmax} \left( \left( \left( \mathbf{Q}^{(1)} \mathbf{h}_i^{(0)} \right)^\top \left( \mathbf{K}^{(1)} \mathbf{h}_j^{(0)} \right) \right)_{j=1}^N \right) \right]_j \cdot \mathbf{V}^{(1)} \mathbf{h}_j^{(0)} \\ &= \mathbf{h}_i^{(0)} + \sum_{j=1}^N \frac{\alpha_{j\%K}^{(0)} \exp\left( -X_i^\top \mu_{j\%K} + \frac{1}{2} \mu_{j\%K}^\top \mu_{j\%K} \right)}{B \sum_{k=1}^K \alpha_k^{(0)} \exp\left( -X_i^\top \mu_k + \frac{1}{2} \mu_k^\top \mu_k \right)} \cdot \mathbf{V}^{(1)} \mathbf{h}_j^{(0)} \\ &= \mathbf{h}_i^{(0)} + \frac{1}{B} \sum_{j=1}^N \hat{w}_{ij\%K}^{(1)} \mathbf{V}^{(1)} \mathbf{h}_j^{(0)} \\ &= \mathbf{h}_i^{(0)} + \sum_{j=1}^K \hat{w}_{ij}^{(1)} \mathbf{V}^{(1)} \mathbf{h}_j^{(0)} \\ &= \mathbf{h}_i^{(0)} + \begin{bmatrix} \mathbf{0}_{d_0} \\ \mathbf{0}_{K_0} \\ \mathbf{0}_{d_0+1} \\ \widehat{\overline{\mathbf{w}}}_i^{(1)} \\ \mathbf{0}_{D-(2d_0+2K_0+1)} \end{bmatrix}, \; i \in [N]. \end{aligned}$$

where $\widehat{\overline{\mathbf{w}}}_i^{(1)} = \left( \hat{w}_{i1}^{(1)}, \hat{w}_{i2}^{(1)}, \cdots, \hat{w}_{iK}^{(1)}, 0, \cdots, 0 \right)^\top \in \mathbb{R}^{K_0}$.

Then we use a two-layer MLP to approximate $\log x$ and clean all $\overline{\boldsymbol{\pi}}_{\log}$ , $\overline{\mu}_{i\%K}$ and $c_{i\%K}$, which is

$$\mathbf{h}_i^{(1)} = \mathrm{MLP}_{\boldsymbol{\Theta}_{\mathtt{mlp}}^{(1)}}\left(\widetilde{\mathbf{h}}_i^{(1)}\right) = \begin{bmatrix} \overline{X}_i \\ \mathbf{0}_{K_0} \\ \mathbf{0}_{d_0+1} \\ \overline{\mathbf{w}}_i^{(1)} \\ \widehat{\mathbf{w}}_{i\,\log}^{(1)} \\ \mathbf{0}_{K_0} \\ \mathbf{0}_{D-(2d_0+5K_0+2)} \\ 1 \\ \mathbf{e}_{i\%K} \end{bmatrix}, i \in [N],$$

where $\overline{\widehat{\mathbf{w}}}_{i\,\log}^{(1)} = \widehat{\overline{\log\widehat{\mathbf{w}}}}_i^{(1)}$. Notice that although $\log x$ is not defined at $0$, the MLP approximation is well defined with some value which we do not care because we will not use it in the M-step. Similarly, for any $\ell\%2 = 1$, $\ell \in \mathbb{N}_+$ we have

$$\mathbf{h}_i^{(\ell)} = \mathrm{MLP}_{\boldsymbol{\Theta}_{\mathtt{mlp}}^{(\ell\%2)}}\left(\left[\mathrm{Attn}_{\boldsymbol{\Theta}_{\mathtt{attn}}^{(\ell\%2)}}(\mathbf{H}^{(\ell-1)})\right]_{:,i}\right) = \begin{bmatrix} \overline{X}_i \\ \mathbf{0}_{K_0} \\ \mathbf{0}_{d_0+1} \\ \overline{\mathbf{w}}_i^{((\ell+1)/2)} \\ \widehat{\mathbf{w}}_{i\,\log}^{((\ell+1)/2)} \\ \mathbf{0}_{K_0} \\ \mathbf{0}_{D-(2d_0+5K_0+2)} \\ 1 \\ \mathbf{e}_{i\%K} \end{bmatrix}, i \in [N],$$

where $\overline{\widehat{\mathbf{w}}}_{i\,\log}^{((\ell+1)/2)} = \widehat{\overline{\log\widehat{\mathbf{w}}}}_i^{((\ell+1)/2)}$.

In M-step, we consider the following attention structures: we similarly define matrices $\mathbf{Q}_m^{(2)}$, $\mathbf{K}_m^{(2)}$, $\mathbf{V}_m^{(2)}$, $m = 1, 2$ such that

$$\mathbf{Q}_1^{(2)}\mathbf{h}_j^{(1)} = \begin{bmatrix} \mathbf{e}_{j\%K} \\ \mathbf{0} \end{bmatrix}, \quad \mathbf{K}_1^{(2)}\mathbf{h}_i^{(1)} = \begin{bmatrix} \overline{\widehat{\mathbf{w}}}_{i\,\log}^{(1)} \\ \mathbf{0} \end{bmatrix}, \quad \mathbf{V}_1^{(2)}\mathbf{h}_i^{(1)} = \begin{bmatrix} \mathbf{0}_{d_0} \\ \mathbf{0}_{K_0} \\ \overline{X}_i \\ 0 \\ \mathbf{0}_{K_0} \\ \mathbf{0}_{K_0} \\ \mathbf{0}_{D-(2d_0+3K_0+1)} \end{bmatrix},$$

and

$$\mathbf{Q}_2^{(2)}\mathbf{h}_j^{(1)} = \mathbf{0}, \quad \mathbf{K}_2^{(2)}\mathbf{h}_i^{(1)} = \mathbf{0}, \quad \mathbf{V}_2^{(2)}\mathbf{h}_i^{(1)} = \begin{bmatrix} \mathbf{0}_{d_0} \\ \mathbf{0}_{K_0} \\ \mathbf{0}_{d_0+1} \\ \mathbf{0}_{K_0} \\ \mathbf{0}_{K_0} \\ \overline{\widehat{\mathbf{w}}}_i^{(1)} \\ \mathbf{0}_{D-(2d_0+4K_0+1)} \end{bmatrix},$$

Then we get

$$\begin{aligned} \widetilde{\mathbf{h}}_j^{(2)} &= \left[\mathrm{Attn}_{\boldsymbol{\Theta}_{\mathtt{attn}}^{(2)}}(\mathbf{H}^{(1)})\right]_{:,j} \\ &= \mathbf{h}_j^{(1)} + \sum_{m=1}^{2}\sum_{i=1}^{N}\left[\mathbf{softmax}\left(\left(\left(\mathbf{Q}_m^{(2)}\mathbf{h}_i^{(1)}\right)^{\top}\left(\mathbf{K}_m^{(2)}\mathbf{h}_j^{(1)}\right)\right)_{j=1}^{N}\right)\right]_j \cdot \mathbf{V}_m^{(2)}\mathbf{h}_i^{(1)} \\ &= \mathbf{h}_j^{(1)} + \sum_{i=1}^{N}\frac{\hat{w}_{ij\%K}^{(1)}}{\sum_{i=1}^{N}\hat{w}_{ij\%K}^{(1)}} \cdot \mathbf{V}_1^{(2)}\mathbf{h}_i^{(1)} + \sum_{i=1}^{N}\frac{1}{N} \cdot \mathbf{V}_2^{(2)}\mathbf{h}_i^{(1)} \end{aligned}$$

$$
= \mathbf{h}_j^{(1)} + \begin{bmatrix} \mathbf{0}_{d_0} \\ \mathbf{0}_{K_0} \\ \overline{\hat{\mu}}_{j\%K}^{(1)} \\ 0 \\ \mathbf{0}_{K_0} \\ \mathbf{0}_{K_0} \\ \mathbf{0}_{D-(2d_0+3K_0+1)} \end{bmatrix} + \begin{bmatrix} \mathbf{0}_{d_0} \\ \mathbf{0}_{K_0} \\ \mathbf{0}_{d_0+1} \\ \mathbf{0}_{K_0} \\ \mathbf{0}_{K_0} \\ \overline{\hat{\pi}}^{(1)} \\ \mathbf{0}_{D-(2d_0+4K_0+1)} \end{bmatrix},
$$

$$
= \mathbf{h}_j^{(1)} + \begin{bmatrix} \mathbf{0}_{d_0} \\ \mathbf{0}_{K_0} \\ \overline{\hat{\mu}}_{j\%K}^{(1)} \\ 0 \\ \mathbf{0}_{K_0} \\ \mathbf{0}_{K_0} \\ \overline{\hat{\pi}}^{(1)} \\ \mathbf{0}_{D-(2d_0+4K_0+1)} \end{bmatrix}, \; j \in [N].
$$

Similarly, we use a two-layer MLP to approximate $\log x$, $x^2$ and clean all $\overline{\mathbf{w}}_i$, $\overline{\mathbf{w}}_{i\log}$ and $\overline{\pi}_i$, which is

$$
\mathbf{h}_j^{(2)} = \mathrm{MLP}_{\mathbf{\Theta}_{\mathrm{mlp}}^{(2)}}\left(\widetilde{\mathbf{h}}_i^{(2)}\right) = \begin{bmatrix} \overline{X}_j \\ \overline{\hat{\pi}}_{\log}^{(1)} \\ \overline{\hat{\mu}}_{j\%K}^{(1)} \\ \hat{c}_{j\%K}^{(1)} \\ \mathbf{0}_{K_0} \\ \mathbf{0}_{K_0} \\ \mathbf{0}_{K_0} \\ \mathbf{0}_{D-(2d_0+5K_0+2)} \\ 1 \\ \mathbf{e}_{j\%K} \end{bmatrix}, \; j \in [N],
$$

where $\overline{\hat{\pi}}_{\log}^{(1)} = \widehat{\log\overline{\hat{\pi}}}^{(1)}$, $\hat{c}_{j\%K}^{(1)} = \|\widehat{\overline{\hat{\mu}}_{j\%K}^{(1)}}\|_2^2$.

Similarly, for any $\ell\%2 = 0$, $\ell \in \mathbb{N}_+$ we have

$$
\mathbf{h}_j^{(\ell)} = \mathrm{MLP}_{\mathbf{\Theta}_{\mathrm{mlp}}^{(\ell\%2)}}\left(\left[\mathrm{Attn}_{\mathbf{\Theta}_{\mathrm{attn}}^{(\ell\%2)}}(\mathbf{H}^{(\ell-1)})\right]_{:,j}\right) = \begin{bmatrix} \overline{X}_j \\ \overline{\hat{\pi}}_{\log}^{(\ell/2)} \\ \overline{\hat{\mu}}_{j\%K}^{(\ell/2)} \\ \hat{c}_{j\%K}^{(\ell/2)} \\ \mathbf{0}_{K_0} \\ \mathbf{0}_{K_0} \\ \mathbf{0}_{K_0} \\ \mathbf{0}_{D-(2d_0+5K_0+2)} \\ 1 \\ \mathbf{e}_{j\%K} \end{bmatrix}, \; j \in [N].
$$

where $\overline{\hat{\pi}}_{\log}^{(\ell/2)} = \widehat{\log\overline{\hat{\pi}}}^{(\ell/2)}$, $\hat{c}_{j\%K}^{(\ell/2)} = \|\widehat{\overline{\hat{\mu}}_{j\%K}^{(\ell/2)}}\|_2^2$.

Thus, we can get $\hat{\pi}^{(\ell)}$ and $\hat{\mu}_j^{(\ell)}$, $j \in [K]$ after $2\ell$ layers of transformer constructed above. (The last-layer MLP block retains $\pi$ as an output parameter without cleaning it.) Our transformer construction is summarized in Figure 9, which is the formal version of Figure 7 in Section 4.2.

**Remark F.3.** *The output of transformer $\mathbf{H}^{(2L)}$ is a large matrix containing lots of elements. To get the estimated parameters, we need to extract specific elements. In details, $\mathbf{H}^{(2L)} =$*

$\left[\mathbf{h}_1^{(2L)}, \cdots, \mathbf{h}_N^{(2L)}\right] \in \mathbb{R}^{D \times N}$, where

$$\mathbf{h}_i^{(2L)} = \begin{bmatrix} \overline{X}_i \\ \overline{\hat{\boldsymbol{\pi}}}_{\log}^{(L)} \\ \overline{\hat{\mu}}_{i\%K}^{(L)} \\ \hat{c}_{i\%K}^{(L)} \\ \mathbf{0}_{K_0} \\ \mathbf{0}_{K_0} \\ \overline{\hat{\boldsymbol{\pi}}}^{(L)} \\ \mathbf{0}_{D-(2d_0+5K_0+2)} \\ 1 \\ \mathbf{e}_{j\%K} \end{bmatrix}, i \in [N].$$

We use the following linear attentive pooling to get the parameters:

$$\mathbf{O} = \frac{1}{N}(\mathbf{V}_o\mathbf{H})\big((\mathbf{K}_o\mathbf{H})^\top\mathbf{Q}_o\big) \in \mathbb{R}^{(d+K) \times K},$$

where $\mathbf{Q}_o = [\mathbf{q}_{o1}, \cdots, \mathbf{q}_{oK}] \in \mathbb{R}^{(d+K) \times K}$, $\mathbf{K}_o, \mathbf{V}_o \in \mathbb{R}^{(d+K) \times N}$ satisfying

$$\mathbf{q}_{oi} = \begin{bmatrix} K\mathbf{e}_i \\ \mathbf{0}_d \end{bmatrix}, \quad \mathbf{K}_o\mathbf{h}_j^{(2L)} = \begin{bmatrix} \mathbf{e}_{j\%K} \\ \mathbf{0}_d \end{bmatrix}, \quad \mathbf{V}_o\mathbf{h}_j^{(2L)} = \begin{bmatrix} \hat{\boldsymbol{\pi}}^{(L)} \\ \hat{\mu}_{j\%K} \end{bmatrix}.$$

Thus by $N/K \in \mathbb{N}$, we have

$$\mathbf{o}_i = \frac{1}{N}\sum_{j \in [N]} \mathbf{q}_{oi}^\top(\mathbf{K}_o\mathbf{h}_j)\mathbf{V}_o\mathbf{h}_j = \frac{K}{N}\frac{N}{K}\begin{bmatrix} \hat{\boldsymbol{\pi}}^{(L)} \\ \hat{\mu}_i \end{bmatrix} = \begin{bmatrix} \hat{\boldsymbol{\pi}}^{(L)} \\ \hat{\mu}_i \end{bmatrix} \in \mathbb{R}^{(d+K)}, i \in [K].$$

Finally, we get

$$\mathbf{O} = [\mathbf{q}_{o1}, \cdots, \mathbf{q}_{oN}] = \begin{bmatrix} \hat{\boldsymbol{\pi}}^{(L)} & \hat{\boldsymbol{\pi}}^{(L)} & \cdots & \hat{\boldsymbol{\pi}}^{(L)} \\ \hat{\mu}_1 & \hat{\mu}_2 & \cdots & \hat{\mu}_K \end{bmatrix}.$$

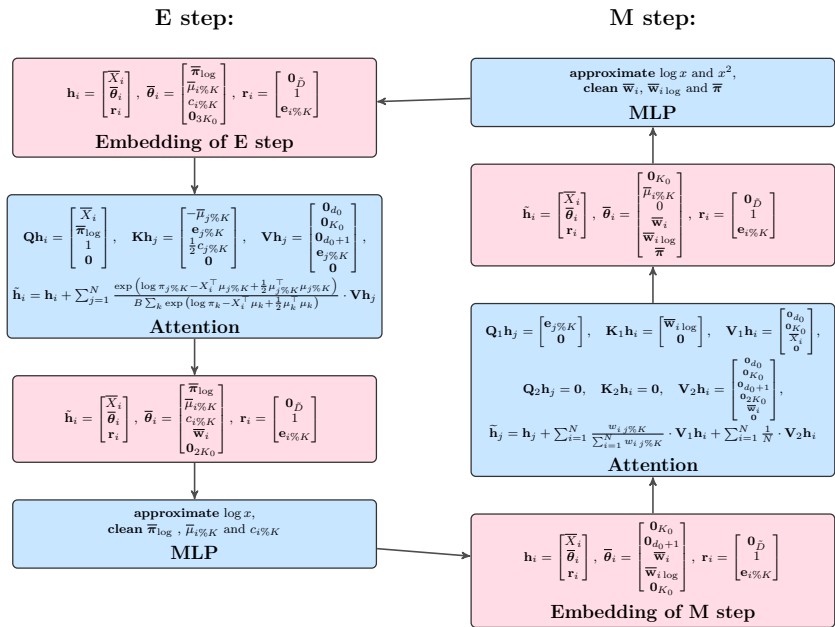

Figure 9: Transformer Construction for Approximating EM Algorithm Iterations. The *pink box* represents the state of tokens, while the *blue box* represents the structure of different parts of the network. The term "**clean**" means setting all positions of the corresponding vector to zero.

### F.3 CONVERGENCE RESULTS FOR EM ALGORITHM

#### F.3.1 CONVERGENCE RESULTS FOR POPULATION-EM ALGORITHM

First, we review some notations. Recall that $\pi_{min} = \min_i \pi_i^*$, $\rho_\pi = \max_i \pi_i^* / \min_i \pi_i^*$, $R_{ij} = \|\mu_i^* - \mu_j^*\|$, $R_{min} = \min_{i \neq j} R_{ij}$ and $R_{max} = (\max_{i \neq j} R_{ij}) \vee (\max_{i \in [K]} \|\mu_i^*\|)$. Without the loss of generality, we assume that $R_{max} \geq 1$. For clarity, we restate assumption ((A1)), which is consistent with Kwon & Caramanis (2020).

(A1) Suppose the GMM has parameters $\{(\pi_j^*, \mu_j^*) : j \in [K]\}$ such that

$$R_{\min} \geq C \cdot \sqrt{\log(\rho_\pi K)}, \tag{7}$$

and suppose the mean initialization $\mu_1^{(0)}, ..., \mu_K^{(0)}$ satisfies

$$\forall i \in [K], \left\|\mu_i^{(0)} - \mu_i^*\right\| \leq \frac{R_{\min}}{16}. \tag{8}$$

Also, suppose the mixing weights are initialized such that

$$\forall i \in [K], \left|\pi_i^{(0)} - \pi_i^*\right| \leq \pi_i/2. \tag{9}$$

For population-EM, the algorithm can be presented as

$$\text{(E-step):} \qquad w_i(X) = \frac{\pi_i \exp(-\|X - \mu_i\|^2/2)}{\sum_{j=1}^K \pi_j \exp(-\|X - \mu_j\|^2/2)},$$

$$\text{(M-step):} \qquad \pi_i^+ = \mathbb{E}[w_i], \quad \mu_i^+ = \mathbb{E}[w_i X]/\mathbb{E}[w_i].$$

The following results gives linear convergenve guarantees of population-EM, which comes from Kwon & Caramanis (2020).

**Theorem F.2** (Kwon & Caramanis (2020), Theorem 1, part i). *Let $C \geq 64$ be a universal constant for which assumption ((A1)) holds. Then, after one-step population-EM update, we have*

$$\forall i \in [K], |\pi_i^+ - \pi_i^*| \leq \pi_i^*/2, \ \|\mu_i^+ - \mu_i^*\| \leq 1/2. \tag{10}$$

Now we define

$$D_m = \max_{i \in [K]} \left(\|\mu_i - \mu_i^*\| \vee |\pi_i - \pi_i^*|/\pi_i^*\right),$$

and

$$D_m^+ = \max_{i \in [K]} \left(\|\mu_i^+ - \mu_i^*\| \vee |\pi_i^+ - \pi_i^*|/\pi_i^*\right).$$

The linear convergence of population-EM is stated by the following theorem.

**Theorem F.3** (Kwon & Caramanis (2020), Theorem 1, part ii). *Let $C \geq 128$ be a large enough universal constant. Fix $0 < a < 1$. Suppose the separation condition (7) holds and suppose the initialization parameter satisfies $D_m \leq a$, then $D_m^+ \leq \beta D_m$ for some $0 < \beta < 1$.*

**Remark F.4.** *Here the contraction parameter $\beta$ is only dependent with $C$ and $a$. In other words, if we fix $a \in (0, 1)$, then for any $\beta \in (0, 1)$, there exists a large enough $C$ such that Theorem F.3 holds. For details, see Appendix E in Kwon & Caramanis (2020).*

Combing Theorem F.2 and Theorem F.3, we can get the linear convergence of population-EM algorithm.

#### F.3.2 CONVERGENCE RESULTS FOR EMPIRICAL-EM ALGORITHM

Now we consider the empirical-EM, i.e., Algorithm B.1. For convenience, the algorithm can be presented as

$$\text{(E-step):} \qquad w_i(X_\ell) = w_{\ell i} = \frac{\pi_i \exp(-\|X_\ell - \mu_i\|^2/2)}{\sum_{j=1}^K \pi_j \exp(-\|X_\ell - \mu_j\|^2/2)}$$

$$\text{(M-step):} \qquad \pi_i^+ = \frac{1}{n}\sum_{l=1}^n w_i(X_\ell), \ \mu_i^+ = \frac{\sum_{l=1}^n w_i(X_\ell)X_\ell}{\sum_{l=1}^n w_i(X_\ell)} = \frac{1}{n\pi_i^+}\sum_{l=1}^n w_i(X_\ell)X_\ell.$$

Similarly, we can define $D_m$ and $D_m^+$ in empirical sense.

For the linear convergence of empirical-EM, we have the following theorem.

**Theorem F.4.** *Fix $0 < \delta, \beta < 1$ and $0 < a < 1$. Let $C \geq 128$ be a large enough universal constant. Suppose the separation condition (7) holds and suppose the initialization parameter satisfies $D_m \leq a$. If $n$ is suffcient large such that*

$$\varepsilon_{unif} := \frac{\tilde{c}}{(1-a)\,\pi_{min}} \sqrt{\frac{Kd\log(\frac{\tilde{C}n}{\delta})}{n}} < a(1-\beta)$$

*where $\tilde{C} = 72K^2(\sqrt{d} + 2R_{max} + \frac{1}{1-a})^2$ and $\tilde{c}$ is a universal constant. Then*

$$D_m^+ \leq \beta D_m + \varepsilon_{unif} \leq a$$

*uniformly holds with probability at least $1 - \delta$.*

*Proof.* First, we have

$$\frac{|\pi_i^+ - \pi_i^*|}{\pi_i^*} = \frac{1}{\pi_i^*} \left| \frac{1}{n} \sum_{l=1}^{n} w_i(X_\ell) - \pi_i^* \right|$$

$$\leq \frac{1}{\pi_i^*} \left( \left| \frac{1}{n} \sum_{l=1}^{n} w_i(X_\ell) - \mathbb{E}\left[w_i(X)\right] \right| + \left| \mathbb{E}\left[w_i(X)\right] - \pi_i^* \right| \right)$$

$$:= (I) + (II).$$

By Theorem F.3, we get

$$(II) = \frac{1}{\pi_i^*} \left| \mathbb{E}\left[w_i(X)\right] - \pi_i^* \right| \leq \tilde{\beta} D_m.$$

And by Lemma F.2, we have

$$(I) = \frac{1}{\pi_i^*} \left| \frac{1}{n} \sum_{l=1}^{n} w_i(X_\ell) - \mathbb{E}\left[w_i(X)\right] \right| \leq \frac{\tilde{c}_1}{\pi_{min}} \sqrt{\frac{Kd\log(\frac{\tilde{C}_1 n}{\delta_1})}{n}},$$

where $\tilde{C}_1 = 18K^2(\sqrt{d} + 2R_{max} + \frac{1}{1-a})$ and $\tilde{c}_1$ is a suitable universal constant. Thus, by taking $\tilde{\beta} = \beta$. $\delta_1 = \delta/2$ and suitable $\tilde{c}$, $|\pi_i^+ - \pi_i^*|/\pi_i^* \leq \beta D_m + \varepsilon_{unif} \leq a, \forall i \in [K]$.

For the second term, we have

$$\|\mu_i^+ - \mu_i^*\| = \left\| \frac{1}{n\pi_i^+} \sum_{l=1}^{n} w_i(X_\ell)(X_\ell - \mu_i^*) \right\|$$

$$\leq \frac{1}{\pi_i^+} \left( \left\| \frac{1}{n} \sum_{l=1}^{n} w_i(X_\ell)(X_\ell - \mu_i^*) - \mathbb{E}[w_i(X)(X - \mu_i^*)] \right\| + \|\mathbb{E}[w_i(X)(X - \mu_i^*)]\| \right)$$

$$:= (III) + (IV),$$

By Theorem F.3 and Remark F.4 we get,

$$(IV) = \frac{1}{\pi_i^+} \|\mathbb{E}[w_i(X)(X - \mu_i^*)]\|$$

$$\overset{(i)}{\leq} \frac{1}{(1-a)\pi_i^*} \|\mathbb{E}[w_i(X)(X - \mu_i^*)]\|$$

$$= \frac{\mathbb{E}[w_i(X)]}{(1-a)\pi_i^*} \left\| \frac{\mathbb{E}[w_i(X)X]}{\mathbb{E}[w_i(X)]} - \mu_i^* \right\|$$

$$\leq \frac{1+a}{1-a} \tilde{\beta} D_m.$$

where $(i)$ follows from $|\pi_i^+ - \pi_i^*|/\pi_i^* \le a$. And by Lemma F.3, we have

$$(III) = \frac{1}{\pi_i^+} \left( \left\| \frac{1}{n} \sum_{l=1}^{n} w_i(X_\ell)(X_\ell - \mu_i^*) - \mathbb{E}[w_i(X)(X - \mu_i^*)] \right\| \right)$$

$$\le \frac{1}{(1-a)\pi_i^*} \left( \left\| \frac{1}{n} \sum_{l=1}^{n} w_i(X_\ell)(X_\ell - \mu_i^*) - \mathbb{E}[w_i(X)(X - \mu_i^*)] \right\| \right)$$

$$\le \frac{\tilde{c}_2}{(1-a)\pi_{min}} \sqrt{\frac{Kd \log(\frac{\tilde{C}_2 n}{\delta_2})}{n}},$$

where $\tilde{C} = 18K^2(\sqrt{d} + 2R_{max} + \frac{1}{1-a})^2$ and $\tilde{c}$ is a suitable universal constant. Thus, by taking $\tilde{\beta} = (1-a)/(1+a)\beta$, $\delta_2 = \delta/2$ and suitable $\tilde{c}$, $\|\mu_i^+ - \mu_i^*\| \le \beta D_m + \varepsilon_{unif} \le a$, $\forall i \in [K]$.

In conclusion, if we take $\tilde{\beta} = (1-a)/(1+a)\beta$, $\tilde{c} = \tilde{c}_1 \vee \tilde{c}_2$, $C = C(\beta, a) \ge 128$ large enough such that Theorem F.3 holds, and take $\delta_1 = \delta_2 = \delta/2$ and use union bound argument, then we get $D_m^+ \le \beta D_m + \varepsilon_{unif} \le a$.

$\square$

We need the following technical lemma.

**Lemma F.1** (Segol & Nadler (2021), Lemma B.2.)**.** *Fix $0 < \delta < 1$. Let $B_1, \ldots, B_K \subset \mathbb{R}^d$ be Euclidean balls of radii $r_1, \ldots, r_K$. Define $\mathcal{B} = \otimes_{k=1}^{K} B_k \subset \mathbb{R}^{Kd}$ and $r = \max_{k \in [K]} r_k$. Let $X$ be a random vector in $\mathbb{R}^d$ and $W : \mathbb{R}^d \times \mathcal{B} \to \mathbb{R}^k$ where $k \le d$. Assume the following hold:*

*1. There exists a constant $L \ge 1$ such that for any $\theta \in \mathcal{B}, \varepsilon > 0$, and $\theta^\varepsilon \in \mathcal{B}$ which satisfies $\max_{i \in [K]} \|\theta_i - \theta_i^\varepsilon\| \le \varepsilon$, then $\mathbb{E}_X \left[ \sup_{\mu \in \mathcal{B}} \|W(X, \theta) - W(X, \theta^\varepsilon)\| \right] \le L\varepsilon$.*

*2. There exists a constant $R$ such that for any $\theta \in \mathcal{B}$, $\|W(X, \theta)\|_{\psi_2} \le R$.*

*Let $X_1, \ldots, X_n$ be i.i.d. random vectors with the same distribution as $X$. Then there exists a universal constant $\tilde{c}$ such that with probability at least $1 - \delta$,*

$$\sup_{\theta \in \mathcal{B}} \left\| \frac{1}{n} \sum_{\ell=1}^{n} W(X_\ell, \theta) - \mathbb{E}[W(X, \theta)] \right\| \le R \sqrt{\tilde{c} \frac{Kd \log\left(1 + \frac{12nLr}{\delta}\right)}{n}}. \tag{11}$$

**Remark F.5.** *There is one difference between Lemma F.1 and LemmaB.2. in Segol & Nadler (2021): in Lemma F.1, we use $1 + \frac{12nLr}{\delta}$ to replace $\frac{18nLr}{\delta}$, thus we avoid the condition $r_1, \cdots, r_K \ge 1$.*

Hence we can get the uniform convergence of $w_i(X, \theta)$ and $w_i(X, \theta)(X - \mu_i^*)$, $i \in [K]$. Our proof is similar to Segol & Nadler (2021), except that we consider the variation of both $\pi$ and $\mu$. From now on, we denote $\theta_i = \{\pi_i, \mu_i\}$, $\theta = \{\theta_i\}_{i=1}^{n}$.

**Lemma F.2.** *Fix $0 < \delta < 1$ and $0 < a < 1$. Consider the parameter region $\mathcal{D}_a := \{D_m \le a\}$. Let $X_1, \cdots, X_n \overset{i.i.d.}{\sim} GMM(\pi^*, \mu^*)$, then with probability at least $1 - \delta$,*

$$\sup_{\theta \in \mathcal{D}_a} \left| \frac{1}{n} \sum_{\ell=1}^{n} w_i(X_\ell, \theta) - \mathbb{E}[w_i(X, \theta)] \right| \le \tilde{c} \sqrt{\frac{Kd \log(\frac{\tilde{C}n}{\delta})}{n}}, \ \forall i \in [K], \tag{12}$$

*where $\tilde{C} = 18K^2(\sqrt{d} + 2R_{max} + \frac{1}{1-a})$ and $\tilde{c}$ is a suitable universal constant.*

*Proof.* The proof is similar to the proof of Lemma 5.1 in Segol & Nadler (2021). For simplicity, we omit it. $\square$

**Lemma F.3.** *Fix $0 < \delta < 1$ and $0 < a < 1$. Consider the parameter region $\mathcal{D}_a := \{D_m \le a\}$. Let $X_1, \cdots, X_n \overset{i.i.d.}{\sim} GMM(\pi^*, \mu^*)$ with $R_{min}$ satisfying (7), then with probability at least $1 - \delta$,*

$$\sup_{\theta \in \mathcal{D}_a} \left| \frac{1}{n} \sum_{\ell=1}^{n} w_i(X_\ell, \theta)(X_\ell - \mu_i^*) - \mathbb{E}[w_i(X, \theta)(X_\ell - \mu_i^*)] \right| \le \tilde{c} \sqrt{\frac{Kd \log(\frac{\tilde{C}n}{\delta})}{n}}, \ \forall i \in [K], \tag{13}$$

*where $\tilde{C} = 36K^2(\sqrt{d} + 2R_{max} + \frac{1}{1-a})^2$ and $\tilde{c}$ is a suitable universal constant.*

*Proof.* The proof is similar to the proof of Lemma 5.4 in Segol & Nadler (2021)(Notice that the condition (36) in Segol & Nadler (2021) is trivial in our case). For simplicity, we omit it. □

For the first step empirical-EM, we have the following results.

**Theorem F.5.** *Fix $0 < \delta < 1$ and $1/2 < a < 1$. Let $C \geq 128$ be a large enough universal constant for which assumption ((A1)) holds. If $n$ is suffcient large such that*

$$\varepsilon_{step1} := \frac{\tilde{c}}{(1-a)\pi_{min}} \sqrt{\frac{R_{max}(R_{max} \vee d) \log\left(\frac{6K}{\delta}\right)}{n}} < \left(a - \frac{1}{2}\right),$$

*where $\tilde{c}$ is a universal constant. Then*

$$D_m^+ \leq \frac{1}{2} + \varepsilon_{step1} \leq a$$

*holds with probability at least $1 - \delta$.*

*Proof.* Notice that we only need simple concentration not uniform concentration in this theorem. We use the same definition of term $(I)$, $(II)$ as in the proof of Theorem F.4. First, by Theorem F.2, we have $(II) \leq 1/2$. Since $0 \leq w_i(X) \leq 1$, by a standard concentration of bounded variables, we can get

$$(I) \leq \frac{\tilde{c}_1}{\pi_{min}} \sqrt{\frac{\log(\frac{K}{\delta_1})}{n}}, \forall i \in [K], \tag{14}$$

where $\tilde{c}_1$ is a universal constant. Taking $\tilde{c} \geq \tilde{c}_1$ and $\delta_1 = \delta/2$, we have

$$\frac{|\pi_i^+ - \pi_i^*|}{\pi_i^*} \leq \frac{1}{2} + \frac{\tilde{c}}{\pi_{min}} \sqrt{\frac{\log\left(\frac{2K}{\delta}\right)}{n}} \leq a, \forall i \in [K].$$

For the second term, we have

$$\|\mu_i^+ - \mu_i^*\| = \left\| \frac{1}{n\pi_i^+} \sum_{l=1}^n w_i(X_\ell)X_\ell - \mu_i^* \right\|$$

$$\leq \left\| \frac{1}{n\pi_i^+} \sum_{l=1}^n w_i(X_\ell)X_\ell - \frac{\mathbb{E}[w_i(X_\ell)X_\ell]}{\mathbb{E}[w_i(X_\ell)]} \right\| + \left\| \frac{\mathbb{E}[w_i(X_\ell)X_\ell]}{\mathbb{E}[w_i(X_\ell)]} - \mu_i^* \right\|$$

$$:= (V) + (VI).$$

By Theorem F.2, we have $(VI) \leq 1/2$. For $(V)$, by triangle inequality,

$$(V) \leq \left\| \frac{1}{n\pi_i^+} \sum_{l=1}^n w_i(X_\ell)X_\ell - \frac{1}{\pi_i^+}\mathbb{E}[w_i(X_\ell)X_\ell] \right\| + \left\| \frac{1}{\pi_i^+}\mathbb{E}[w_i(X_\ell)X_\ell] - \frac{\mathbb{E}[w_i(X_\ell)X_\ell]}{\mathbb{E}[w_i(X_\ell)]} \right\|$$

$$= \frac{1}{\pi_i^+} \left\| \frac{1}{n} \sum_{l=1}^n w_i(X_\ell)X_\ell - \mathbb{E}[w_i(X_\ell)X_\ell] \right\| + \frac{\|\mathbb{E}[w_i(X_\ell)X_\ell]\|}{\pi_i^+\mathbb{E}[w_i(X_\ell)]} |\pi_i^+ - \mathbb{E}[w_i(X_\ell)]|. \tag{15}$$

Using Lemma B.1 and Lemma B.2 in Zhao et al. (2020), we can get $\|w_i(X_\ell)X_\ell\|_{\psi_2} \leq \|X_\ell\|_{\psi_2} \leq \tilde{c}_3 R_{max}, \forall i \in [K]$. Hence by Lemma B.1 in Segol & Nadler (2021), with probability at least $1 - \delta_2$,

$$\left\| \frac{1}{n} \sum_{l=1}^n w_i(X_\ell)X_\ell - \mathbb{E}[w_i(X_\ell)X_\ell] \right\| \leq \tilde{c}_4 \sqrt{\frac{R_{max}d \log\left(\frac{3K}{\delta_2}\right)}{n}}, \forall i \in [K],$$

where $\tilde{c}_4$ is an universal constant. And by Theorem F.2, we have

$$\frac{\|\mathbb{E}[w_i(X_\ell)X_\ell]\|}{\mathbb{E}[w_i(X_\ell)]} \leq R_{max} + \frac{1}{2} \leq 2R_{max}, \forall i \in [K].$$

Finally, by (14),

$$\left|\pi_i^+ - \mathbb{E}[w_i(X_\ell)]\right| \le \tilde{c}_1 \sqrt{\frac{\log(\frac{K}{\delta_1})}{n}}.$$

Combining all terms together and taking $\delta_1 = \delta_2 = \delta/2$ we can bound (15) by

$$(V) \le \frac{1}{\pi_i^+} \left( \tilde{c}_4 \sqrt{\frac{R_{max} d \log\left(\frac{3K}{\delta_2}\right)}{n}} + 2\tilde{c}_1 R_{max} \sqrt{\frac{\log(\frac{K}{\delta_1})}{n}} \right)$$

$$\le \frac{\tilde{c}_6}{(1-a)\pi_{min}} \sqrt{\frac{R_{max}(R_{max} \vee d) \log\left(\frac{6K}{\delta}\right)}{n}}.$$

Taking $\tilde{c} \ge \tilde{c}_6$, we get

$$\|\mu_i^+ - \mu_i^*\| \le \frac{1}{2} + \frac{\tilde{c}}{(1-a)\pi_{min}} \sqrt{\frac{R_{max}(R_{max} \vee d) \log\left(\frac{6K}{\delta}\right)}{n}} \le a.$$

$\square$

### F.3.3 CONVERGENCE RESULTS FOR TRANSFORMER-BASED EM IN SECTION F.2

We first state some useful approximation lemmas.

**Lemma F.4** (Lemma 9 in Mei (2024)). *For any $A > 0$, $\delta > 0$, take $M = \lceil 2\log A/\delta \rceil + 1 \in \mathbb{N}$. Then there exists $\{(a_j, w_j, b_j)\}_{j \in [M]}$ with*

$$\sup_j |a_j| \le 2A, \quad \sup_j |w_j| \le 1, \quad \sup_j |b_j| \le A, \tag{16}$$

*such that defining $\log_\delta : \mathbb{R} \to \mathbb{R}$ by*

$$\log_\delta(x) = \sum_{j=1}^M a_j \cdot \mathrm{ReLU}(w_j x + b_j),$$

*we have $\log_\delta$ is non-decreasing on $[1/A, A]$, and*

$$\sup_{x \in [1/A, A]} |\log(x) - \log_\delta(x)| \le \delta.$$

**Remark F.6.** *There is a small improvement $M = \lceil 2\log A/\delta \rceil + 1$ compared to $M = \lceil 2A/\delta \rceil + 1$ in Mei (2024). Further more, it is easy to check that $\log_\delta(x) \le -\log A$ for $x \in [0, 1/A]$.*

**Lemma F.5.** *For any $A > 0$, $\delta > 0$, take $M = \lceil 2A^2/\delta \rceil + 1 \in \mathbb{N}$. Then there exists $\{(a_j, w_j, b_j)\}_{j \in [M]}$ with*

$$\sup_j |a_j| \le 2A, \quad \sup_j |w_j| \le 1, \quad \sup_j |b_j| \le A, \tag{17}$$

*such that defining $\phi_\delta : \mathbb{R} \to \mathbb{R}$ by*

$$\phi_\delta(x) = \sum_{j=1}^M a_j \cdot \mathrm{ReLU}(w_j x + b_j),$$

*we have $\phi_\delta$ is non-decreasing on $[-A, A]$, and*

$$\sup_{x \in [-A, A]} |\phi_\delta(x) - x^2| \le \delta.$$

*Proof.* Similar to Lemma F.4. Omitted. $\square$

**Lemma F.6** (Lemma A.1 in Bai et al. (2023))**.** *Let $\beta \sim \mathcal{N}(0, I_d)$. Then we have*

$$\mathbb{P}\left( \|\beta\|^2 \geq d(1+\delta)^2 \right) \leq e^{-d\delta^2/2}.$$

**Lemma F.7** (Lemma 18 in Lin et al. (2024))**.** *For any $\mathbf{u}, \mathbf{v} \in \mathbb{R}^d$, we have*

$$\left\| \log\left( \frac{e^{\mathbf{u}}}{\|e^{\mathbf{u}}\|_1} \right) - \log\left( \frac{e^{\mathbf{v}}}{\|e^{\mathbf{v}}\|_1} \right) \right\|_{\infty} \leq 2 \|\mathbf{u} - \mathbf{v}\|_{\infty}.$$

**Corollary F.1.** *For any $\mathbf{u}, \mathbf{v} \in \mathbb{R}^d$, we have*

$$\left\| \frac{e^{\mathbf{u}}}{\|e^{\mathbf{u}}\|_1} - \frac{e^{\mathbf{v}}}{\|e^{\mathbf{v}}\|_1} \right\|_{\infty} \leq \exp\left( 2 \|\mathbf{u} - \mathbf{v}\|_{\infty} \right) - 1$$

*Proof.* This follows directly from Lemma F.7 and simple calculations. $\qquad\square$

Now we propose the results for transformer-based EM. Similar to Section F.2, we use notations with superscript "^" to represent the output of the transformer-based EM.

**Theorem F.6.** *Fix $0 < \delta < 1$ and $1/2 < a < 1$. Let $C \geq 128$ be a large enough universal constant for which assumption ((A1)) holds. If $n$ is sufficient large and $\varepsilon \leq 1/100$ sufficient small such that*

$$\frac{\tilde{c}_1}{(1-a)\pi_{min}} \sqrt{\frac{R_{max}(R_{max} \vee d) \log\left( \frac{12K}{\delta} \right)}{n}}$$
$$+ \tilde{c}_2 \left( R_{\max} + d\left( 1 + \sqrt{\frac{2\log(\frac{2n}{\delta})}{d}} \right) \right) n\varepsilon < \frac{1}{2}\left( a - \frac{1}{2} \right),$$

*where $\tilde{c}_1$, $\tilde{c}_2$ are universal constants. Then there exists a 2-layer transformer $\mathrm{TF}_{\mathbf{\Theta}}$ such that $\hat{D}_m^+ \leq a$ holds with probability at least $1 - \delta$. Moreover, $\mathrm{TF}_{\mathbf{\Theta}}$ falls within the class $\mathcal{F}$ with parameters satisfying:*

$$D = O(d_0 + K_0), D' \leq \tilde{O}\left( K_0 R_{\max}(R_{\max} + d_0)\varepsilon^{-1} \right),$$
$$M = O(1),$$
$$\log B_{\mathbf{\Theta}} \leq O(K_0 R_{\max}(R_{\max} + d_0)).$$

*Proof.* Recall that $\hat{D}_m^+ = \max_{i \in [K]} \left( \|\hat{\mu}_i^+ - \mu_i^*\| \vee |\hat{\pi}_i^+ - \pi_i^*|/\pi_i^* \right)$. Thus

$$\hat{D}_m^+ \leq \max_{i \in [K]} \left( \|\mu_i^+ - \mu_i^*\| \vee |\pi_i^+ - \pi_i^*|/\pi_i^* \right) + \max_{i \in [K]} \left( \|\hat{\mu}_i^+ - \mu_i^+\| \vee |\hat{\pi}_i^+ - \pi_i^+|/\pi_i^* \right)$$
$$= D_m^+ + \max_{i \in [K]} \left( \|\hat{\mu}_i^+ - \mu_i^+\| \vee |\hat{\pi}_i^+ - \pi_i^+|/\pi_i^* \right)$$

We first claim that with probability at least $1 - \delta/2$,

$$\max_{i \in [K]} \left( \|\hat{\mu}_i^+ - \mu_i^+\| \vee |\hat{\pi}_i^+ - \pi_i^+|/\pi_i^* \right) \leq \tilde{c}_2 \left( R_{\max} + d\left( 1 + \sqrt{\frac{2\log(\frac{2n}{\delta})}{d}} \right) \right) n\varepsilon. \quad (18)$$

Then by Theorem F.5, with probability at least $1 - \delta$, we have

$$\hat{D}_m^+ \leq D_m^+ + \max_{i \in [K]} \left( \|\hat{\mu}_i^+ - \mu_i^+\| \vee |\hat{\pi}_i^+ - \pi_i^+|/\pi_i^* \right)$$
$$\leq \frac{1}{2} + \frac{\tilde{c}}{(1-a)\pi_{min}} \sqrt{\frac{R_{max}(R_{max} \vee d) \log\left( \frac{12K}{\delta} \right)}{n}} + \tilde{c}_2 \left( R_{\max} + d\left( 1 + \sqrt{\frac{2\log(\frac{2n}{\delta})}{d}} \right) \right) n\varepsilon$$
$$\leq a.$$

Now we only need to prove (18). By the construction in Section F.2, we can see that $w_{\ell i}$ in first step can be well calculated, thus $|\hat{\pi}_i^+ - \pi_i^+| = 0$ and the error comes only from the calculation of $\{\hat{\mu}_i^+\}$. Recall that $\mu_i^+ = \frac{\sum_{\ell=1}^n w_{\ell i} X_\ell}{\sum_{\ell=1}^n w_{\ell i}}$ and

$$\hat{\mu}_i^+ = \frac{\sum_{\ell=1}^n \exp\left(\widehat{\log}(w_{\ell i})\right) X_\ell}{\sum_{l=1}^n \exp\left(\widehat{\log}(w_{\ell i})\right)}.$$

Recall that

$$w_{\ell i} = \frac{\pi_i \exp(-\|X_\ell - \mu_i\|^2/2)}{\sum_{j=1}^K \pi_j \exp(-\|X_\ell - \mu_j\|^2/2)} = \frac{1}{1 + \sum_{j \neq i} \frac{\pi_j}{\pi_i} \exp\left((\mu_j - \mu_i)^\top X_l - \|\mu_j\|^2/2 + \|\mu_i\|^2/2\right)}.$$

By the initial condition (9) and (8), we have

$$\|\mu_j - \mu_i\| \leq R_{\max} + 2 * \frac{1}{16} R_{\min} = O(R_{\max}), \quad \|\mu_j\|^2 = O(R_{\max}^2).$$

Since $X_\ell \overset{\text{i.i.d.}}{\sim} \text{GMM}(\pi^*, \mu^*)$, using Lemma F.6, with probability at least $1 - \delta/2$, we have

$$\sup_{\ell \in [n]} \|X_\ell\| \leq R_{\max} + d\left(1 + \sqrt{\frac{2\log(\frac{2n}{\delta})}{d}}\right) = \tilde{O}(R_{\max} + d).$$

Combine all things together, we get that with probability at least $1 - \delta/2$,

$$w_{\ell i}^{-1} \leq \exp\left(\tilde{O}(K_0 R_{\max}(R_{\max} + d_0))\right), \quad \forall \ell \in [n] \text{ and } i \in [K].$$

Thus taking $A = \exp\left(\tilde{O}(K_0 R_{\max}(R_{\max} + d_0))\right)$ and and $\delta = \varepsilon$ in Lemma F.4, we can get $|\log - \widehat{\log}|\big|_{[1/A,A]} \leq \varepsilon$. Then by Lemma F.7, we have

$$\|\hat{\mu}_i^+ - \mu_i^+\| = \left\|\frac{\sum_{\ell=1}^n \exp\left(\widehat{\log}(w_{\ell i})\right) X_\ell}{\sum_{l=1}^n \exp\left(\widehat{\log}(w_{\ell i})\right)} - \frac{\sum_{\ell=1}^n \exp\left(\log w_{\ell i}\right) X_\ell}{\sum_{\ell=1}^n \exp\left(\log w_{\ell i}\right)}\right\|$$

$$\leq \sum_{\ell=1}^n \left\|\frac{\exp\left(\widehat{\log}(w_{\ell i})\right) X_\ell}{\sum_{l=1}^n \exp\left(\widehat{\log}(w_{\ell i})\right)} - \frac{\exp\left(\log w_{\ell i}\right) X_\ell}{\sum_{\ell=1}^n \exp\left(\log w_{\ell i}\right)}\right\|$$

$$\leq \sup_{\ell \in [n]} \|X_\ell\| \left(\sum_{\ell=1}^n \left|\frac{\exp\left(\widehat{\log}(w_{\ell i})\right)}{\sum_{l=1}^n \exp\left(\widehat{\log}(w_{\ell i})\right)} - \frac{\exp\left(\log w_{\ell i}\right)}{\sum_{\ell=1}^n \exp\left(\log w_{\ell i}\right)}\right|\right)$$

$$\leq n\left(R_{\max} + d + \sqrt{2d\log\left(\frac{2n}{\delta}\right)}\right)\left(\exp\left(2\left\|\left(\widehat{\log}(w_{\ell i})\right)_\ell - (\log(w_{\ell i}))_\ell\right\|_\infty\right) - 1\right)$$

$$\leq 4n\left(R_{\max} + d + \sqrt{2d\log\left(\frac{2n}{\delta}\right)}\right)\varepsilon, \quad \forall i \in [K].$$

Thus (18) is proved. The parameter bounds can be directly computed by the construction in Section F.2 and Lemma F.4.

$\square$

**Theorem F.7.** *Fix $0 < \delta, \beta < 1$ and $1/2 < a < 1$. Let $C \geq 128$ be a large enough universal constant. Suppose the separation condition (7) holds and suppose the initialization parameter input to transformer satisfies $D_m \leq a$. If $n$ is suffcient large and $K_0\varepsilon \leq 1/100$ sufficient small such that*

$$\epsilon(n, \varepsilon, \delta, a) := \frac{\tilde{c}_1}{(1-a)\pi_{\min}} \sqrt{\frac{Kd\log(\frac{\tilde{C}n}{\delta})}{n}} + \tilde{c}_2\left(\frac{1}{\pi_{\min}} + n\left(R_{\max} + d + \sqrt{2d\log\left(\frac{2n}{\delta}\right)}\right)\right)\varepsilon$$

$$< a(1 - \beta),$$

where $\tilde{c}_1$, $\tilde{c}_2$ are universal constants, $\tilde{C} = 144K^2(\sqrt{d} + 2R_{max} + \frac{1}{1-a})^2$. Then there exists a 2-layer transformer $\mathrm{TF}_\Theta$ such that

$$\hat{D}_m^+ \leq \beta D_m + \epsilon(n, \varepsilon, \delta, a) \leq a$$

uniformly holds with probability at least $1 - \delta$. Moreover, $\mathrm{TF}_\Theta$ falls within the class $\mathcal{F}$ with parameters satisfying:

$$D = O(d_0 + K_0), D' \leq \tilde{O}\big(K_0 R_{\max}(R_{\max} + d_0)\varepsilon^{-1}\big),$$
$$M = O(1),$$
$$\log B_\Theta \leq \tilde{O}(K_0 R_{\max}(R_{\max} + d_0)).$$

*Proof.* Similar to the proof of Theorem F.6, using Theorem F.4, we only need to prove that with probability at least $1 - \delta/2$,

$$\max_{i \in [K]} \left( \|\hat{\mu}_i^+ - \mu_i^+\| \vee |\hat{\pi}_i^+ - \pi_i^+|/\pi_i^* \right) \leq \tilde{c}_2 \left( \frac{1}{\pi_{\min}} + n\left( R_{\max} + d + \sqrt{2d \log\left(\frac{2n}{\delta}\right)} \right) \right) \varepsilon. \quad (19)$$

Define $u_\ell = (u_{\ell,1}, \cdots, u_{\ell,K})^\top$, $\hat{u}_\ell = (\hat{u}_{\ell,1}, \cdots, \hat{u}_{\ell,K})^\top$, where $u_{\ell,i} = \log \pi_i + \mu_i^\top X_\ell - 1/2\|\mu_i\|^2$ and $\hat{u}_{\ell,i} = \widehat{\log \pi_i} + \mu_i^\top X_\ell - 1/2\widehat{\|\mu_i\|^2}$ By the construction in Section F.2 and Corollary F.1, we have

$$\|\hat{\mathbf{w}}_\ell - \mathbf{w}_\ell\|_\infty = \left\| \frac{e^{\hat{u}_\ell}}{\|e^{\hat{u}_\ell}\|_1} - \frac{e^{u_\ell}}{\|e^{u_\ell}\|_1} \right\|_\infty \leq \exp\left(2\|\hat{u}_\ell - u_\ell\|_\infty\right) - 1, \ \forall \ell \in [n].$$

Now taking $\delta = \varepsilon$, $A = ((1-a)\pi_{\min})^{-1}$ in Lemma F.4 and $\delta = \varepsilon/K$, $A = (R_{\max} + a)^2$ in Lemma F.5, we have $\|\hat{u}_\ell - u_\ell\|_\infty \leq 3\varepsilon/2$, hence

$$\|\hat{\mathbf{w}}_\ell - \mathbf{w}_\ell\|_\infty \leq \exp\left(2\|\hat{u}_\ell - u_\ell\|_\infty\right) - 1 \leq \exp(3\varepsilon) - 1 \leq 6\varepsilon, \ \forall \ell \in [n].$$

Then by the construction in Section F.2, we have

$$|\hat{\pi}_i^+ - \pi_i^+| \leq 6\varepsilon, \ \forall i \in [K]. \quad (20)$$

For the term $\|\hat{\mu}_i^+ - \mu_i^+\|$, we can calculate it similar to the proof of Theorem F.6. First, we recall that with probability at least $1 - \delta/2$,

$$w_{\ell i}^{-1} \leq \exp\left(\tilde{O}(K_0 R_{\max}(R_{\max} + d_0))\right), \ \forall \ell \in [n] \text{ and } i \in [K].$$

Similarly, for $\hat{w}_{\ell,i}$, we can also get(just calculate again) that with probability at least $1 - \delta/2$,

$$\hat{w}_{\ell i}^{-1} \leq \exp\left(\tilde{O}(K_0 R_{\max}(R_{\max} + d_0))\right), \ \forall \ell \in [n] \text{ and } i \in [K].$$

Then following the same argument in Theorem F.6, taking $A = \exp\left(\tilde{O}(K_0 R_{\max}(R_{\max} + d_0))\right)$ and and $\delta = \varepsilon$ in Lemma F.4, we have also

$$\|\hat{\mu}_i^+ - \mu_i^+\| \leq 4n\left( R_{\max} + d + \sqrt{2d \log\left(\frac{2n}{\delta}\right)} \right)\varepsilon, \ \forall i \in [K]. \quad (21)$$

Combining (20) and (21), (19) is proved. The parameter bounds can be directly computed by the construction in Section F.2, Lemma F.4, Lemma F.5 and the parameter $A$, $\delta$ taken in the proof.

$\square$

## F.4 PROOF OF THEOREM F.1

First, by Theorem F.6 and the first condition in Theorem F.1 , there exist a 2-layer transformer $\mathrm{TF}_{\Theta_1}$ such that

$$D_{\Theta_1}^{\mathrm{TF}} \leq a, \quad (22)$$

holds with probability at least $1 - \delta/2$. Then using Theorem F.3, (22) and the second condition in Theorem F.1, there 2-layer transformer $\mathrm{TF}_{\boldsymbol{\Theta}_2}$ such that

$$D_{\boldsymbol{\Theta}_1 \cup \boldsymbol{\Theta}_2}^{\mathrm{TF}} \leq \beta D_{\boldsymbol{\Theta}_1}^{\mathrm{TF}} + \epsilon(n, \varepsilon, \delta/2, a) \leq a,$$

uniformly holds with probability at least $1 - \delta/2$. Denote as $\boldsymbol{\Theta}_2^L = \cup_{\ell \in [L]} \boldsymbol{\Theta}_2$. Thus, for any $L \in \mathbb{N}$, by reduction, we have

$$D_{\boldsymbol{\Theta}_1 \cup \boldsymbol{\Theta}_2^L}^{\mathrm{TF}} \leq \beta^L D_{\boldsymbol{\Theta}_1}^{\mathrm{TF}} + \left(1 + \beta + \cdots + \beta^{L-1}\right) \epsilon(n, \varepsilon, \delta/2, a),$$

uniformly holds with probability at least $1 - \delta/2$. Combine all things together, we have, for any $L \in \mathbb{N}$,

$$D_{\boldsymbol{\Theta}_1 \cup \boldsymbol{\Theta}_2^L}^{\mathrm{TF}} \leq \beta^L D_{\boldsymbol{\Theta}_1}^{\mathrm{TF}} + \left(1 + \beta + \cdots + \beta^{L-1}\right) \epsilon(n, \varepsilon, \delta/2, a)$$

$$\leq \beta^L a + \frac{1}{1 - \beta} \epsilon(n, \varepsilon, \delta/2, a)$$

holds with probability at least $1 - \delta$ (Note that the definitions of $\epsilon(\cdot)$ in Theorem F.7 and Theorem F.1 differ slightly). The parameter bounds can be directly computed by Theorem F.6 and Theorem F.7. The theorem is proved.

# G  FORMAL STATEMENT OF THEOREM 2 AND PROOFS

Following Section F, we implement $\mathrm{Readin}$ as an identity transformation and define $\mathrm{Readout}$ to extract targeted matrix elements hence they are both fixed functions.

## G.1  FORMAL STATEMENT OF THEOREM 2

In this section, we give the formal statement of Theorem 2. First, we need to introduce the embeddings of the transformer. Let $\mathbf{T}$ be the matrix representation of the cubic tensor $T$, which is

$$\mathbf{T} := [\mathbf{t}_1, \mathbf{t}_2, \cdots, \mathbf{t}_d] := \begin{bmatrix} T_{:,1,1} & T_{:,2,1} & \cdots & T_{:,d,1} \\ T_{:,1,2} & T_{:,2,2} & \cdots & T_{:,d,2} \\ \vdots & \vdots & \ddots & \vdots \\ T_{:,1,d} & T_{:,2,d} & \cdots & T_{:,d,d} \end{bmatrix} \in \mathbb{R}^{d^2 \times d},$$

where $T_{:,i,j} = (T_{1,i,j}, T_{2,i,j}, \cdots, T_{d,i,j}) \in \mathbb{R}^d$, $i, j \in [d]$. For dimension adaptation, we assume $d \leq d_0$. The augment version of $\mathbf{T}$ is defined as

$$\overline{\mathbf{T}} := \left[\overline{\mathbf{t}}_1, \overline{\mathbf{t}}_2, \cdots, \overline{\mathbf{t}}_{d_0}\right] := \begin{bmatrix} \overline{T}_{:,1,1} & \overline{T}_{:,2,1} & \cdots & \overline{T}_{:,d_0,1} \\ \overline{T}_{:,1,2} & \overline{T}_{:,2,2} & \cdots & \overline{T}_{:,d_0,2} \\ \vdots & \vdots & \ddots & \vdots \\ \overline{T}_{:,1,d_0} & \overline{T}_{:,2,d_0} & \cdots & \overline{T}_{:,d_0,d_0} \end{bmatrix} \in \mathbb{R}^{d_0^2 \times d_0}, \tag{23}$$

where $\overline{T}_{:,i,j} \in \mathbb{R}^{d_0}$. If $i \leq d$ and $j \leq d$, $\overline{T}_{:,i,j} = \left[T_{:,i,j}^\top, \mathbf{0}_{d_0-d}^\top\right]^\top$; Else $\overline{T}_{:,i,j} = \mathbf{0}_{d_0}$. We construct the following input sequence:

$$\mathbf{H} = \begin{bmatrix} \overline{\mathbf{t}}_1 & \overline{\mathbf{t}}_2 & \cdots & \overline{\mathbf{t}}_d \\ \mathbf{p}_1 & \mathbf{p}_2 & \cdots & \mathbf{p}_d \end{bmatrix} \in \mathbb{R}^{D \times d}, \ \mathbf{p}_i = \begin{bmatrix} \overline{v}^{(0)} \\ \mathbf{e}_i \\ 1 \\ d \\ \mathbf{0}_{\tilde{D}} \end{bmatrix}, \tag{24}$$

where $\overline{\mathbf{t}}_i \in \mathbb{R}^{d_0^2}$ is defined as (23), $\overline{v}^{(0)} = \left[v^{(0)\top}, \mathbf{0}_{d_0-d}^\top\right]^\top \in \mathbb{R}^{d_0}$, $\mathbf{e}_i \in \mathbb{R}^{d_0}$ denotes the $i$-th standard unit vector in $\mathbb{R}^{d_0}$. We choose $D = O(d_0^2)$ and $\tilde{D} = D - d_0^2 - 2d_0 - 2$ to get the encoding above. Then we give a rigorous definition of ReLU-activated transformer following Bai et al. (2023).

**Definition 7** (ReLU-attention layer)**.** *A (self-)attention layer activated by ReLU function with $M$ heads is denoted as $\mathrm{Attn}_{\boldsymbol{\Theta}_{\mathrm{attn}}}(\cdot)$ with parameters $\boldsymbol{\Theta}_{\mathrm{attn}} = \{(\mathbf{V}_m, \mathbf{Q}_m, \mathbf{K}_m)\}_{m \in [M]} \subset \mathbb{R}^{D \times D}$. On any input sequence $\mathbf{H} \in \mathbb{R}^{D \times N}$,*

$$\widetilde{\mathbf{H}} = \mathrm{Attn}_{\boldsymbol{\Theta}_{\mathrm{attn}}}^R(\mathbf{H}) := \mathbf{H} + \frac{1}{N} \sum_{m=1}^M (\mathbf{V}_m \mathbf{H}) \sigma\left((\mathbf{K}_m \mathbf{H})^\top (\mathbf{Q}_m \mathbf{H})\right) \in \mathbb{R}^{D \times N},$$

*In vector form,*

$$\widetilde{\mathbf{h}}_i = \left[\mathrm{Attn}^R_{\boldsymbol{\Theta}_{\mathrm{attn}}}(\mathbf{H})\right]_i = \mathbf{h}_i + \sum_{m=1}^M \frac{1}{N} \sum_{j=1}^N \sigma\left(\left(\mathbf{Q}_m\mathbf{h}_i\right)^\top \left(\mathbf{K}_m\mathbf{h}_j\right)\right)\mathbf{V}_m\mathbf{h}_j.$$

*Here $\sigma(x) = x \vee 0$ denotes the ReLU function.*

The MLP layer is the same as Definition 5. The ReLU-activated transformer is defined as follows.

**Definition 8** (ReLU-activated transformer). *An $L$-layer transformer, denoted as $\mathrm{TF}^R_{\boldsymbol{\Theta}}(\cdot)$, is a composition of $L$ ReLU-attention layers each followed by an MLP layer:*

$$\mathrm{TF}^R_{\boldsymbol{\Theta}}(\mathbf{H}) = \mathrm{MLP}_{\boldsymbol{\Theta}^{(L)}_{\mathrm{mlp}}}\left(\mathrm{Attn}^R_{\boldsymbol{\Theta}^{(L)}_{\mathrm{attn}}}\left(\cdots \mathrm{MLP}_{\boldsymbol{\Theta}^{(1)}_{\mathrm{mlp}}}\left(\mathrm{Attn}^R_{\boldsymbol{\Theta}^{(1)}_{\mathrm{attn}}}(\mathbf{H})\right)\right)\right).$$

*Above, the parameter $\boldsymbol{\Theta} = (\boldsymbol{\Theta}^{(1:L)}_{\mathrm{attn}}, \boldsymbol{\Theta}^{(1:L)}_{\mathrm{mlp}})$ consists of the attention layers $\boldsymbol{\Theta}^{(\ell)}_{\mathrm{attn}} = \{(\mathbf{V}^{(\ell)}_m, \mathbf{Q}^{(\ell)}_m, \mathbf{K}^{(\ell)}_m)\}_{m\in[M^{(\ell)}]} \subset \mathbb{R}^{D\times D}$ and the MLP layers $\boldsymbol{\Theta}^{(\ell)}_{\mathrm{mlp}} = (\mathbf{W}^{(\ell)}_1, \mathbf{W}^{(\ell)}_2) \in \mathbb{R}^{D^{(\ell)}\times D} \times \mathbb{R}^{D\times D^{(\ell)}}.*

Similar to Section 4.1, We consider the following function class of transformer.

$$\mathcal{F} := \mathcal{F}(L, D, D', M, B_{\boldsymbol{\Theta}}) = \left\{\mathrm{TF}^R_{\boldsymbol{\Theta}}, \|\boldsymbol{\Theta}\| \leq B_{\boldsymbol{\Theta}}, D^{(\ell)} \leq D', M^\ell \leq M, \ell \in [L]\right\}.$$

Now we can give the formal statement of Theorem 2.

**Theorem G.8** (Formal version of Theorem 2). *There exists a transformer $\mathrm{TF}_{\boldsymbol{\Theta}}$ with ReLU activation such that for any $d \leq d_0$, $T \in \mathbb{R}^{d\times d\times d}$ and $v^{(0)} \in \mathbb{R}^d$, given the encoding (24), $\mathrm{TF}_{\boldsymbol{\Theta}}$ implements $L$ steps of (3) exactly. Moreover, $\mathrm{TF}_{\boldsymbol{\Theta}}$ falls within the class $\mathcal{F}$ with parameters satisfying:*

$$D = D' = O(d_0^2), M = O(d_0), \log B_{\boldsymbol{\Theta}} \leq O(1).$$

**Remark G.1.** *In fact, Theorem G.8 is also hold for attention-only transformers since the MLP layer do not use in the proof. To do that, we only need to add another head in every odd attention layer to clean the terms $\{d\overline{v}_i\}$. For details, see the proof.*

**Remark G.2.** *Readers might question why the normalization step is omitted in our theorem. The key challenge is that we have absolutely no knowledge of a lower bound for $\left\|T\left(I, v^{(j)}, v^{(j)}\right)\right\|$. Without this bound, approximating the normalization step becomes infeasible.*

**Remark G.3.** *The use of the ReLU activation function here is primarily for technical convenience and does not alter the fundamental nature of the attention mechanism. Several studies have demonstrated that transformers with ReLU-based attention perform comparably to those using softmax attention(Shen et al., 2023; Bai et al., 2023; He et al., 2025a).*

## G.2 Proof of Theorem G.8

*Proof.* For simplicity, we only proof the case that $\sigma(x) = x$ in the attention layer. For ReLU activated transformer, the result can be similarly proved by $\mathrm{ReLU}(x) - \mathrm{ReLU}(-x) = x$ and the $\sigma(x) = x$ case. Hence we omit the notation $\sigma$ in the following proof. We take $\mathbf{H}^{(0)} = \mathbf{H}$. In the first attention layer, consider the following attention structures:

$$\mathbf{Q}^{(1)}\mathbf{h}_i^{(0)} = \begin{bmatrix} \mathbf{e}_i \\ \mathbf{0} \end{bmatrix}, \ \mathbf{K}^{(1)}\mathbf{h}_j^{(0)} = \begin{bmatrix} \overline{v}^{(0)} \\ \mathbf{0} \end{bmatrix}, \ \mathbf{V}^{(1)}\mathbf{h}_j^{(0)} = \begin{bmatrix} \mathbf{0}_{d_0^2} \\ \mathbf{0}_{d_0} \\ 0 \\ 0 \\ d \\ \mathbf{0} \end{bmatrix}.$$

After the attention operation, we have

$$\widetilde{\mathbf{h}}_i^{(1)} = \left[\text{Attn}_{\boldsymbol{\Theta}_{\text{attn}}^{(1)}}^R \left(\mathbf{H}^0\right)\right]_{:,i} = \mathbf{h}_i^{(0)} + \frac{1}{d} \sum_{j=1}^d \left(\left(\mathbf{Q}^{(1)}\mathbf{h}_i^0\right)^\top \left(\mathbf{K}^{(1)}\mathbf{h}_j^0\right)\right) \mathbf{V}^{(1)}\mathbf{h}_j^0$$

$$= \mathbf{h}_i^{(0)} + \begin{bmatrix} \mathbf{0}_{d_0^2} \\ \mathbf{0}_{d_0} \\ 0 \\ 0 \\ d\overline{v}_i^{(0)} \\ \mathbf{0} \end{bmatrix} = \begin{bmatrix} \overline{\mathbf{t}}_i \\ \overline{v}^{(0)} \\ 1 \\ d \\ d\overline{v}_i^{(0)} \\ \mathbf{0} \end{bmatrix}, \ i \in [d].$$

Then we use a two-layer MLP to implement identity operation, which is

$$\mathbf{h}_i^{(1)} = \text{MLP}_{\boldsymbol{\Theta}_{\text{mlp}}^{(1)}}\left(\widetilde{\mathbf{h}}_i^{(1)}\right) = \begin{bmatrix} \overline{\mathbf{t}}_i \\ \overline{v}^{(0)} \\ 1 \\ d \\ d\overline{v}_i^{(0)} \\ \mathbf{0} \end{bmatrix}, \ i \in [d].$$

Now we use an attention layer with $d_0 + 1$ heads to implement the power iteration step of the cubic tensor. Consider the following attention structure:

$$\mathbf{Q}_m^{(2)}\mathbf{h}_i^{(1)} = \begin{bmatrix} \overline{v}_m^{(0)} \\ \mathbf{0} \end{bmatrix}, \ \mathbf{K}_m^{(2)}\mathbf{h}_j^{(1)} = \begin{bmatrix} d\overline{v}_j^{(0)} \\ \mathbf{0} \end{bmatrix}, \ \mathbf{V}_m^{(2)}\mathbf{h}_j^{(1)} = \begin{bmatrix} \mathbf{0}_{d_0^2} \\ \overline{T}_{:,j,m} \\ \mathbf{0} \end{bmatrix}, \ m \in [d_0],$$

and

$$\mathbf{Q}_{d_0+1}^{(2)}\mathbf{h}_i^{(1)} = \begin{bmatrix} 1 \\ \mathbf{0} \end{bmatrix}, \ \mathbf{K}_{d_0+1}^{(2)}\mathbf{h}_j^{(1)} = \begin{bmatrix} d \\ \mathbf{0} \end{bmatrix}, \ \mathbf{V}_{d_0+1}^{(2)}\mathbf{h}_j^{(1)} = \begin{bmatrix} \mathbf{0}_{d_0^2} \\ -\overline{v}^{(0)} \\ \mathbf{0} \end{bmatrix}.$$

After the attention operation, we have

$$\widetilde{\mathbf{h}}_i^{(2)} = \left[\text{Attn}_{\boldsymbol{\Theta}_{\text{attn}}^{(2)}}^R \left(\mathbf{H}^{(1)}\right)\right]_{:,i}$$

$$= \mathbf{h}_i^{(1)} + \sum_{m=1}^{d_0} \frac{1}{d} \sum_{j=1}^d \left(\left(\mathbf{Q}_m^{(2)}\mathbf{h}_i^{(1)}\right)^\top \left(\mathbf{K}_m^{(2)}\mathbf{h}_j^{(1)}\right)\right) \mathbf{V}_m^{(2)}\mathbf{h}_j^{(1)}$$

$$+ \frac{1}{d} \sum_{j=1}^d \left(\left(\mathbf{Q}_{d_0+1}^{(2)}\mathbf{h}_i^{(1)}\right)^\top \left(\mathbf{K}_{d_0+1}^{(2)}\mathbf{h}_j^{(1)}\right)\right) \mathbf{V}_{d_0+1}^{(2)}\mathbf{h}_j^{(1)}$$

$$= \mathbf{h}_i^{(1)} + \sum_{m=1}^{d_0} \frac{1}{d} \sum_{j=1}^d \left(d\overline{v}_m^{(0)}\overline{v}_j^{(0)}\right) \begin{bmatrix} \mathbf{0}_{d_0^2} \\ \overline{T}_{:,j,m} \\ \mathbf{0} \end{bmatrix} + \frac{1}{d} \sum_{j=1}^d d \begin{bmatrix} \mathbf{0}_{d_0^2} \\ -\overline{v}^{(0)} \\ \mathbf{0} \end{bmatrix}$$

$$= \mathbf{h}_i^{(1)} + \begin{bmatrix} \mathbf{0}_{d_0^2} \\ \overline{v}^{(1)} \\ \mathbf{0} \end{bmatrix} + \begin{bmatrix} \mathbf{0}_{d_0^2} \\ -\overline{v}^{(0)} \\ \mathbf{0} \end{bmatrix}$$

$$= \begin{bmatrix} \overline{\mathbf{t}}_i \\ \overline{v}^{(1)} \\ 1 \\ d \\ d\overline{v}_i^{(0)} \\ \mathbf{0} \end{bmatrix}, \ i \in [d],$$

where $\overline{v}^{(1)} = \left[v^{(1)\top}, \mathbf{0}_{d_0-d}^\top\right]^\top$ and $v^{(1)} = \sum_{j,m\in[d]} v_m^{(0)} v_j^{(0)} T_{:,j,m}$.

Then we use a two-layer MLP to clean the term $d\overline{v}_i^{(0)}$, which is

$$\mathbf{h}_i^{(2)} = \text{MLP}_{\mathbf{\Theta}_{\text{mlp}}^{(2)}}\left(\widetilde{\mathbf{h}}_i^{(2)}\right) = \begin{bmatrix} \overline{\mathbf{t}}_i \\ \overline{v}^{(1)} \\ 1 \\ d \\ \mathbf{0} \end{bmatrix}, \ i \in [d].$$

Similarly, for any $\ell \in \mathbb{N}_+$, we have

$$\mathbf{h}_i^{(2\ell)} = \left[\text{MLP}_{\mathbf{\Theta}_{\text{mlp}}^{(2)}}\left(\text{Attn}_{\mathbf{\Theta}_{\text{attn}}^{(2)}}\left(\text{MLP}_{\mathbf{\Theta}_{\text{mlp}}^{(1)}}\left(\text{Attn}_{\mathbf{\Theta}_{\text{attn}}^{(1)}}\left(\mathbf{H}^{(2\ell-2)}\right)\right)\right)\right)\right]_{:,i} = \begin{bmatrix} \overline{\mathbf{t}}_i \\ \overline{v}^{(\ell)} \\ 1 \\ d \\ \mathbf{0} \end{bmatrix}, \ i \in [d].$$

The parameter bounds can be directly computed by the construction above. The theorem is proved.

$\square$

## H    MORE ON EMPIRICAL STUDIES

### H.1    MORE ON EXPERIMENTAL SETUPS

**Anisotropic adjustments**    We consider anisotropic Gaussian mixtures that takes the following form: A $K$-component anisotropic Gaussian mixture distribution is defined with parameters $\boldsymbol{\theta} = \boldsymbol{\pi} \cup \boldsymbol{\mu} \cup \boldsymbol{\sigma}$, where $\boldsymbol{\pi} := \{\pi_1^*, \pi_2^*, \cdots, \pi_K^*\}$, $\pi_k^* \in \mathbb{R}$, $\boldsymbol{\mu} = \{\mu_1^*, \mu_2^*, \cdots, \mu_K^*\}$, $\mu_k^* \in \mathbb{R}^d$, $k \in [K]$ and $\boldsymbol{\sigma} = \{\sigma_1^*, \sigma_2^*, \cdots, \sigma_K^*\}$, $\sigma_k^* \in \mathbb{R}_+^d$, $k \in [K]$. A sample $X_i$ from the aforementioned anisotropic GMM is expressed as:

$$X_i = \mu_{y_i}^* + \sigma_{y_i}^* Z_i, \tag{25}$$

where $\{y_i\}_{i\in[N]}$ are iid discrete random variables with $\mathbb{P}(y = k) = \pi_k^*$ for $k \in [K]$ and $\{Z_i\}_{i\in[N]}$ are iid standard Gaussian random vector in $\mathbb{R}^d$. Analogous to that in the isotropic case and overload some notations, we define an anisotropic GMM task to be $\mathcal{T} = (\mathbf{X}, \boldsymbol{\theta}, K)$.

To adapt the TGMM framework to be compatible to anisotropic problems, we expand the output dimension of the attentive pooling module from $(d + K) \times K$ to $(d + 2K) \times K$, with the additional $K$ rows reserved for the estimate $\widehat{\boldsymbol{\sigma}}$ of $\boldsymbol{\sigma}$, with the corresponding estimation loss function augmented with a scale estimation part:

$$\widehat{L}_n(\mathbf{\Theta}) = \frac{1}{n}\sum_{i=1}^n \ell_\mu(\widehat{\boldsymbol{\mu}}_i, \boldsymbol{\mu}_i) + \ell_\pi(\widehat{\boldsymbol{\pi}}_i, \boldsymbol{\pi}_i) + \ell_\sigma(\widehat{\boldsymbol{\sigma}}_i, \boldsymbol{\sigma}_i), \tag{26}$$

where the loss function $\ell_\sigma$ is chosen as the mean-square loss. During the experiments, we inherit configurations from those of isotropic counterparts, except for the calculation of the $\ell_2$-error metric, where we additionally considered contributions from the estimation error of scales.

**Configurations related to Mamba2 architecture**    We adopt a Mamba2 Dao & Gu (2024) model comprising 12-layers and 128-dimensional hidden states, with the rest hyper-parameters chosen so as to approximately match the number of a 12-layer transformer with 128-dimensional hidden states. As the Mamba series of models are essentially recurrent neural networks (RNNs), we tested two different kinds of Readout design with either (i). the attentive pooling module as used in the case of transformer backbone and (ii). a more natural choice of using simply the last hidden state to decode all the estimates, as RNNs compress input information in an ordered fashion. We observe from our empirical investigations that using attentive pooling yields better performance even with a Mamba2 backbone. The other training configurations are cloned from those in TGMM experiments with transformer backbones.

**Software and hardware infrastructures**    Our framework is built upon PyTorch Paszke et al. (2019) and `transformers` Wolf et al. (2020) libraries, which are open-source software released under BSD-style [2] and Apache license [3]. The code implementations will be open-sourced after the reviewing process of this paper. All the experiments are conducted using 8 NVIDIA A100 GPUs with 80 GB memory each.

## H.2    A complete report regarding different evaluation metrics

In this section, we present complete reports of empirical performance regarding the evaluation problems mentioned in section 3. Aside from the $\ell_2$-error metric that was reported in section 3.2, we additionally calculated all the experimental performance under the following metrics:

**Clustering accuracy**    We compare estimated cluster membership with the true component assignment, after adjusting for permutation invariance as mentioned in section 2.3.

**Log-likelihood**    We compute average log-likelihood as a standard metric in unsupervised statistical estimation.

The results are reported in figure 10, 11, 12, 13 and 14, respectively. According to the evaluations, the learned TGMM models show comparable clustering accuracy against the spectral algorithm and outperform EM algorithm when $K > 2$ across all comparisons. Regarding the log-likelihood metric, TGMM demonstrates comparable performance with the other two classical algorithms in comparatively lower dimensional cases. i.e., $d \in \{2, 8\}$, but underperforms both baselines in larger dimensional problems. We conjecture that is might be due to the fact that EM algorithm is essentially a maximum-likelihood algorithm Dempster et al. (1977), while the TGMM estimation objective (2) is not explicitly related to likelihood-based training.

## H.3    On the impact of inference-time sample size $N$

Motivated by the classical statistical phenomenon that estimation quality tends to improve with sample size, we test whether TGMM's estimation performance increases as $N$ goes up. We run corresponding experiments by varying the sample size to be $N \in \{32, 64, 128\}$ during both train and inference, while controlling other experimental configurations same as those in section 3.1. The results are reported in $\ell_2$-error, clustering accuracy as log-likelihood and summarized in figure 15. The results exhibit a clear trend that aligns with our hypothesis, justifying the TGMM learning process as learning a statistically meaningful algorithm for solving GMMs.

## H.4    On the impact of backbone scale

The scaling phenomenon is among the mostly discussed topics in modern AI, as choosing a suitable scale is often critical to the performance of transformer-based architectures like LLMs. In this section we investigate the scaling properties of TGMM via comparing performances produced by varying sizes of backbones that differ either in per-layer width (i.e., the dimension of attention embeddings) or in the total number of layers $L$. With the rest hyper-parameters controlled to be the same as those in section 3.1. The results are reported in three metrics and summarized in figure 16 and figure 17, respectively. According to these investigations, while in general a larger-sized backbone yields slightly better performance as compared to smaller ones. The performance gaps remain mild especially for tasks with relative lower complexity, i.e., $K = 2$. Consequently, even a 3-layer transformer backbone is able to achieve non-trivial learning performance for solving isotropic GMMs, a phenomenon that was also observed in a recent work He et al. (2025b).

---

[2]https://github.com/pytorch/pytorch/blob/master/LICENSE
[3]https://github.com/huggingface/transformers/blob/main/LICENSE

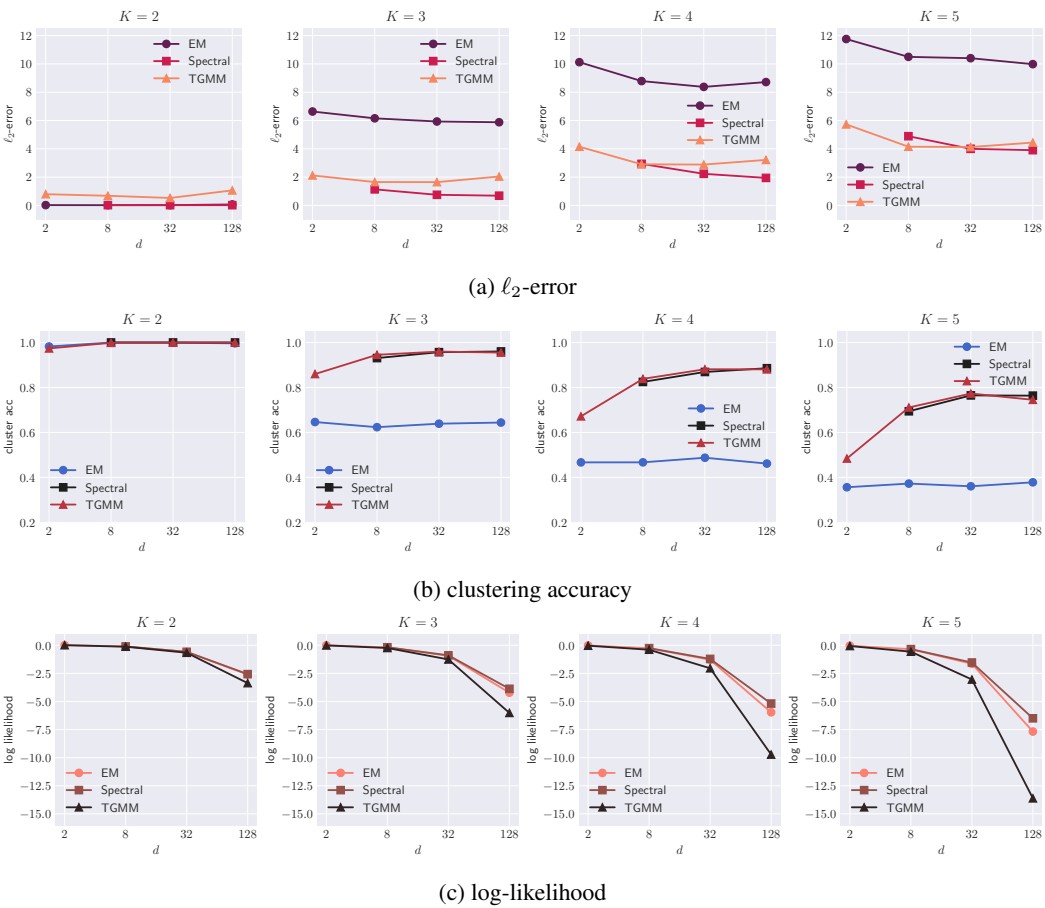

(a) $\ell_2$-error

(b) clustering accuracy

(c) log-likelihood

Figure 10: Performance comparison between TGMM and two classical algorithms, reported in three metrics.

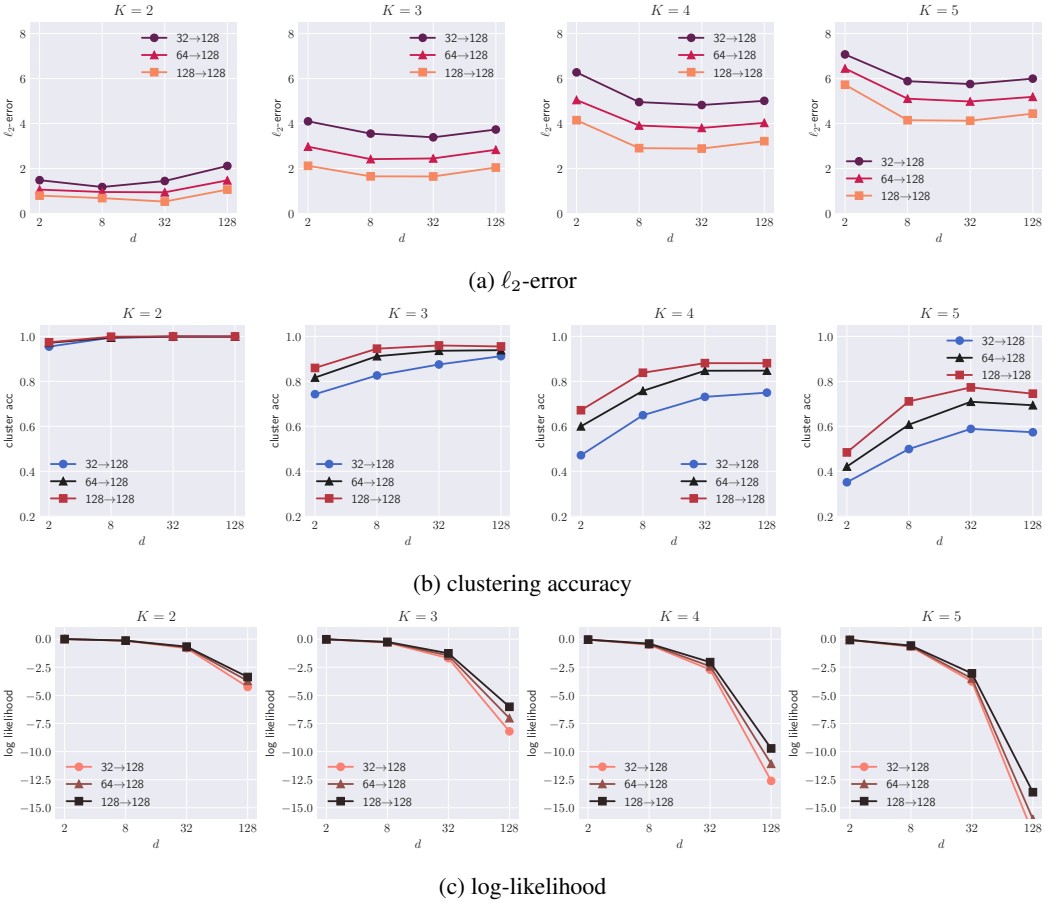

(a) $\ell_2$-error

(b) clustering accuracy

(c) log-likelihood

Figure 11: Assessments of TGMM under test-time task distribution shifts I: A line with $N_0^{\text{train}} \to N^{\text{test}}$ draws the performance of a TGMM model trained over tasks with sample size randomly sampled in $[N_0^{\text{train}}/2, N_0^{\text{train}}]$ and evaluated over tasks with sample size $N^{\text{test}}$. We can view the configuration $128 \to 128$ as an in-distribution test and rest as out-of-distribution tests.

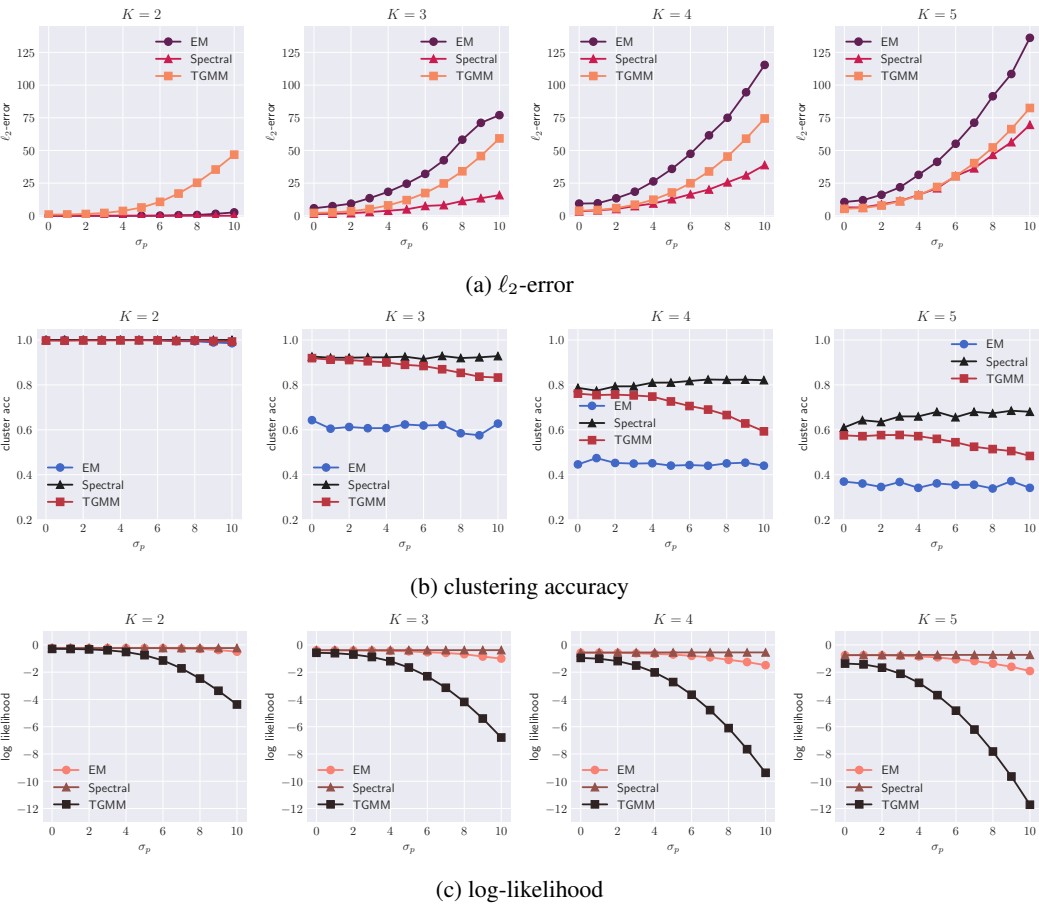

Figure 12: Assessments of TGMM under test-time task distribution shifts II: $\ell_2$-error of estimation when the test-time tasks $\mathcal{T}^{\text{test}}$ are sampled using a mean vector sampling distribution $p_\mu^{\text{test}}$ different from the one used during training.

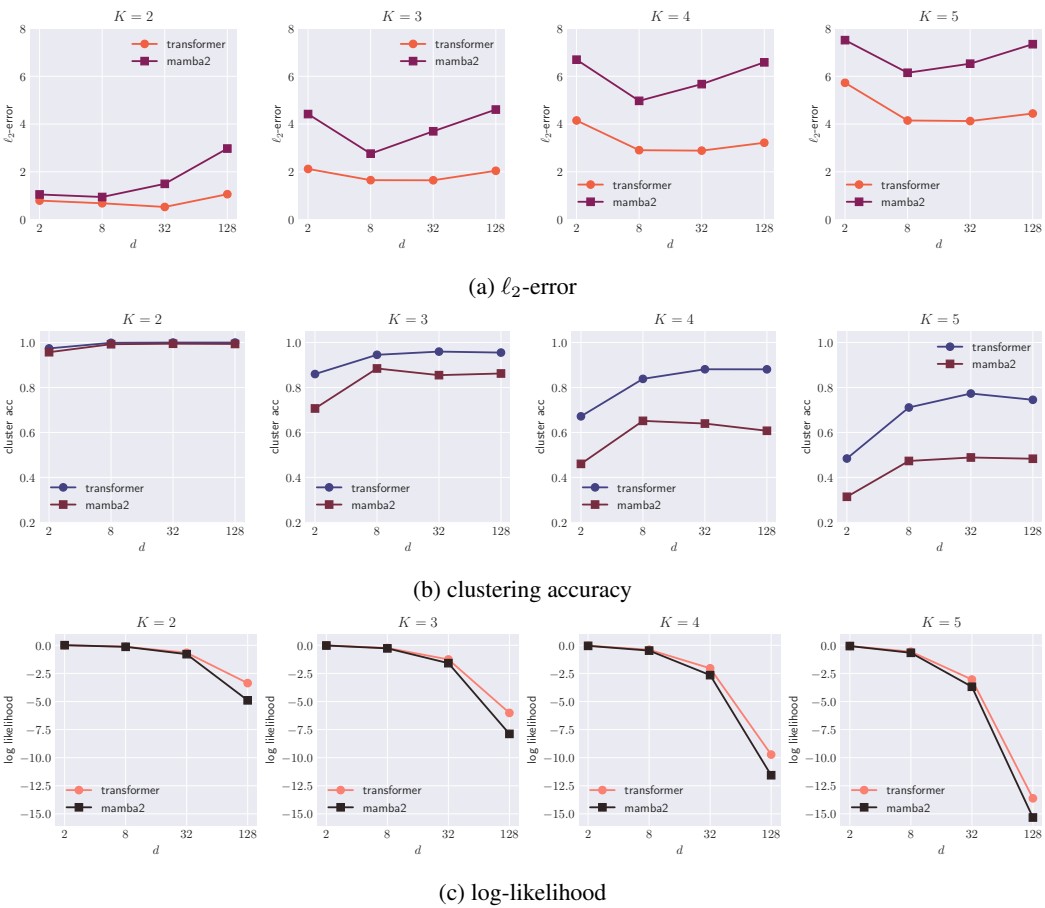

Figure 13: Performance comparisons between TGMM using transformer and Mamba2 as backbone, reported in three metrics.

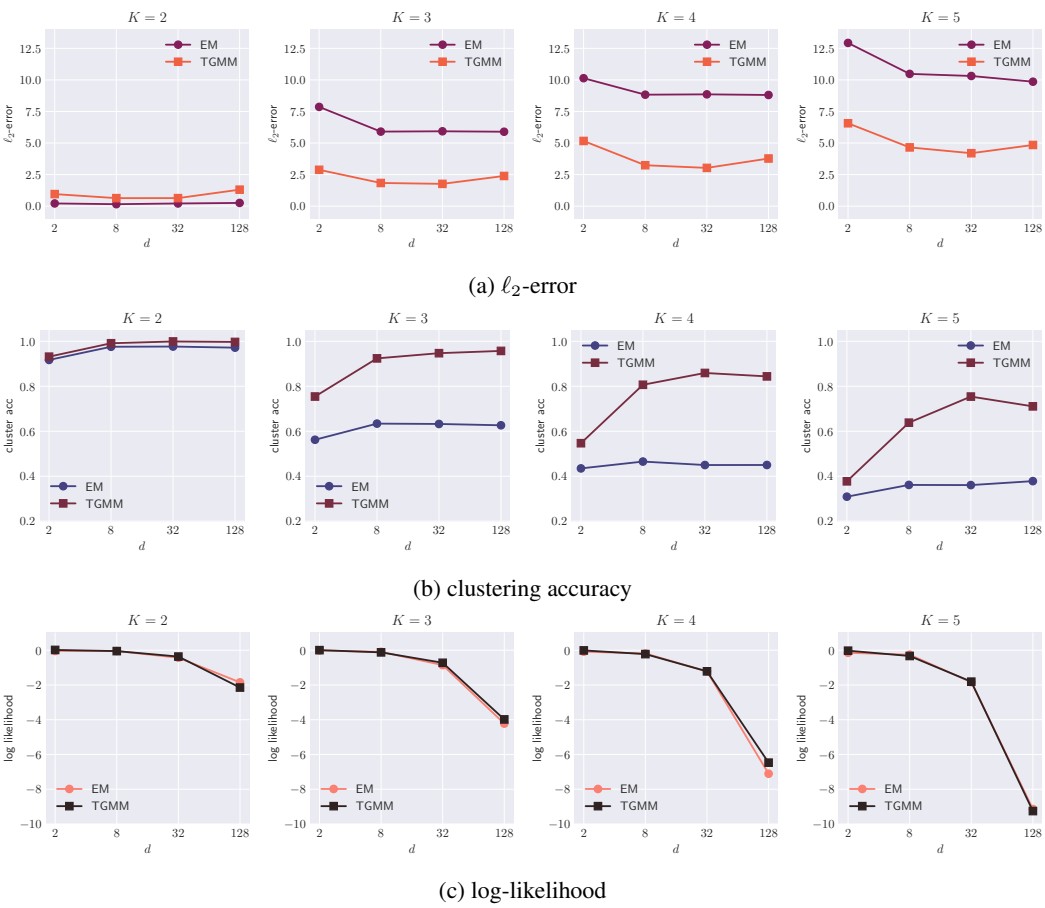

(a) $\ell_2$-error

(b) clustering accuracy

(c) log-likelihood

Figure 14: Performance comparison between TGMM and the EM algorithm on anisotropic GMM tasks, reported in three metrics

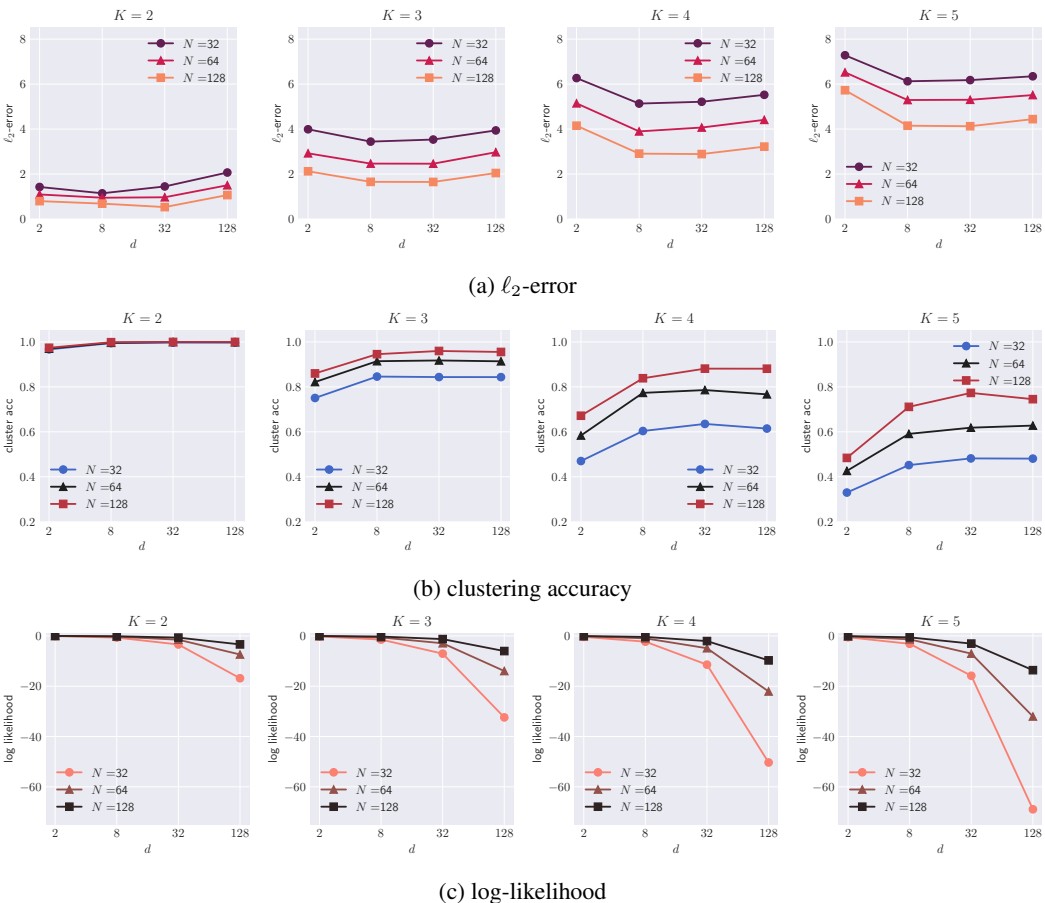

Figure 15: Performance comparison between TGMM models trained under varying configurations of sample-size. For example, $N = 64$ means that the model is trained over GMM tasks with (randomly chosen) sample sizes within the range $[32, 64]$ and tested on tasks with sample size $64$.

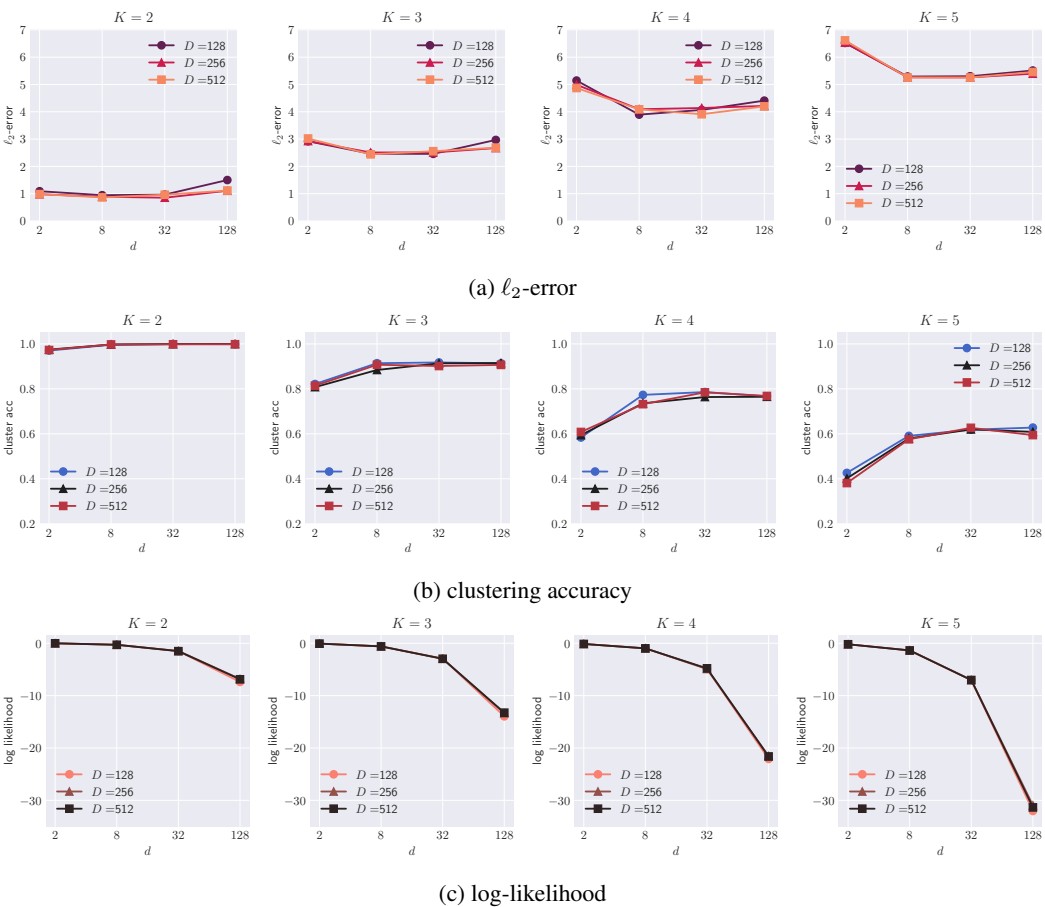

Figure 16: Performance comparison between TGMM under backbones of varying scales I: We fix embedding size at $d = 128$ and tested over different number of transformer layers $L \in \{3, 6, 12\}$. Results are reported in three metrics.

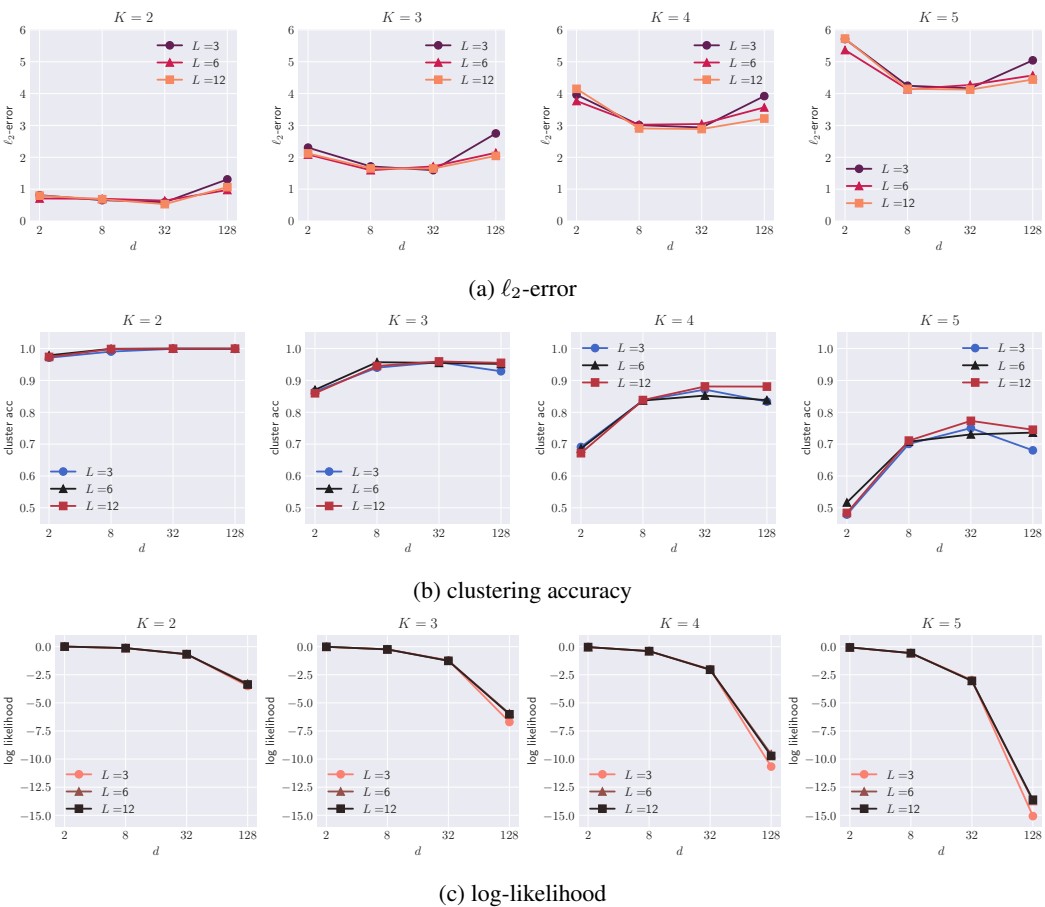

Figure 17: Performance comparison between TGMM under backbones of varying scales II: We fix the number of transformer layers at $L = 12$ and tested over different number of hidden states $d \in \{128, 256, 512\}$. Results are reported in three metrics.

