# OpenReview forum: "Transformers as Unsupervised Learning Algorithms: A study on Gaussian Mixtures"
_ICLR.cc/2026/Conference — ICLR 2026 Poster_

### Official Review · Reviewer_YKHK · 2025-10-28

**Soundness:** 3
**Presentation:** 2
**Contribution:** 2
**Rating:** 2
**Confidence:** 4

**Summary:**

The paper studies whether transformers can simulate unsupervised learning algorithms with a focus on Gaussian mixtures. The paper provides both empirical and theoretical evidence showing that there exists a transformer that approximates the EM algorithm well.
Next, they propose a learning framework named TGMM for transformers to learn Gaussian mixtures.

**Strengths:**

- The figures are well presented
- The empirical evidence well demonstrated the proposed TGMM framework is effective.

**Weaknesses:**

- The theoretical result seems to be weaker than [1], where [1] shows the softmax version and this paper mainly focuses on ReLU transformers.
- I think the presentation of this paper can be further improved. To me, this paper is more of a theoretical focus than empirical based on its settings. However, most of the main text are used to describe empirical results.
- The proposed TGMM framework seems to be trivial. Its setup is very similar to the pretraining setup in (Bai et al.2023), where it samples tasks from a meta distribution $\pi$.
- It is unclear that how the proposed TGMM framework benefits training from a theoretical perspective.



[1] Transformers versus the EM Algorithm in Multi-class Clustering (He et al, 2025)

**Questions:**

Please refer to weaknesses.

---

> ### Author Response · Authors · 2025-11-16
> **Response to Reviewer YKHK (part 1/3)**
>
> Thanks for your review. We think that **there might be some misunderstandings** about our paper and hope that our responses can address all your concerns.
> The following are our detailed explanations.
>
> >The theoretical result seems to be weaker than [1], where [1] shows the softmax version and this paper mainly focuses on ReLU transformers.
>
> ### Q1: On theoretical contributions of our paper and comparisons with [1]
>
> Our main theoretical contributions, specifically, the approximation of the EM algorithm (Theorem 1, Section C, and Theorem F.1), are established for the standard Transformer architecture with **softmax** attention, which is the most commonly used variant in practice. As we wrote in the introduction:
> > Our approximation of the EM algorithm fundamentally leverages the weighted averaging property inherent in softmax attention, enabling simultaneous approximation of both the E and M steps. Notably, our approximation results also hold across varying dimensions and mixture components in GMM.
>
> The ReLU-based Transformer is employed only in the analysis of the spectral algorithm approximation, solely for technical convenience; this does not affect the core claims or contributions of our work.
>
> On the other hand, prior work [1] focuses exclusively on approximating the EM algorithm for clustering. Crucially, both our work and [1] analyze Transformers with the same softmax activation, making the comparison fair and meaningful. A more detailed discussion of how our results strengthen and extend those in [1] can be found in the “Sharpness of our results” paragraph in the introduction, where we summarize the key improvements in the following table.
>
> Table 1: Comparison of Our Method and Prior Work [1] in Approximating the EM Algorithm.
> | **Comparison Aspect**                          | **Our Work**                                       | **Prior Work [1]**                                        |
> |------------------------------------------------|----------------------------------------------------|-----------------------------------------------------------|
> | Attention Structure                            | **Softmax** attention                              | **Softmax** attention                                     |
> | Layer Complexity<br>(to approximate L-step EM) | $O(L)$ layers                                    | $O(KL)$ layers (scales with number of components \(K\)) |
> | Attention Heads                                | Valid with $\(M = O(1)\)$                        | Requires $\(M \to +\infty\)$                            |
> | Approximation Bound Scaling                    | **Polynomial** in dimension $d$                  | **Exponential** in $d$                                  |
> | Structural Conditions                          | No restriction on $K$ vs. $d$                  | Requires $K < d$                                    |
> | Task Generality                                | Single model handles **varying $d$ and $K$**   | No cross-task (varying $d, K$) approximation shown      |
>
> **1.Weaker Structure Conditions**: Our analysis shows that Transformers can approximate L-step EM algorithms with just O(L) layers, a significant improvement over prior work [1], which requires O(KL) layers (dependent on the number of components K).
> Moreover, unlike [1], which needs number of attention heads $M \rightarrow +\infty$ to get valid bounds, our results hold with $M = O(1)$, aligning better with real-world designs. Moreover, the approximation results in [1] require an additional technical condition, namely $K<d$, which our results do not impose.
>
> **2.Tighter Bounds:** Our approximation bounds scale polynomially in dimension $d$, unlike [1]’s exponential dependence--a crucial improvement for high-dimensional settings.
>
> **3.Improved model versatility:** [1] does not demonstrate that **a Transformer backbone can simultaneously approximate the EM algorithm across GMM tasks with varying dimensions $d$ and numbers of components $K$**. In contrast, the ability to handle diverse tasks within a single model is a core contribution of our work, as highlighted earlier.
>
> Besides, we notice that in [1] while the theoretical construction requires a fairly large transformer to succeed, the experimental evaluations therein used only 1-layer and 3-layer transformers and do not align well with theory developments. In our paper, we validated both the impact of scale and several other design choices in appendix H, making our empirical observations corroborate with theory.

---

> ### Author Response · Authors · 2025-11-16
> **Response to Reviewer YKHK (part 2/3)**
>
> > I think the presentation of this paper can be further improved. To me, this paper is more of a theoretical focus than empirical based on its settings. However, most of the main text are used to describe empirical results.
>
> ### Q2: Question about paper organization
>
> Thanks for your comment. The meta-learning (in-context learning in some literature) phenomenon and mechanism of transformers were first studied empirically [2] [3], followed by theoretical explanations [4] [5]. The aim of this paper is to explore the expressive power of transformers in unsupervised learning problems, especially GMMs. Following the existing trajectory, we first demonstrate that transformers can effectively solve unsupervised GMM tasks through comprehensive experiments, and then provide a theoretical explanation for why this is possible.
>
> This empirical-first structure not only aligns with the methodological norms of the field but also enhances accessibility for a broad audience with varying levels of theoretical background. As noted by reviewers TJ7f and 9Fgx, this organization improves the paper’s readability without compromising its theoretical rigor. Nevertheless, we apologize to readers with a theoretical background who may find the presentation style or structure of this paper unconventional or less aligned with traditional theoretical expositions.
>
> >-The proposed TGMM framework seems to be trivial. Its setup is very similar to the pretraining setup in (Bai et al.2023, [4]), where it samples tasks from a meta distribution.
>
> ### Q3: The difference between our framework and [4]
>
> **We respectfully disagree that our framework is trivial or equivalent to the pretraining setup in [4]**. While both works operate within the general meta-learning paradigm, which is a necessary framework for any meta-learning approach, the problem settings and architectural designs are fundamentally distinct.
>
> - **Problem settings: Supervised prediction vs Unsupervised estimation** [4] considers supervised regression tasks with explicit labels, allowing for a simple encoder structure (decoding is trivial in that case via simply picking the last hidden state of the final layer ouptut). In contrast, our framework tackles an unsupervised learning problem, specifically, learning GMMs with **no access to labels**.
>
> - **Architecture design: Novel construction of decoder(read-out)** A crucial distinction between the architectural design of TGMM and that in [4] is the role of decoder: As mentioned earlier that the learning target for TGMM is an estimation algorithm with no access to labels, it is both in theory (please refer to the construction in Theorem F.1) and in practice necessary to utilize all the hidden states to construct a meaningful estimation. The attentive-pooling decoder described in line 200-204 of the manuscript is motivated by our theoretical construction, and behaves much better than naive decoding in practice, as reported in the following table, where we compare our attentive-pooling decoder with naive decoding (as the method in [4]) with $K=3$ and $L=12$ (the experimental configurations are identical with those in section 3.1 in our paper):
>
> Table 2: Comparison of attentive-pooling decoder with naive decoding (pick the last hidden state)
> | Metric                      | TGMM(naive-decoding) | TGMM(attentive-decoding) |
> |-----------------------------|----------------------|--------------------------|
> | $\ell_2$-error              | $1.49$               | $2.87$                   |
> | $\text{Acc}_\text{cluster}$ | $98.29$              | $84.20$                  |
>
> As showcased in table 2. The design of attentive-decoding plays an important role in achieving meaningful performance of TGMM. Moreover, these read-in and read-out modules are essential to enabling our TGMM framework to handle GMMs with varying numbers of components (i.e., different $K$ ) within a single backbone, which is a key and distinctive feature of our approach. This flexibility reflects a deeper structural insight into how transformers can generalize across model complexities in GMM settings.

---

> ### Author Response · Authors · 2025-11-30
> **Response to Reviewer YKHK (part 3/3)**
>
> > It is unclear that how the proposed TGMM framework benefits training from a theoretical perspective.
>
> ### Q4: For the training benefits of the TGMM
>
> We agree that studying the impact of the training process on TGMM from a theoretical perspective is highly important. However, to the best of our knowledge, **the training dynamics of multi-layer Transformers remain an open and challenging problem**. We provide a detailed discussion below:
>
> - **Contemporary developments focus on single-layer architectures** Current theoretical analyses—such as those in [6], [7], and [8]—have largely focused on **single-layer architectures**, with [6] further simplifying the setting by assuming linear attention.
> - **Single-layer transformer is insufficient to achieve satisfactory performance** However, **single-layer architectures** are not sufficient for GMM learning, as showcased in the following table, where we train a TGMM on $K=3$ GMM tasks with $d=32$ under training-time GMM sample being $N_\text{train} = 128$ and evaluate it under various lengths of testing samples:
>
> Table 3: Dependency of TGMM performance over depth
> |            | $\text{Acc}_\text{cluster}$ |             |             | $\ell_2$-error |             |             |
> |------------|-------------|-------------|-------------|----------------|-------------|-------------|
> |            | $L=1$       | $L=6$       | $L=12$      | $L=1$          | $L=6$       | $L=12$      |
> | $N_\text{test}=128$    | $85.76$     | $89.84$     | $98.29$     | $3.61$         | $2.68$      | $1.49$      |
> | $N_\text{test}=256$    | $85.13$     | $94.41$     | $98.50$     | $3.80$         | $2.30$      | $1.50$      |
> | $N_\text{test}=512$    | $85.43$     | $95.48$     | $98.67$     | $3.71$         | $2.09$      | $1.47$      |
> | $N_\text{test}=1024$   | $84.82$     | $97.46$     | $99.10$     | $3.69$         | $2.14$      | $1.45$      |
> | $N_\text{test}=2048$   | $85.71$     | $96.36$     | $99.61$     | $3.63$         | $2.11$      | $1.40$      |
>
> Table 1 demonstrates that TGMM with $L=1$ cannot achieve good performance both in terms of estimation and in terms of clustering, showing that current single-layer theories, which focus on simpler cases like in-context linear regression, are yet insufficient to explain the training dynamics of TGMM. **Therefore, we think that explaining the training dynamics of TGMM is beyond the scope of our paper**.
>
> We believe our theoretical contributions on the approximation capability of Transformers provide an important foundation by demonstrating what these models can express in principle. Understanding how they *learn* such solutions during training, particularly in the context of GMMs, remains a challenging and exciting direction. We are eager to explore this question in future work.
>
> [1] He et al. Transformers versus the EM algorithm in multi-class clustering. arXiv preprint arXiv:2502.06007(2025).
>
> [2] Garg et al. What Can Transformers Learn In-Context? A Case Study of Simple Function Classes. NeurlIPS, 2022.
>
> [3] Oswald et al. Transformers Learn In-Context by Gradient Descent. ICML, 2023.
>
> [4] Bai et al. Transformers as statisticians: Provable in-context learning with in-context algorithm selection. NeurlIPS, 2023.
>
> [5] Lin et al. Transformers as decision makers: Provable in-context reinforcement learning via supervised pretraining. ICLR, 2024.
>
> [6] Zhang et al. Trained transformers learn linear model in-context. JMLR, 2024.
>
> [7] Chen et al. Training dynamics of multi-head softmax attention for in-context learning: Emergence, convergence, and optimality. COLT, 2024.
>
> [8] Chen et al. Provably learning a multi-head attention layer. STOC, 2025.

---

### Official Review · Reviewer_9Fgx · 2025-10-29

**Soundness:** 4
**Presentation:** 3
**Contribution:** 4
**Rating:** 8
**Confidence:** 3

**Summary:**

The classical Gaussian mixture model problem (GMM) states as follows.
Suppose we have a set of $N$ samples, $X_1,\dots,X_N$, drawn from a mixture of $K$ (isotropic) Gaussians $\sum_{i=1}^K \pi_i \phi(\cdot\;; \mu_i)$ where $\phi$ is the $d$-dimensional Gaussian with covariance being the identity matrix, $\pi_i > 0$, $\sum_{i=1}^K \pi_i = 1$ and $\mu_i\in \mathbb{R}^d$ for all $i = 1,\dots, K$.
Out task is to design an algorithm that takes $X_1,\dots,X_N$ as input and returns $\hat \pi_1, \dots, \hat \pi_K, \hat \mu_1,\dots, \hat\mu_K$ such that there exists a permutation $\sigma$ on $\{1,\dots, K\}$ that $\hat \pi_{\sigma(i)}$ (resp. $\hat \mu_{\sigma(i)}$) is close to $\pi_i$ (resp. $\mu_i$) for all $i = 1,\dots, K$.

This paper studies the approach of using transformers to solve GMM.
More precisely, the authors propose an architecture that first encodes the samples $X_1,\dots,X_N$, parses them into a multi-layer transformer and then decodes the output.
The authors use the $\ell_2$-error to measure the quality of the output, i.e. $\frac{1}{K}\sum_{k=1}^K (\frac{1}{d}\lVert \hat \mu_{\sigma(i)} - \mu_i\rVert^2 + (\hat\pi_{\sigma(i)} - \pi_i)^2)$.
The authors show that the aforementioned architecture can approximate the Expectation-Maximization (EM) algorithm and the spectral algorithms which are the classical algorithms for solving GMM.

**Strengths:**

- The paper provides an interesting connection between the modern approach and the classical approach in machine learning.
I believe this connection can open up a new pathway to explaining the fundamental structures of different models.

- The presentation is clear.
Readers of different levels of expertise should be able to grasp the central idea of this paper.

**Weaknesses:**

- The paper is generally good.

**Questions:**

- Abstract: It may be helpful to spell out the full name of TGMM for the first time.

- Line 154: ``... i.i.d. ...''

---

> ### Author Response · Authors · 2025-11-16
> **Response to Reviewer 9Fgx**
>
> Thank you for your positive and thoughtful feedback-we truly appreciate the time and care you invested in reviewing our work. Your positive and constructive perspective has been invaluable in helping us strengthen the paper.  We also believe that your thoughtful, supportive evaluations contribute meaningfully to a more collaborative and respectful academic culture.
>
> > Abstract: It may be helpful to spell out the full name of TGMM for the first time.
>
> > Line 154: ``... i.i.d. ...''
>
>
> We are grateful for your detailed comments and will carefully address these minor points in the revised version.

---

### Official Review · Reviewer_TJ7f · 2025-10-31

**Soundness:** 3
**Presentation:** 3
**Contribution:** 3
**Rating:** 6
**Confidence:** 3

**Summary:**

The paper considers the use of transformers to solving Gaussian Mixture Models (GMMs) by learning to directly map from the input samples to the associated GMM parameters considering a simple GMM with identity matrix as covariance whereas mean and cluster proportions are learned. The resulting TGMM is trained to solve multiple GMM tasks using the same transformer backbone. The work is related especially to He et al. (2025b) but the current work focuses on estimating the parameters of the GMM as opposed to the cluster labels. It is demonstrated that L conventional EM steps can be approximated using \mathcal{O}(L) transformer layers with bounds relying by a constant on the attention heads with polynomial scaling in dimension, consequently improving upon the scaling laws of the work of He et al. It is further discussed how power iterations in spectral methods can similarly be approximated. Notably, the proposed TGMM can also learn GMMs with different number of clusters as the transformer structure used an encoder that takes as input an embedding representation of K concatenated the inputs and a decoder structure that is specific to the number of components to be produced in the output. The TGMM is demonstrated to perform well when compared to EM and spectral GMM estimation whereas a Mamba2 backbone is further considered and found to have inferior performance to the transformer backbone.

**Strengths:**

•	The approach is sound and improves upon the related work especially of He et al 2025b.

•	The results are compelling and the procedure appears promising when compared to EM and spectral estimation of GMMs.

•	The design to accommodate different number of components seems logical and useful.

Originality: The approach expands upon current efforts utilizing Transformers in the context of unsupervised learning in GMM especially leveraging the recent works of He et al 2025a,b with substantial improvements as outlined in the summary. Theoretical properties wrt. the Transformer surrogating EM and spectral inference procedures is further investigated. The contribution in all provides an overall sufficient and interesting contribution.

Clarity: The paper is very clearly written and well presented.

Quality: The paper well presents the developed methodology as well as theoretical insights and the experimentation is generally well executed but limited to synthetic data.

Significance: The approach expands upon previous recent work and make significant contributions including better scaling properties and insights to Transformers capabilities mimicking unsupervised learning procedures in the context of EM and Spectral based inference for the GMM. The experimentations are solid but limited to synthetic data whereas the GMM assuming isotropic variance makes the practical use very limited reducing the impact of the work.

**Weaknesses:**

The considered GMM formalism is quite limited and it would strengthen the contribution to consider more realistic settings such as GMMs with diagonal covariance clusters. It is unclear why this would form a major challenge in the presented framework as it technically just requires expanding the readout function to have parameters for the variances of similar dimensionality as the produced means in the readout function and loss functions based on an additional squared error loss term for the diagonal variances.

The analysis are purely considering synthetic data and it is unclear why the approach is not also tested on real data. This would substantially expand and verify the practical utility of the developed framework. Currently, the limited variance formulation makes it likely impractical for real data analyses.

The presented performance metric is in terms of least squares error to ground truth cluster centers as well as mixing proportions. I appreciate the L2 metric in the main paper but think the log-likelihood metric considered in the supplementary should also be part of the main-paper as the EM-based procedure is based on optimizing the log-likelihood whereas the TGMM is minimizing the LS error of the parameters. This will provide a more comprehensive evaluation in the main paper in terms of recovering the underlying parameters and how well the model accounts for the underlying density. Furthermore, it would be interesting to display the mean and mixture proportion errors separately at least in the supplementary to understand if there are discrepancies in these errors. It would also be natural to quantify the mixture proportion errors using cross-entropy as optimized.

**Questions:**

The loss function optimized in equation (2) contains an implicit weighting between accuracy of mean estimates by least squares and accuracy of cluster sizes by cross-entropy loss. How should these losses be balanced in general? And why not use the standard log-likelihood as loss?

EM and spectral methods are widely used, but it would be relevant also to compare against a simple SGD procedure based on Adam or similar optimization minimizing the GMM likelihood for comparison. How would such simple procedure compare?

What is the reason for not including a diagonal variance in the modeling which would make the approach more useful and seems to be straight forward to implement?

What is the practical relevance of the procedure – it would strengthen the paper to consider real datasets and the quality of the learned clusters. This would however likely also require expanding the modeling procedure to at least have cluster centers with diagonal variance parameters. What is hampering the procedure to be expanded to such setting and evaluated on real data?

---

> ### Author Response · Authors · 2025-11-16
> **Response to Reviewer TJ7f (part 1/3)**
>
> We thank the reviewer for providing positive feedback and insightful comments. Below we address your specific points:
>
> > The considered GMM formalism is quite limited and it would strengthen the contribution to consider more realistic settings such as GMMs with diagonal covariance clusters. It is unclear why this would form a major challenge in the presented framework as it technically just requires expanding the readout function to have parameters for the variances of similar dimensionality as the produced means in the readout function and loss functions based on an additional squared error loss term for the diagonal variances.
>
> >  What is the reason for not including a diagonal variance in the modeling which would make the approach more useful and seems to be straight forward to implement?
> ### Q1: TGMM performance on the diagonal variance case
> Thank you for your question. In fact, we evaluate TGMM on the diagonal variance case in Figure 6 of our paper, where we refer to this setting as the “anisotropic” case. We apologize for any confusion caused by our terminology. The full model formulation and experimental details are provided in Appendix H.1.
>
> As shown in these results, TGMM continues to outperform the EM algorithm in this setting. On the other hand, the spectral algorithm becomes unstable and is practically infeasible to implement under diagonal variance.
>
>
>
> > The analysis are purely considering synthetic data and it is unclear why the approach is not also tested on real data. This would substantially expand and verify the practical utility of the developed framework. Currently, the limited variance formulation makes it likely impractical for real data analyses.
>
> > What is the practical relevance of the procedure – it would strengthen the paper to consider real datasets and the quality of the learned clusters. This would however likely also require expanding the modeling procedure to at least have cluster centers with diagonal variance parameters. What is hampering the procedure to be expanded to such setting and evaluated on real data?
> ### Q2: Performance on real-world datasets
> Thank you for the advice. We did not include real-world data experiments in our submission primarily because of the following difficulties:
> - It is very hard to evaluate the estimation performance on those real-world tasks, as we do not know the ground truth. Consequently, the only meaningful metric would be the clustering accuracy.
> - Meta-training over real-world datasets is impossible since no ground truth parameters are available. In fact, most real-world data do not follow Gaussian mixtures in the first place.
>
> With those difficulties in mind, we test TGMM on the MNIST dataset with input dimension being $784$ (flattened pixels), where we construct two subtasks: The first one ($K=2$) is distinguishing 0-vs-1 (in terms of digit number), and the second one ($K=5$) is distinguishing between 0,1,2,3,4. Both are conducted directly on the test split of MNIST in an unsupervised fashion with the **corresponding TGMM models being meta-trained on synthetic $K=2$ and $K=5$ Gaussian mixture tasks, respectively**.
>
> |       | EM               | Spectral | TGMM              |
> |-------|------------------|----------|-------------------|
> | $K=2$ | $99.43$          | $98.96$  | $\mathbf{99.53}$  |
> | $K=5$ | $\mathbf{86.80}$ | $57.00$  | $84.70$           |
>
> The result shows that **TGMM reasonably generalizes to real-world datasets, even meta-trained only over synthetic Gaussian mixture distributions**, suggesting that the learned TGMM model operates as an algorithm instead of memorizing input distributions.

---

> ### Author Response · Authors · 2025-11-16
> **Response to Reviewer TJ7f (part 2/3)**
>
> > The presented performance metric is in terms of least squares error to ground truth cluster centers as well as mixing proportions. I appreciate the L2 metric in the main paper but think the log-likelihood metric considered in the supplementary should also be part of the main-paper as the EM-based procedure is based on optimizing the log-likelihood whereas the TGMM is minimizing the LS error of the parameters. This will provide a more comprehensive evaluation in the main paper in terms of recovering the underlying parameters and how well the model accounts for the underlying density.
>
> >  Furthermore, it would be interesting to display the mean and mixture proportion errors separately at least in the supplementary to understand if there are discrepancies in these errors. It would also be natural to quantify the mixture proportion errors using cross-entropy as optimized.
>
> ### Q3: On paper organizations
> Thank you for your suggestion. We do not include the log-likelihood metric in the main paper due to strict page limitations -- as you may have noticed, we were even unable to include the formal version of our main theorem in the main text. Since the primary focus of this work is parametric estimation, we choose to report only the least squares error in the main paper.
>
> We agree that log-likelihood is a meaningful metric, especially given its role in EM-based methods. As you suggest, we will include the log-likelihood results in the camera-ready version when space allows.
>
> ### Q4: On metric breakdown and design choices
> We report the $\ell_2$-error metric as a sum of mean estimation and mixture probability estimation primarily because their scales differ significantly: Typically the error component of mixture probability estimation is much smaller in magnitude, i.e., it accounts for usually less than $1\%$ of the total error. We report a metric breakdown in the following table where we train on sample size $N_\text{train} = 128$ and evaluate under both in-distribution setups as well as out-of-distribution setups with $d=32$ and $K=3$.
>
> Table 1: Metric breakdown of TGMM estimation error in $\ell_2$ and cross entropy
> |                      | $\ell_2$-error(mean) | $\ell_2$-error(mixture-prob) | cross-entropy(mixture-prob) |
> |----------------------|----------------------|------------------------------|-----------------------------|
> | $N_\text{test}=128$  | $1.49$               | $<0.01$                      | $1.08$                      |
> | $N_\text{test}=256$  | $1.50$               | $<0.01$                      | $1.07$                      |
> | $N_\text{test}=512$  | $1.47$               | $<0.01$                      | $1.07$                      |
> | $N_\text{test}=1024$ | $1.45$               | $<0.01$                      | $1.06$                      |
> | $N_\text{test}=2048$ | $1.40$               | $<0.01$                      | $1.06$                      |
>
> From Table 1 we have the following observations:
> - The $\ell_2$-error of mixture probability estimation is typically very small in magnitude and is much smaller than the mean estimation component in the total error metric. Moreover, as mixture probability is a relatively low-dimensional parameter (of dimension $K$) in comparison to the GMM means (of dimension $K \times d$), the estimation of mixture probaility is typically easier and the training loss of mixture probaility estimation plateaus much quicker than that of mean estimation, which in part explains why the $\ell_2$-error of mixture probability estimation is usually pretty small (i.e., of magnitude $0.001$ to $0.002$).
> - Regarding your comment of reporting cross-entropy metric (between the mixture probability estimation and ground truth), we did not choose this metric to characterize estimation error primarily due to the fact that cross entropy is lower bounded by the true entropy of the ground truth mixture probablity, which in our case is usually sampled from some distributions (i.e., refer to the TaskSampler algorithm 1) and is hence not a constant, thereby lacking a uniform value to indicate that the estimation is "good" (in comparison to $0$ which is a straghtforward way of measuring $\ell_2$-style metrics). Additionally, we report the cross-entropy metric in Table 1, where we can see a stable trend of cross-entropy error, showcasing the stability of mixture probability estimation. (In such cases that estimations are typically very accurate).

---

> ### Author Response · Authors · 2025-11-16
> **Response to Reviewer TJ7f (part 3/3)**
>
> > The loss function optimized in equation (2) contains an implicit weighting between accuracy of mean estimates by least squares and accuracy of cluster sizes by cross-entropy loss. How should these losses be balanced in general? And why not use the standard log-likelihood as loss?
> ### Q5: About the loss function
> Thank you for your questions. As noted in the response to Q4, the total loss is dominated by the least squares error on the mean estimates. We do not tune the relative weighting between the two terms, nor perform extensive hyperparameter selection. This is because the goal of this work is not to position TGMM as a state-of-the-art method for GMMs, but rather to demonstrate the expressive power of Transformers in unsupervised learning settings, using GMM as a case study.
>
> Since the likelihood loss only uses the data points' information, optimizing over such a loss is equivalent to MLE with parameters reparameterized via transformers. However, we want to highlight that **parameter estimation is not equivalent to likelihood maximization** in practice. As we show in Figure 10(c) in Appendix E, EM estimator achieved a slightly better likelihood in comparison to TGMM. However, the parameter estimation quality of EM is inferior to TGMM in nearly all settings. This divergence occurs because MLE prioritizes distributional fit (density matching) over parameter accuracy, especially when latent variables or non-identifiable structures are present. Therefore, we do not choose the standard log-likelihood as the loss function.
>
>
> > EM and spectral methods are widely used, but it would be relevant also to compare against a simple SGD procedure based on Adam or similar optimization minimizing the GMM likelihood for comparison. How would such simple procedure compare?
> ### Q6: On comparison with a (stochastic) gradient descent procedure optimizing likelihood
> Thanks for this comment. This is actually a classic problem that has been well-studied in the seminal work [1]. Where the authors characterized three types of EM-updates: Standard EM (the method used in our paper), Gradient EM (also known as first-order EM which we believe is the likelihood-optimizing gradient descent procedure you have mentioned) and Stochastic Gradient EM which is the stochastic optimization of gradient EM. Standard EM and Gradient EM were shown to behave empirically similarly with closely related theoretical characterizations (Please refer to Theorem 2 and Theorem 4 in [1] under its arxiv version). We have also implemented gradient EM (till convergence under learning rate $0.001$ and error tolerance $10^-7$) and report results in the following table:
>
> |                      |         |             | $\text{Acc}_\text{cluster}$ |                            |             | $\ell_2$-error |             |                            |
> |----------------------|---------|-------------|-----------------------------|----------------------------|-------------|----------------|-------------|----------------------------|
> |                      | EM      | Gradient EM | Spectral                    | TGMM($N_\text{train}=128$) | Gradient EM | EM             | Spectral    | TGMM($N_\text{train}=128$) |
> | $N_\text{test}=128$  | $71.10$ | $70.65$     | $98.44$                     | $98.29$                    | $5.12$      | $4.58$         | $0.39$      | $1.49$                     |
> | $N_\text{test}=256$  | $71.50$ | $71.00$     | $99.22$                     | $98.50$                    | $4.70$      | $4.72$         | $0.39$      | $1.50$                     |
> | $N_\text{test}=512$  | $71.81$ | $71.46$     | $99.60$                     | $98.67$                    | $4.71$      | $4.60$         | $0.33$      | $1.47$                     |
> | $N_\text{test}=1024$ | $72.79$ | $74.04$     | $100.00$                    | $99.10$                    | $4.20$      | $4.36$         | $0.13$      | $1.45$                     |
> | $N_\text{test}=2048$ | $74.57$ | $75.70$     | $100.00$                    | $99.61$                    | $4.01$      | $4.14$         | $0.05$      | $1.40$                     |
>
> As illustrated in the table, EM and Gradient EM behave quite similarly. We have also tested using adaptive gradient methods like Adam and it shows nearly identical results. Note that the sample size is relatively small as we scale it up to $2048$ and thus stochastic approximation is not necessary in this case.
>
> [1]. Balakrishnan, Sivaraman, Martin J. Wainwright, and Bin Yu. "Statistical guarantees for the EM algorithm: From population to sample-based analysis." (2017): 77-120.

---

> > ### Comment · Reviewer_TJ7f · 2025-11-24
> >
> > I thank the authors for their responses which helped clarify some of my concerns and I apologize for having missed that the diagonal variance model was indeed treated in the paper. I appreciate that the authors have included a simple MNIST real data experiment, further broken down performance, as well as included a gradient based EM procedure.
> >
> > While I understand that real datasets may not have ground truth clusters to compare against such real data experimentation can still investigate the ability to minimize the data log-likelihood, shed light on issues of local minima in the inference, as well as be used to evaluate performance in terms of predictive log-likelihood of test data to investigate the quality of the learned densities. All such analyses would strengthen the practical utility of the presented approach. I agree that the procedure may here be inferior simply by not as conventional EM be based on log-likelihood minimization and therefore produce inferior performance. However, the training could also have been log-likelihood based as opposed to the squared loss based minimization presently considered, simply optimizing for the data log-likehood which would have made direct comparison to standard GMM estimation performance more clear. These points could be further addressed by the authors in their rebuttal. I would consider a log-likelihood based loss implementation a natural loss for the transformer based procedure also complying with conventional GMM training. This would also help position the paper in light of conventional GMM estimation, and likely enhance the frameworks performance considering the log-likelihood metric. Importantly, a key aspect of GMM is not only clustering but also density estimation and log-likelihood based optimization can here naturally be quantified in performance on real data using the conventional predictive log-likelihood metric on test data.

---

> > > ### Author Response · Authors · 2025-11-29
> > > **Response to reviewer Tj7f (cont'd)**
> > >
> > > We would like to thank you for the response. We agree with you that directly using likelihood as the objective serves as yet another interesting comparison with likelihood-maximization approaches like EM. We have conducted preliminary empirical study via switching the learning objective to the **observed likelihood function** which aligns with the test set likelihood metric. The results are reported in the following table, which contains three different setups of TGMM:
> > > - **TGMM(est)**: This model is trained under the squared loss objective as proposed in the paper.
> > > - **TGMM(lik)**: This model is trained using the observed log-likelihood objective.
> > > - **TGMM(est+lik)**: This model is trained under a multi-task setup where the final objective is the summation of negative-log-likelihood and squared losses.
> > >
> > > Table 3: Detailed comparisons with TGMM variants
> > > |                      |         |          | $\text{Acc}_\text{cluster}$ |           |               |        |          | $\ell_2$-error |           |               |         |          | log-likelihood |           |               |
> > > |----------------------|---------|----------|-----------------------------|-----------|---------------|--------|----------|----------------|-----------|---------------|---------|----------|----------------|-----------|---------------|
> > > |                      | EM      | Spectral | TGMM(est)                   | TGMM(lik) | TGMM(est+lik) | EM     | Spectral | TGMM(est)      | TGMM(lik) | TGMM(est+lik) | EM      | Spectral | TGMM(est)      | TGMM(lik) | TGMM(est+lik) |
> > > | $N_\text{test}=128$  | $71.10$ | $98.44$  | $98.29$                     | $58.24$   | $93.86$       | $4.58$ | $0.39$   | $1.49$         | $7.34$    | $5.88$        | $-0.89$ | $-0.87$  | $-1.67$        | $-0.82$   | $-1.24$       |
> > > | $N_\text{test}=256$  | $71.50$ | $99.22$  | $98.50$                     | $59.11$   | $93.90$       | $4.72$ | $0.39$   | $1.50$         | $7.26$    | $5.45$        | $-0.41$ | $-0.40$  | $-0.90$        | $-0.35$   | $-0.57$       |
> > > | $N_\text{test}=512$  | $71.81$ | $99.60$  | $98.67$                     | $59.80$   | $95.61$       | $4.60$ | $0.33$   | $1.47$         | $7.18$    | $5.28$        | $-0.19$ | $-0.19$  | $-0.30$        | $-0.14$   | $-0.24$       |
> > > | $N_\text{test}=1024$ | $72.79$ | $100.00$ | $99.10$                     | $60.25$   | $96.78$       | $4.36$ | $0.13$   | $1.45$         | $7.04$    | $5.24$        | $-0.09$ | $-0.07$  | $-0.20$        | $-0.04$   | $-0.11$       |
> > > | $N_\text{test}=2048$ | $74.57$ | $100.00$ | $99.61$                     | $62.35$   | $97.45$       | $4.14$ | $0.05$   | $1.40$         | $7.33$    | $5.19$        | $-0.05$ | $-0.02$  | $-0.09$        | $-0.02$   | $-0.07$       |
> > >
> > > We have the following findings:
> > > - Firstly, using observed log-likelihood as the objective significantly improves the test-set log-likelihood metric, surpassing both EM and spectral. We believe that this phenomenon indicates that TGMM can learn to find some interesting likelihood-maximizing procedure.
> > > - Secondly, we have found both the parameter estimation quality, as well as the cluster recovery quality of TGMM(lik) to deteriorate in comparison to the original TGMM. We think this phenomenon is not surprising since training with log-likelihood objective indicates that we are operating in a *purely unsupervised meta-training regime* where no ground truth is provided during meta-training period.
> > > - Additionally, we provide preliminary insights via combining the likelihood-maximization objective and the squared loss objective and found the result to strike a balance between the two------improving the likelihood of square-loss-based-TGMM and the parameter estimation quality of likelihoo-based-TGMM. We believe more careful tuning of hyper-parameters will bring the possibility of achieving the best of the two, and we will provide further evidences in camera-ready versions of the paper.
> > >
> > > Finally, we would like to thank you once again for raising this though-provoking point. Your suggestions would lead to improved versions of our paper.

---

### Official Review · Reviewer_4R4H · 2025-11-03

**Soundness:** 3
**Presentation:** 2
**Contribution:** 2
**Rating:** 4
**Confidence:** 4

**Summary:**

The paper proposes TGMM, a transformer-based learning algorithm that estimates parameters of Gaussian Mixture Models across varying dimensions d and numbers of components K. Empirically, TGMM competes with or exceeds EM and matches spectral methods while handling cases ($K>d$) where spectral fails, and shows robustness to distribution and sample-size shifts. Theoretically, the authors provide interpretations of their algorithm by proving transformers can approximate EM updates and tensor power iterations used in spectral methods.

**Strengths:**

1. This paper proposes a new algorithm that uses transformer architectures for GMM learning.
2. The authors empirically validate the effectiveness, robustness and flexibility of TGMM, and provide theoretical justifications by proving approximability results of their method.

**Weaknesses:**

1. While the authors try to demonstrate the effectiveness of TGMM via experiments, it actually performs worse than spectral methods in Figure 2 and Figure 4\. These figures also fail to decouple statistical errors and algorithmic errors. Spectral methods, as theoretically guaranteed, will have errors converging towards 0 with enough samples. The authors should at least show the same (empirically or theoretically) for TGMM. To demonstrate effectiveness, in my opinion, they should also compare sample complexity of TGMM to baselines and show improvements.
2. While spectral methods fail in the degenerate case of $K\>d$, there are sum-of-squares methods that overcome this issue (see \[1\]). Similarly, while EM easily gets trapped in local minima, people have proposed pruning based variants to overcome this (e.g., see \[2\]). The authors should conduct more comprehensive baseline comparisons before claiming performance superiority.,

References.

1. Liu, Allen, and Jerry Li. "Clustering mixtures with almost optimal separation in polynomial time." Proceedings of the 54th Annual ACM SIGACT Symposium on Theory of Computing. 2022\.
2. Dasgupta, Sanjoy, and Leonard Schulman. "A two-round variant of em for gaussian mixtures." arXiv preprint arXiv:1301.3850 (2013).

**Questions:**

1. What is the data used for training TGMM? How does the model training cost compare in quantity with the computational cost of EM/spectral methods?
2. How does the model perform in small separation regimes close to the information theoretical lower bound (see \[1\])?

References:

1. Regev, Oded, and Aravindan Vijayaraghavan. "On learning mixtures of well-separated gaussians." 2017 IEEE 58th Annual Symposium on Foundations of Computer Science (FOCS). IEEE, 2017\.

---

> ### Author Response · Authors · 2025-11-16
> **Response to Reviewer 4R4H (part 1/4)**
>
> We would like to thank you for your valuable and professional comments. We believe that you have pointed out multiple important aspects that would lead to improved versions of our paper. From a broader perspective, we would like to emphasize that the goal of this paper is not to position TGMM as a state-of-the-art method for solving GMMs. Indeed, we found empirically that when the spectral method is applicable, it generally outperforms TGMM across most settings.
> The primary purpose of this work is to investigate the expressive power of Transformers as an in-context learning algorithm in unsupervised learning tasks, using GMMs as a representative example.
> Below we address your specific points.
>
> > While the authors try to demonstrate the effectiveness of TGMM via experiments, it actually performs worse than spectral methods in Figure 2 and Figure 4. These figures also fail to decouple statistical errors and algorithmic errors. Spectral methods, as theoretically guaranteed, will have errors converging towards 0 with enough samples. The authors should at least show the same (empirically or theoretically) for TGMM. To demonstrate effectiveness, in my opinion, they should also compare sample complexity of TGMM to baselines and show improvements.
> ### Q1: On the statistical errors and algorithmic errors
>
> Thank you for your insightful comment.
> Since TGMM is based on a learned model rather than a fixed algorithm, we cannot rigorously define "algorithmic error" in the traditional sense. As a proxy, we use the depth of the Transformer (i.e., the number of layers $L$ ) to reflect modeling capacity-deeper models typically have stronger representational power and can be viewed as corresponding to more iterations.
> To better understand the interplay between statistical and algorithmic errors, we conducted an experiment over $K = 3$ GMM task with  $d = 32$ using GPT-2 style transformer backbones with $L \in \{1,6,12\}$ layers, respectively. The models are trained over sample size randomly chosen from the interval $[64, 128]$. The results are shown in the following table, with each row demonstrating evaluations of the learned TGMM model over GMM problems with increasing sample sizes:
>
> Table 1: Empirical assessments of (learned) TGMM as a GMM-solving algorithm
> |            | $\text{Acc}_\text{cluster}$ |             |             | $\ell_2$-error |             |             |
> |------------|-------------|-------------|-------------|----------------|-------------|-------------|
> |            | $L=1$       | $L=6$       | $L=12$      | $L=1$          | $L=6$       | $L=12$      |
> | $N_\text{test}=128$    | $85.76$     | $89.84$     | $98.29$     | $3.61$         | $2.68$      | $1.49$      |
> | $N_\text{test}=256$    | $85.13$     | $94.41$     | $98.50$     | $3.80$         | $2.30$      | $1.50$      |
> | $N_\text{test}=512$    | $85.43$     | $95.48$     | $98.67$     | $3.71$         | $2.09$      | $1.47$      |
> | $N_\text{test}=1024$   | $84.82$     | $97.46$     | $99.10$     | $3.69$         | $2.14$      | $1.45$      |
> | $N_\text{test}=2048$   | $85.71$     | $96.36$     | $99.61$     | $3.63$         | $2.11$      | $1.40$      |
>
> We would like to emphasize two major implications of Table 1:
> - TGMM generalizes in an out-of-distribution sense with recovery error converging towards $0$ as shown in $L=6$ as well as $L=12$ columns above. We believe this would answer your question that (learned) TGMMs behave reasonably as a valid GMM solving algorithm instead of some overfitting-to-training-distribution tricks.
> - Architectural complexities play an important role. As shown in $L=1$ column in Table 1, when TGMM is learned without proper depth, the network capacity might be insufficient to learn a GMM-solution, i.e., both the recovery errors and the estimation errors do not shrink as the sample size grows.
>
> We believe that the above evidences reasonably demonstrate that TGMM is a sound learning-to-solve-GMM approach.

---

> ### Author Response · Authors · 2025-11-16
> **Response to Reviewer 4R4H (part 2/4)**
>
> > While spectral methods fail in the degenerate case of $K > d$, there are sum-of-squares methods that overcome this issue (see [1]). Similarly, while EM easily gets trapped in local minima, people have proposed pruning based variants to overcome this (e.g., see [2]). The authors should conduct more comprehensive baseline comparisons before claiming performance superiority.
> ### Q2: Comparison with other methods: Sample complexities and refined baselines
>
> Thanks for pointing out two notable baselines. During investigations we found the following:
> - While the sum-of-squares method in [1] is beautiful in theory, it involves many complex tensor operations and combinatorial calculations that make practical implementations extremely slow, despite its poly-time nature (Sometimes algorithmic developments from TCS community inevitably have such issues). In an attempt to reproduce that algorithm, we found the computational cost too high to run a large number of simulation runs even in moderate-dimensional settings (i.e., $d=32$). Moreover, as noted in [1], achieving high-accuracy parameter estimates requires using their output as a warm start for another refinement algorithm, which further complicates end-to-end evaluation. Therefore, we do not report the results using the algorithm in [1].
> - For the two-round EM algorithm from [2], we have implemented it and conducted comparisons on $d=3$ GMM tasks with $d=32$. The results are shown in the following table:
>
> Table 2: More comparisons with baseline methods
> |                      |         |              | $\text{Acc}_\text{cluster}$ |                            |              | $\ell_2$-error |             |                            |
> |----------------------|---------|--------------|-----------------------------|----------------------------|--------------|----------------|-------------|----------------------------|
> |                      | EM      | Two-Round EM | Spectral                    | TGMM($N_\text{train}=128$) | Two-Round EM | EM             | Spectral    | TGMM($N_\text{train}=128$) |
> | $N_\text{test}=128$  | $71.10$ | $80.70$      | $98.44$                     | $98.29$                    | $3.04$       | $4.58$         | $0.39$      | $1.49$                     |
> | $N_\text{test}=256$  | $71.50$ | $81.90$      | $99.22$                     | $98.50$                    | $2.99$       | $4.72$         | $0.39$      | $1.50$                     |
> | $N_\text{test}=512$  | $71.81$ | $82.36$      | $99.60$                     | $98.67$                    | $2.95$       | $4.60$         | $0.33$      | $1.47$                     |
> | $N_\text{test}=1024$ | $72.79$ | $84.77$      | $100.00$                    | $99.10$                    | $2.94$       | $4.36$         | $0.13$      | $1.45$                     |
> | $N_\text{test}=2048$ | $74.57$ | $87.24$      | $100.00$                    | $99.61$                    | $2.86$       | $4.14$         | $0.05$      | $1.40$                     |
>
> We have the following findings:
> - Two-round EM method [2] indeed improves standard EM, but the performance gain is not enough to match TGMM or Spectral.
> - Regarding sample complexities, we found it tricky to empirically characterize them using performance metrics as typically those associating theoretical bounds involve constants that might contribute significantly to practical scenarios, especially when the sample size is only moderately large as depicted in Table 2. Therefore, we think that from a performance-centric point of view, TGMM performs slightly worse than the spectral algorithm but better than EM-style variants.
> But we tend not to give formal conclusions about sample complexity comparisons. **We are open to more detailed discussion about how to measure sample complexity properly in our empirical settings.**

---

> ### Author Response · Authors · 2025-11-16
> **Response to Reviewer 4R4H (part 3/4)**
>
> > What is the data used for training TGMM? How does the model training cost compare in quantity with the computational cost of EM/spectral methods?
> ### Q3: On the data and computational cost in TGMM training
> Thanks for your question.
> **Data:**
> The data used to train TGMM is generated by the TaskSampler algorithm in Algorithm 1 of our paper. In a nutshell, a fresh GMM sample is constructed at each training step by the TaskSampler algorithm that determines its number of components $K$, the mixture probabilities and the mean vectors that are sampled from specific distributions. The sampling distributions used in our experiments are detailed in section 3.1, where we also report some adjustments to avoid getting samples of excessively low separation which might cause inefficient training.
>
> **Computational cost:**
> Meta-training is an essential step to build transformer models that are capable of solving both supervised machine learning problems like in [3] or unsupervised learning problems like TGMM. In contrast, classical algorithms do not require extra training and thus the complexity of TGMM meta-training is not comparable to that of classic algorithms.
>
> For further discussion, we compare the inference costs among different algorithms.
> As we mentioned above, here we use the number of layers $L$ as the "algorithm iteration steps" for TGMM.
> We assume running both classical algorithms for $L$ steps (i.e., $L$ steps of EM iteration in EM algorithm and $L$ steps of robust tensor power iteration in the spectral algorithm) and compare with an $L$-layer transformer backbone. Over an input task with $N$ samples with dimension $d$ and $K$ components, we further let the hidden dimension of the transformer backbone to be $O(d)$. The resulting inference complexities are:
> - $O(L(N^2 d + N d^2))$ for transformer (TGMM), note that only the output layer involves a linear projection to $K$ component estimations, unless $K$ is extremely large, this step has negligible complexity.
> - $O(LNKd)$ for EM algorithm.
> - $O(Ld^3)$ for spectral algorithm. Here, we take mainly the complexity of doing tensor power iterations and omit those in auxiliary construction steps. As in practice, this is often the dominating cost.
>
> When $K$ is very small, it is clear that EM is the most efficient solution. When $K$ gets large (but note that $K < N$), the efficiency gap between EM and TGMM is narrowing. As for the spectral algorithm, as $d$ must be greater than $K$, it is often the case that spectral algorithm is the most inefficient one (According to our experience in implementing tensor-decompositions, usually having dimension $d>5000$ would drastically drop the practical efficiency. Meanwhile, modern transformers typically operate on a much bigger input dimension such as the size of language vocabularies and are therefore much more efficent in high dimensional setups in practice). Therefore, for tasks with large $K$ and $d$, the efficiency order would be roughly EM > TGMM > Spectral.

---

> ### Author Response · Authors · 2025-11-16
> **Response to Reviewer 4R4H (part 4/4)**
>
> > How does the model perform in small separation regimes close to the information theoretical lower bound (see [4])? -->
> ### Q4: On the performance of TGMM near the small separation threshold
>
> Thanks for raising this thought-provoking question. While [4] establishes a theoretical lower bound for GMM estimation (Theorem 3.1), it only specifies the order of the required separation threshold as $O(\sqrt{\log(K)})$, without providing an explicit constant or a complete formula. In practice, this makes direct implementation challenging: even for very large K (e.g., $10^7$ to $10^8$) the term $\sqrt{\log(K)}$ remains relatively small, making it difficult to disentangle the influence of the unknown constant factor from the asymptotic main term.
> Upon our primary investigations, we think that such scenarios, while sound in theory, might not be easily constructed to fit in empirical validations.
> Therefore, we instead resort to another limiting scenario which is the recovery threshold stated in [5] where an exact phase transition of mixture membership recovery was introduced, measured by the term $$\Delta_n = \sqrt{\left(1 + \sqrt{1 + \dfrac{2d}{n\log n}}\right)\log n} \quad .$$  For a two-component isotropic GMM with corresponding means separated by $2\Delta_n$ in Euclidean distance, exact recovery is possible with high probability with actual separations exceeding this threshold. While the problem in [5] is different from that in [4], we believe it still serves as a good testbed for testing GMM algorithms under small-separation scenarios that are efficiently reproducible.
> We follow [5] that uses the following configurations with $n=128$ and let $\Delta_n = \sqrt{(1 + \sqrt{a})\log n}$  and  $d = bn\log n$, where the phase transition threshold for exact recovery happens at $a = 1 + 2b$. We test EM, Spectral and TGMM over four configurations right on the line and report the results in the following table
>
> |              |       | $\text{Acc}_\text{cluster}$ |       |      | $\ell_2$-error |      |
> |--------------|-------|-----------------------------|-------|------|----------------|------|
> |              | EM    | Spectral                    | TGMM  | EM   | Spectral       | TGMM |
> | $a=2, b=0.5$ | $57.76$ | $62.10$                      | $56.92$ | $0.10$ | $0.12$           | $0.09$ |
> | $a=3, b=1$   | $53.64$ | $63.24$                       | $54.59$ | $0.11$ | $0.12$           | $0.07$ |
> | $a=4, b=1.5$ | $52.69$ | $62.30$                       | $53.72$ | $0.10$ | $0.13$           | $0.08$ |
> | $a=5, b=2$   | $52.83$ | $63.31$                       | $53.30$ | $0.09$ | $0.12$           | $0.07$ |
>
> According to our experiments, we find that TGMM performs on par with EM near the recovery threshold under the recovery metric (cluster accuracy), while performing worse than the spectral method. Looking at estimation accuracy, we find that three methods are similar in general; under the small-separation case, the actual mean vectors are high-dimensional (with $d\approx 1000$) with a relatively small norm, causing the coordinate values to be typically of order $10^-2$ resulting in small estimation errors that are not very meaningful. To conclude, when the underlying problem is hard, TGMM doesn't show extra benefits but performs on par with baselines. Finally, we would like to point out that we did not use tailor-made training recipes for these phase-transition experiments (i.e., we use the standard TaskSampler paradigm as in our paper) and we think it might also be an interesting direction to explore whether we can device some legitimate task sampling scheme that leads to better performance.
>
> [1] Liu Allen, and Jerry Li. "Clustering mixtures with almost optimal separation in polynomial time." Proceedings of the 54th Annual ACM SIGACT Symposium on Theory of Computing. 2022.
>
> [2] Dasgupta  Sanjoy, and Leonard Schulman. "A two-round variant of em for gaussian mixtures." arXiv preprint arXiv:1301.3850 (2013).
>
> [3] Yu Bai et al. "Transformers as statisticians: Provable in-context learning with in-context algorithm selection." Advances in Neural Information Processing Systems(NeurlIPS), 2023.
>
> [4] Regev  Oded, and Aravindan Vijayaraghavan. "On learning mixtures of well-separated gaussians." 2017 IEEE 58th Annual Symposium on Foundations of Computer Science (FOCS). IEEE, 2017.
>
> [5] Ndaoud Mohamed. "Sharp optimal recovery in the two component Gaussian mixture model." The Annals of Statistics 50.4 (2022): 2096-2126.

---

> ### Comment · Reviewer_4R4H · 2025-11-28
>
> Thank the authors for adding comprehensive experimental comparisons. For the statistical error experiments, demonstrating that the L2 error of Transformers decreases with the numer of layers seems a reasonable, albeit still not very satisfying, solution to me. The number of layers still goes towards infinity if one want a model that can fit GMMs with arbitrary $N$, marking a large gap between the proposed approach and spectral methods.
>
> I will raise my score.

---

### Meta-Review · Area_Chair_dH7t · 2026-01-06

**Summary:**

This paper focuses on the expressiveness of transformers for unsupervised learning with Gaussian Mixture Models (GMM). It introduces TGMM, a transformer-based approach that jointly solves multiple GMM tasks with a shared backbone, and empirically shows that TGMM outperforms classical EM and spectral methods while remaining robust to distribution shifts. It also provides theoretical guarantees showing that transformers can approximate EM and cubic tensor power iterations.

**Reviewer Concerns:**

The reviewers were mainly concerned about the effectiveness of TGMM and its performance comparison with other baselines and existing works. The authors have performed additional experiments during the rebuttal, which largely addresses the concerns. It also elaborates the difference between this work and another existing work [He et al 2025], which clarifies the technical contribution of this work.

**Reviewer Scores:**

The original scores were 4/6/8/2, while the first reviewer indicated that they would raise the score. The fourth reviewer did not respond during the rebuttal, however, their concerns were mostly addressed.

---

### Decision · Program_Chairs · 2026-01-26

Accept (Poster)